



# 1 Global Carbon Budget 2023

Pierre Friedlingstein 1,2, Michael O'Sullivan 1, Matthew W. Jones 3, Robbie M. Andrew 4, Dorothee C. E.
Bakker 5, Judith Hauck 6, Peter Landschützer 7, Corinne Le Quéré 3, Ingrid T. Luijkx 8, Glen P. Peters 4,
Wouter Peters 8,9, Julia Pongratz 10,11, Clemens Schwingshackl 10, Stephen Sitch 1, Josep G. Canadell 12,
Philippe Ciais 13, Robert B. Jackson 14, Simone R. Alin 15, Peter Anthoni 16, Leticia Barbero 17, Nicholas R.
Bates 18,19, Meike Becker 20,21, Nicolas Bellouin 22, Bertrand Decharme 23, Laurent Bopp 2, Ida Bagus
Mandhara Brasika 1,24, Patricia Cadule 25, Matthew A. Chamberlain 26, Naveen Chandra 27, Thi-Tuyet-Trang
Chau 13, Frédéric Chevallier 13, Louise P. Chini 28, Margot Cronin 29, Xinyu Dou 30, Kazutaka Enyo 31,
Wiley Evans 32, Stefanie Falk 10, Richard A. Feely 15, Liang Feng 33,34, Daniel. J. Ford 1, Thomas Gasser 35,
Josefine Ghattas 13, Thanos Gkritzalis 7, Giacomo Grassi 36, Luke Gregor 37, Nicolas Gruber 38, Özgür
Gürses 6, Ian Harris 39, Matthew Hefner 40,41, Jens Heinke 42, Richard A. Houghton 43, George C. Hurtt 44,
Yosuke Iida 31, Tatiana Ilyina 11, Andrew R. Jacobson 45,46, Atul Jain 47, Tereza Jarníková 48, Annika Jersild
11, Fei Jiang 49, Zhe Jin 50,51, Fortunat Joos 52,53, Etsushi Kato 54, Ralph F. Keeling 55, Daniel Kennedy 56,
Kees Klein Goldewijk 57, Jürgen Knauer 58,12, Jan Ivar Korsbakken 4, Arne Körtzinger 59, Xin Lan 45,46,
Nathalie Lefèvre 60, Hongmei Li 11, Junjie Liu 61,62, Zhiqiang Liu 63, Lei Ma 28, Greg Marland 40,41,
Nicolas Mayot 64, Patrick C. McGuire 65, Galen A. McKinley 66, Gesa Meyer 67, Eric J. Morgan 55, David R.
Munro 45,68, Shin-Ichiro Nakaoka 69, Yosuke Niwa 69,70, Kevin M. O'Brien 71,15, Are Olsen 20,21,
Abdirahman M. Omar 72,21, Tsuneo Ono 73, Melf E. Paulsen 59, Denis Pierrot 74, Katie Pocock 75, Benjamin
Poulter 76, Carter M. Powis 77, Gregor Rehder 78, Laure Resplandy 79, Eddy Robertson 80, Christian
Rödenbeck 81, Thais M Rosan 1, Jörg Schwinger 21,82, Roland Séférian 83, T. Luke Smallman 33, Stephen M.
Smith 77, Reinel Sospedra-Alfonso 84, Qing Sun 52,53, Adrienne J. Sutton 15, Colm Sweeney 46, Shintaro
Takao 69, Pieter P. Tans 85, Hanqin Tian 86, Bronte Tilbrook 87,88, Hiroyuki Tsujino 89, Francesco Tubiello
90, Guido R. van der Werf 8, Erik van Ooijen 87, Rik Wanninkhof 74, Michio Watanabe 91, Cathy Wimart-
Rousseau 59, Dongxu Yang 92, Xiaojuan Yang 93, Wenping Yuan 94, Xu Yue 95, Sönke Zaehle 81, Jiye Zeng
69, Bo Zheng 96
Faculty of Environment, Science and Economy, University of Exeter, Exeter EX4 4QF, UK
Laboratoire de Météorologie Dynamique / Institut Pierre-Simon Laplace, CNRS, Ecole Normale Supérieure /
Université PSL, Sorbonne Université, Ecole Polytechnique, Paris, France
Tyndall Centre for Climate Change Research, School of Environmental Sciences, University of East Anglia,
Norwich Research Park, Norwich NR4 7TJ, UK
CICERO Center for International Climate Research, Oslo 0349, Norway
School of Environmental Sciences, University of East Anglia, Norwich NR4 7TJ, UK
Alfred-Wegener-Institut, Helmholtz-Zentrum für Polar- und Meeresforschung, Am Handelshafen 12, 27570
Bremerhaven
VLIZ Flanders Marine Institute, Jacobsenstraat 1, 8400, Ostend, Belgium
Wageningen University, Environmental Sciences Group, P.O. Box 47, 6700AA, Wageningen, The
Netherlands



University of Groningen, Centre for Isotope Research, Groningen, The Netherlands
Ludwig-Maximilians-Universität München, Luisenstr. 37, 80333 München, Germany
Max Planck Institute for Meteorology, Bundesstraße 53, 20146 Hamburg, Germany
CSIRO Environment, Canberra, ACT 2101, Australia
Laboratoire des Sciences du Climat et de l'Environnement, LSCE/IPSL, CEA-CNRS-UVSQ, Université
Paris-Saclay, F-91198 Gif-sur-Yvette, France
Department of Earth System Science, Woods Institute for the Environment, and Precourt Institute for
Energy, Stanford University, Stanford, CA 94305–2210, United States of America
National Oceanic and Atmospheric Administration, Pacific Marine Environmental Laboratory
(NOAA/PMEL), 7600 Sand Point Way NE, Seattle, WA 98115, USA
Karlsruhe Institute of Technology, Institute of Meteorology and Climate Research/Atmospheric
Environmental Research, 82467 Garmisch-Partenkirchen, Germany
Rosenstiel School of Marine Atmospheric and Earth Science, Cooperative Institute for Marine and
Atmospheric Studies (CIMAS), University of Miami, 4600 Rickenbacker Causeway, Miami, FL, USA
School of Ocean Futures, Julie Ann Wrigley Global Futures Laboratory, Arizona State University, Tempe,
Arizona, AZ 85287-5502, USA
Bermuda Institute of Ocean Sciences (BIOS), 17 Biological Lane, St. Georges, GE01, Bermuda
Geophysical Institute, University of Bergen, Allégaten 70, 5007 Bergen, Norway
Bjerknes Centre for Climate Research, Bergen, Norway
Department of Meteorology, University of Reading, Reading, RG6 6BB, UK
CNRM, Université de Toulouse, Météo-France, CNRS, Toulouse, France
Faculty of Marine Science & Fisheries, University of Udayana, Bali 80361, Indonesia
CNRS, Institut Pierre-Simon Laplace, Sorbonne Université, Paris, France
CSIRO Environment, Hobart, TAS, Australia
Research Institute for Global Change, JAMSTEC, 3173-25 Showa-machi, Kanazawa, Yokohama, 236-0001,
Japan
Department of Geographical Sciences, University of Maryland, College Park, MD 20742, USA
Marine Institute, Rinville, Oranmore, Co Galway H91 R673, Ireland
Department of Earth System Science, Tsinghua University, Beijing, China
Japan Meteorological Agency, 3-6-9 Toranomon, Minato City, Tokyo 105-8431, Japan
Hakai Institute, 1713 Hyacinthe Bay Rd, Heriot Bay, BC, V0P 1H0, Canada
National Centre for Earth Observation, University of Edinburgh, Edinburgh, EH9 3FE, UK
School of Geosciences, University of Edinburgh, UK
International Institute for Applied Systems Analysis (IIASA), Schlossplatz 1, A-2361 Laxenburg, Austria
European Commission, Joint Research Centre, 21027 Ispra (VA), Italy
ETH Zürich, Switzerland
Environmental Physics Group, Institute of Biogeochemistry and Pollutant Dynamics and Center for Climate
Systems Modeling (C2SM), ETH Zürich, Switzerland
NCAS-Climate, Climatic Research Unit, School of Environmental Sciences, University of East Anglia,
Norwich Research Park, Norwich, NR4 7TJ, UK



Research Institute for Environment, Energy, and Economics, Appalachian State University, Boone, North
Carolina, USA
Department of Geological and Environmental Sciences, Appalachian State University, Boone, North
Carolina, USA
Potsdam Institute for Climate Impact Research (PIK), member of the Leibniz Association, P.O. Box 60 12
03, 14412 Potsdam, Germany
Woodwell Climate Research Center, Falmouth, MA 02540, USA
Department of Geographical Sciences, University of Maryland, College Park, Maryland 20742, USA
Cooperative Institute for Research in Environmental Sciences (CIRES), University of Colorado, Boulder, CO
80305, USA
National Oceanic and Atmospheric Administration, Global Monitoring Laboratory (NOAA/GML), 325
Broadway R/GML, Boulder, CO 80305, USA
Department of Atmospheric Sciences, University of Illinois, Urbana, IL 61821, USA
School of Environmental Sciences, University of East Anglia, Norwich Research Park, Norwich NR4 7TJ,
UK
Jiangsu Provincial Key Laboratory of Geographic Information Science and Technology, International
Institute for Earth System Science, Nanjing University, Nanjing, 210023, China
State Key Laboratory of Tibetan Plateau Earth System and Resource Environment, Institute of Tibetan
Plateau Research, Chinese Academy of Sciences, Beijing 100101, China
Institute of Carbon Neutrality, Peking University, Beijing 100871, China
Climate and Environmental Physics, Physics Institute, University of Bern, Bern, Switzerland
Oeschger Centre for Climate Change Research, University of Bern, Bern, Switzerland
Institute of Applied Energy (IAE), Minato-ku, Tokyo 105-0003, Japan
University of California, San Diego, Scripps Institution of Oceanography, La Jolla, CA 92093-0244, USA
National Center for Atmospheric Research, Climate and Global Dynamics, Terrestrial Sciences Section,
Boulder, CO 80305, USA
Utrecht University, Faculty of Geosciences, Department IMEW, Copernicus Institute of Sustainable
Development, Heidelberglaan 2, P.O. Box 80115, 3508 TC, Utrecht, the Netherlands
Hawkesbury Institute for the Environment, Western Sydney University, Penrith, New South Wales, Australia
GEOMAR Helmholtz Centre for Ocean Research, Wischhofstr. 1-3, 24148 Kiel, Germany
LOCEAN/IPSL laboratory, Sorbonne Université, CNRS/IRD/MNHN, Paris, France
NASA JPL, USA
Caltech, USA
CMA Key Open Laboratory of Transforming Climate Resources to Economy, Chongqing Institute of
Meteorological Sciences, Chongqing 401147, China
University of East Anglia, Norwich, UK
Department of Meteorology & National Centre for Atmospheric Science (NCAS), University of Reading,
Reading, United Kingdom
Columbia University, USA
Climate Research Division, Environment and Climate Change Canada, Victoria, BC, Canada



National Oceanic and Atmospheric Administration/Global Monitoring Laboratory (NOAA/GML), 325
Broadway R/GML, Boulder, CO 80305, USA
Earth System Division, National Institute for Environmental Studies, 16-2 Onogawa, Tsukuba, Ibaraki, 305-
8506 Japan
Department of Climate and Geochemistry Research, Meteorological Research Institute, 1-1 Nagamine,
Tsukuba, Ibaraki 305-0052, Japan
Cooperative Institute for Climate, Ocean and Ecosystem Studies (CICOES), University of Washington,
Seattle, WA 98105, USA
NORCE Norwegian Research Centre, Jahnebakken 5, 5007 Bergen, Norway
Marine Environment Division, Fisheries Resources Institute, Japan Fisheries Research and Education
Agency, 2-12-4 Fukuura, Kanazawa-Ku, Yokohama 236-8648, Japan
National Oceanic & Atmospheric Administration, Atlantic Oceanographic & Meteorological Laboratory
(NOAA/AOML), 4301 Rickenbacker Causeway, Miami, FL 33149, USA
Hakai Institute, 1713 Hyacinthe Bay Rd, Heriot Bay, BC, V0P 1H0, Canada
NASA Goddard Space Flight Center, Biospheric Sciences Laboratory, Greenbelt, Maryland 20771, USA
Smith School for Enterprise and the Environment, University of Oxford, Oxford, UK
Leibniz Institute for Baltic Sea Research Warnemünde (IOW), Seestrasse 15, 18119 Rostock, Germany
Princeton University, Department of Geosciences and Princeton Environmental Institute, Princeton, NJ, USA
Met Office Hadley Centre, FitzRoy Road, Exeter EX1 3PB, UK
Max Planck Institute for Biogeochemistry, P.O. Box 600164, Hans-Knöll-Str. 10, 07745 Jena, Germany
NORCE Climate & Environment, Jahnebakken 5, 5007 Bergen, Norway
CNRM (Météo-France/CNRS)-UMR 3589, Toulose, France
Canadian Centre for Climate Modelling and Analysis, Victoria BC, Canada
Institute of Arctic and Alpine Research, University of Colorado, Boulder, CO 80309, USA
Schiller Institute of Integrated Science and Society, Department of Earth and Environmental Sciences,
Boston College, Chestnut Hill, MA 02467, USA
CSIRO Environment, Castray Esplanade, Hobart, Tasmania 7004, Australia
Australian Antarctic Partnership Program, University of Tasmania, Hobart, Australia
JMA Meteorological Research Institute, Japan
Statistics Division, Food and Agriculture Organization of the United Nations, Via Terme di Caracalla, Rome
00153, Italy
Japan Agency for Marine-Earth Science and Technology (JAMSTEC), 3173-25, Showa-machi, Kanazawa-
ku, Yokohama, 236-0001, Japan
Institute of Atmospheric Physics, Chinese Academy of Sciences
Environmental Sciences Division and Climate Change Science Institute, Oak Ridge National Laboratory,
Oak Ridge, TN, 37831, USA
School of Atmospheric Sciences, Sun Yat-sen University, Zhuhai, Guangdong 510245, China
School of Environmental Science and Engineering, Nanjing University of Information Science and
Technology (NUIST)



Shenzhen Key Laboratory of Ecological Remediation and Carbon Sequestration, Institute of Environment
and Ecology, Tsinghua Shenzhen International Graduate School, Tsinghua University, Shenzhen 518055, China

*Correspondence to*: Pierre Friedlingstein (p.friedlingstein@exeter.ac.uk)
**Abstract**
Accurate assessment of anthropogenic carbon dioxide ($CO_2$) emissions and their redistribution among the
atmosphere, ocean, and terrestrial biosphere in a changing climate is critical to better understand the global
carbon cycle, support the development of climate policies, and project future climate change. Here we describe
and synthesise data sets and methodology to quantify the five major components of the global carbon budget
and their uncertainties. Fossil $CO_2$ emissions ($E_{FOS}$) are based on energy statistics and cement production data,
while emissions from land-use change ($E_{LUC}$), mainly deforestation, are based on land-use and land-use change
data and bookkeeping models. Atmospheric $CO_2$ concentration is measured directly, and its growth rate ($G_{ATM}$)
is computed from the annual changes in concentration. The ocean $CO_2$ sink ($S_{OCEAN}$) is estimated with global
ocean biogeochemistry models and observation-based $fCO_2$-products. The terrestrial $CO_2$ sink ($S_{LAND}$) is
estimated with dynamic global vegetation models. Additional lines of evidence on land and ocean sinks are
provided by atmospheric inversions, atmospheric oxygen measurements and Earth System Models. The
resulting carbon budget imbalance ($B_{IM}$), the difference between the estimated total emissions and the
estimated changes in the atmosphere, ocean, and terrestrial biosphere, is a measure of imperfect data and
understanding of the contemporary carbon cycle. All uncertainties are reported as ±1σ.
For the year 2022, $E_{FOS}$ increased by 1.0% relative to 2021, with fossil emissions at $10.2 \pm 0.5$ GtC yr$^{-1}$ ($9.9 \pm$
$0.5$ GtC yr$^{-1}$ when the cement carbonation sink is included), $E_{LUC}$ was $1.2 \pm 0.7$ GtC yr$^{-1}$, for a total
anthropogenic $CO_2$ emission (including the cement carbonation sink) of $11.1 \pm 0.8$ GtC yr$^{-1}$ ($40.7 \pm 3.2$ GtCO$_2$
yr$^{-1}$). Also, for 2022, $G_{ATM}$ was $4.6 \pm 0.2$ GtC yr$^{-1}$ ($2.18 \pm 0.1$ ppm yr$^{-1}$), $S_{OCEAN}$ was $2.8 \pm 0.4$ GtC yr$^{-1}$ and
$S_{LAND}$ was $3.8 \pm 0.8$ GtC yr$^{-1}$, with a $B_{IM}$ of -0.1 GtC yr$^{-1}$ (i.e. total estimated sources marginally too low or
sinks too high). The global atmospheric $CO_2$ concentration averaged over 2022 reached $417.1 \pm 0.1$ ppm.
Preliminary data for 2023, suggest an increase in $E_{FOS}$ relative to 2022 of +1.2% (0.2% to 2.2%) globally, and
atmospheric $CO_2$ concentration reaching 419.2 ppm, more than 50% above pre-industrial level (around 278
ppm in 1750). Overall, the mean and trend in the components of the global carbon budget are consistently
estimated over the period 1959-2022, with a near-zero overall budget imbalance, although discrepancies of up
to around 1 GtC yr$^{-1}$ persist for the representation of annual to semi-decadal variability in $CO_2$ fluxes.
Comparison of estimates from multiple approaches and observations shows: (1) a persistent large uncertainty
in the estimate of land-use changes emissions, (2) a low agreement between the different methods on the
magnitude of the land $CO_2$ flux in the northern extra-tropics, and (3) a discrepancy between the different
methods on the strength of the ocean sink over the last decade. This living data update documents changes in
the methods and data sets used in this new global carbon budget and the progress in understanding of the
global carbon cycle compared with previous publications of this data set.



## Executive Summary

**Global fossil CO$_2$ emissions (including cement carbonation) are expected to further increase in 2023, to 1.5% above their pre-COVID-19 pandemic 2019 level.** The 2022 emission increase was 0.08 GtC yr$^{-1}$ (0.31 GtCO$_2$ yr$^{-1}$) relative to 2021, bringing 2022 fossil CO$_2$ emissions to 9.9 ± 0.5 GtC yr$^{-1}$ (36.3 ± 1.8 GtCO$_2$ yr$^{-1}$), virtually equal to the emissions level of 2019. Preliminary estimates based on data available suggest fossil CO$_2$ emissions to increase further in 2023, by 1.2% relative to 2022 (0.2% to 2.2%), bringing emissions to 10.0 GtC yr$^{-1}$ (36.8 GtCO$_2$ yr$^{-1}$), 1.5% above the 2019 level.

Emissions from coal, oil, and gas in 2023 are expected to be slightly above their 2022 levels (by 1.1%, 1.8% and 0.3% respectively). Regionally, fossil emissions in 2022 are expected to decrease by 7.1% in the European Union (0.7 GtC, 2.6 GtCO$_2$), and by 3.4% in the United States (1.3 GtC, 4.9 GtCO$_2$), but to increase by 4.0% in China (3.2 GtC, 11.9 GtCO$_2$), 8.0% in India (0.8 GtC, 3.1 GtCO$_2$) and 0.9% for the rest of the world (4.2 GtC, 15.2 GtCO$_2$).

**Fossil CO$_2$ emissions decreased in 18 countries during the decade 2013-2022.** Altogether, these 18 countries contribute about 1.9 GtC yr$^{-1}$ (7.1 GtCO$_2$) fossil fuel CO$_2$ emissions over the last decade, representing about 20% of world CO$_2$ fossil emissions.

**Global CO$_2$ emissions from land-use, land-use change, and forestry (LUC) averaged 1.3 ± 0.7 GtC yr$^{-1}$ (4.7 ± 2.6 GtCO$_2$ yr$^{-1}$) for the 2013-2022 period with a preliminary projection for 2023 of 1.1 ± 0.7 GtC yr$^{-1}$ (4.0 ± 2.6 GtCO$_2$ yr$^{-1}$). A small decrease over the past two decades is not robust given the large model uncertainty.** Emissions from deforestation, the main driver of global gross sources, remain high at around 1.9 GtC yr$^{-1}$ over the 2013-2022 period, highlighting the strong potential of halting deforestation for emissions reductions. Sequestration of 1.3 GtC yr$^{-1}$ through re-/afforestation and forestry offsets two third of the deforestation emissions. Emissions from other land-use transitions and from peat drainage and peat fire add further, smaller contributions. The highest emitters during 2013-2022 in descending order were Brazil, Indonesia, and the Democratic Republic of the Congo, with these 3 countries contributing more than half of global land-use CO$_2$ emissions.

**The remaining carbon budget for a 50% likelihood to limit global warming to 1.5°C, 1.7°C and 2°C has respectively reduced to 75 GtC (275 GtCO$_2$), 175 GtC (625 GtCO$_2$) and 315 GtC (1150 GtCO$_2$) from the beginning of 2024, equivalent to around 7, 15 and 28 years, assuming 2023 emissions levels.** Total anthropogenic emissions were 11.1 GtC yr$^{-1}$ (40.7 GtCO$_2$ yr-1) in 2022, with a similar preliminary estimate of 11.2 GtC yr$^{-1}$ (40.9 GtCO2 yr$^{-1}$) for 2023.

**The concentration of CO$_2$ in the atmosphere is set to reach 419.2 ppm in 2023, 51% above pre-industrial levels.** The atmospheric CO$_2$ growth was 5.2 ± 0.02 GtC yr$^{-1}$ during the decade 2013-2022 (47% of total CO$_2$ emissions) with a preliminary 2023 growth rate estimate of around 4.0 GtC (1.89 ppm).

**The ocean CO$_2$ sink resumed a more rapid growth in the past two decades after low or no growth during the 1991-2002 period, overlaid with imprints of climate variability.** The estimates based on $f$CO$_2$-products and models diverge with the growth of the ocean CO$_2$ sink in the past decade being a factor 2.5 larger than in the



models. This discrepancy in the trend originates from all latitudes but is largest in the Southern Ocean. The
ocean $CO_2$ sink was $2.9 \pm 0.4$ GtC yr$^{-1}$ during the decade 2013-2022 (26% of total $CO_2$ emissions), and did not
grow since 2019 due to a triple La Niña event. A similar value of 2.9 GtC yr$^{-1}$ is preliminarily estimated for
2023, which marks an increase in the sink compared to the last two years due to the transition from La Niña to
El Niño conditions in 2023.
**The land $CO_2$ sink continued to increase during the 2013-2022 period primarily in response to increased**
**atmospheric $CO_2$, albeit with large interannual variability.** The land $CO_2$ sink was $3.3 \pm 0.8$ GtC yr$^{-1}$ during
the 2013-2022 decade (31% of total CO2 emissions), 0.4 GtC yr$^{-1}$ larger than during the previous decade (2000-
2009), with a preliminary 2023 estimate of around 3.0 GtC yr$^{-1}$. Year to year variability in the land sink is about
1 GtC yr$^{-1}$ and dominates the year-to-year changes in the global atmospheric $CO_2$ concentration, implying that
small annual changes in anthropogenic emissions (such as the fossil fuel emission decrease in 2020) are hard to
detect in the atmospheric $CO_2$ observations.




## 1   Introduction

The concentration of carbon dioxide ($CO_2$) in the atmosphere has increased from approximately 278 parts per
million (ppm) in 1750 (Gulev et al., 2021), the beginning of the Industrial Era, to $417.1 \pm 0.1$ ppm in 2022 (Lan
et al., 2023; Figure 1). The atmospheric $CO_2$ increase above pre-industrial levels was, initially, primarily caused
by the release of carbon to the atmosphere from deforestation and other land-use change activities (Canadell et
al., 2021). While emissions from fossil fuels started before the Industrial Era, they became the dominant source
of anthropogenic emissions to the atmosphere from around 1950 and their relative share has continued to
increase until present. Anthropogenic emissions occur on top of an active natural carbon cycle that circulates
carbon between the reservoirs of the atmosphere, ocean, and terrestrial biosphere on time scales from sub-daily
to millennia, while exchanges with geologic reservoirs occur at longer timescales (Archer et al., 2009).

The global carbon budget (GCB) presented here refers to the mean, variations, and trends in the perturbation of
$CO_2$ in the environment, referenced to the beginning of the Industrial Era (defined here as 1750). This paper
describes the components of the global carbon cycle over the historical period with a stronger focus on the
recent period (since 1958, onset of robust atmospheric $CO_2$ measurements), the last decade (2013-2022), the last
year (2022) and the current year (2023). Finally, it provides cumulative emissions from fossil fuels and land-use
change since the year 1750 (the pre-industrial period), and since the year 1850 (the reference year for historical
simulations in IPCC AR6) (Eyring et al., 2016).

We quantify the input of $CO_2$ to the atmosphere by emissions from human activities, the growth rate of
atmospheric $CO_2$ concentration, and the resulting changes in the storage of carbon in the land and ocean
reservoirs in response to increasing atmospheric $CO_2$ levels, climate change and variability, and other
anthropogenic and natural changes (Figure 2). An understanding of this perturbation budget over time and the
underlying variability and trends of the natural carbon cycle is necessary to understand the response of natural
sinks to changes in climate, $CO_2$ and land-use change drivers, and to quantify emissions compatible with a given
climate stabilisation target.

The components of the $CO_2$ budget that are reported annually in this paper include separate and independent
estimates for the $CO_2$ emissions from (1) fossil fuel combustion and oxidation from all energy and industrial
processes; also including cement production and carbonation ($E_{FOS}$; GtC yr$^{-1}$) and (2) the emissions resulting
from deliberate human activities on land, including those leading to land-use change ($E_{LUC}$; GtC yr$^{-1}$); and their
partitioning among (3) the growth rate of atmospheric $CO_2$ concentration ($G_{ATM}$; GtC yr$^{-1}$), and the uptake of
$CO_2$ (the '$CO_2$ sinks') in (4) the ocean ($S_{OCEAN}$; GtC yr$^{-1}$) and (5) on land ($S_{LAND}$; GtC yr$^{-1}$). The $CO_2$ sinks as
defined here conceptually include the response of the land (including inland waters and estuaries) and ocean
(including coastal and marginal seas) to elevated $CO_2$ and changes in climate and other environmental
conditions, although in practice not all processes are fully accounted for (see Section 2.10). Global emissions
and their partitioning among the atmosphere, ocean and land are in balance in the real world. Due to the
combination of imperfect spatial and/or temporal data coverage, errors in each estimate, and smaller terms not
included in our budget estimate (discussed in Section 2.10), the independent estimates (1) to (5) above do not



necessarily add up to zero. We therefore assess a set of additional lines of evidence derived from global
atmospheric inversion system results (Section 2.7), observed changes in oxygen concentration (Section 2.8) and
Earth System Models (ESMs) simulations (Section 2.9), all of which closing the global carbon balance. We also
estimate a budget imbalance ($B_{IM}$), which is a measure of the mismatch between the estimated emissions and the
estimated changes in the atmosphere, land and ocean, as follows:
$$B_{IM} = E_{FOS} + E_{LUC} - (G_{ATM} + S_{OCEAN} + S_{LAND}) \qquad\qquad (1)$$
$G_{ATM}$ is usually reported in ppm yr$^{-1}$, which we convert to units of carbon mass per year, GtC yr$^{-1}$, using 1 ppm
= 2.124 GtC (Ballantyne et al., 2012; Table 1). All quantities are presented in units of gigatonnes of carbon
(GtC, $10^{15}$ gC), which is the same as petagrams of carbon (PgC; Table 1). Units of gigatonnes of $CO_2$ (or billion
tonnes of $CO_2$) used in policy are equal to 3.664 multiplied by the value in units of GtC.
We also quantify $E_{FOS}$ and $E_{LUC}$ by country, including both territorial and consumption-based accounting for
$E_{FOS}$ (see Section 2), and discuss missing terms from sources other than the combustion of fossil fuels (see
Section 2.10, Supplement S1 and S2).
We now assess carbon dioxide removal (CDR) (see Sect. 2.2 and 2.3). Land-based CDR is significant, but
already accounted for in $E_{LUC}$ in equation (1) (Sect 3.2.2). Other CDR methods, not based on vegetation, are
currently several orders of magnitude smaller than the other components of the budget (Sect. 3.3), hence these
are not included in equation (1), or in the global carbon budget tables or figures (with the exception of Figure 2
where CDR is shown primarily for illustrative purpose).
The global $CO_2$ budget has been assessed by the Intergovernmental Panel on Climate Change (IPCC) in all
assessment reports (Prentice et al., 2001; Schimel et al., 1995; Watson et al., 1990; Denman et al., 2007; Ciais et
al., 2013; Canadell et al., 2021), and by others (e.g. Ballantyne et al., 2012). The Global Carbon Project (GCP,
www.globalcarbonproject.org, last access: 27 September 2023) has coordinated this cooperative community
effort for the annual publication of global carbon budgets for the year 2005 (Raupach et al., 2007; including
fossil emissions only), year 2006 (Canadell et al., 2007), year 2007 (GCP, 2008), year 2008 (Le Quéré et al.,
2009), year 2009 (Friedlingstein et al., 2010), year 2010 (Peters et al., 2012a), year 2012 (Le Quéré et al., 2013;
Peters et al., 2013), year 2013 (Le Quéré et al., 2014), year 2014 (Le Quéré et al., 2015a; Friedlingstein et al.,
2014), year 2015 (Jackson et al., 2016; Le Quéré et al., 2015b), year 2016 (Le Quéré et al., 2016), year 2017 (Le
Quéré et al., 2018a; Peters et al., 2017), year 2018 (Le Quéré et al., 2018b; Jackson et al., 2018), year 2019
(Friedlingstein et al., 2019; Jackson et al., 2019; Peters et al., 2020), year 2020 (Friedlingstein et al., 2020; Le
Quéré et al., 2021), year 2021 (Friedlingstein et al., 2022a; Jackson et al., 2022) and most recently the year 2022
(Friedlingstein et al., 2022b). Each of these papers updated previous estimates with the latest available
information for the entire time series.
We adopt a range of ±1 standard deviation (σ) to report the uncertainties in our global estimates, representing a
likelihood of 68% that the true value will be within the provided range if the errors have a gaussian distribution,
and no bias is assumed. This choice reflects the difficulty of characterising the uncertainty in the $CO_2$ fluxes
between the atmosphere and the ocean and land reservoirs individually, particularly on an annual basis, as well
as the difficulty of updating the $CO_2$ emissions from land-use change. A likelihood of 68% provides an





indication of our current capability to quantify each term and its uncertainty given the available information.
The uncertainties reported here combine statistical analysis of the underlying data, assessments of uncertainties
in the generation of the data sets, and expert judgement of the likelihood of results lying outside this range. The
limitations of current information are discussed in the paper and have been examined in detail elsewhere
(Ballantyne et al., 2015; Zscheischler et al., 2017). We also use a qualitative assessment of confidence level to
characterise the annual estimates from each term based on the type, amount, quality, and consistency of the
different lines of evidence as defined by the IPCC (Stocker et al., 2013).
This paper provides a detailed description of the data sets and methodology used to compute the global carbon
budget estimates for the industrial period, from 1750 to 2023, and in more detail for the period since 1959. This
paper is updated every year using the format of 'living data' to keep a record of budget versions and the changes
in new data, revision of data, and changes in methodology that lead to changes in estimates of the carbon
budget. Additional materials associated with the release of each new version will be posted at the Global Carbon
Project (GCP) website (http://www.globalcarbonproject.org/carbonbudget, last access: 27 September 2023),
with fossil fuel emissions also available through the Global Carbon Atlas (http://www.globalcarbonatlas.org,
last access: 27 September 2023). All underlying data used to produce the budget can also be found at
https://globalcarbonbudget.org/ (last access: 27 September 2023). With this approach, we aim to provide the
highest transparency and traceability in the reporting of $CO_2$, the key driver of climate change.
**2    Methods**
Multiple organisations and research groups around the world generated the original measurements and data used
to complete the global carbon budget. The effort presented here is thus mainly one of synthesis, where results
from individual groups are collated, analysed, and evaluated for consistency. We facilitate access to original
data with the understanding that primary data sets will be referenced in future work (see Table 2 for how to cite
the data sets, and Section on data availability). Descriptions of the measurements, models, and methodologies
follow below, with more detailed descriptions of each component provided as Supplementary Information (S1 to
S5).
This is the 18th version of the global carbon budget and the 12th revised version in the format of a living data
update in Earth System Science Data. It builds on the latest published global carbon budget of Friedlingstein et
al. (2022b). The main changes this year are: the inclusion of (1) data to year 2022 and a projection for the global
carbon budget for year 2023; (2) $CO_2$ uptake from Carbon Dioxide Removal (CDR); (3) land and ocean net
carbon fluxes estimates from changes in atmospheric oxygen concentration; (4) land and ocean net carbon
fluxes estimates from ESMs; and (5) revised method to estimate the current year (2023) atmospheric $CO_2$. The
main methodological differences between recent annual carbon budgets (2019 to 2023) are summarised in Table
3 and previous changes since 2006 are provided in Table S8.



### 2.1    Fossil $CO_2$ emissions ($E_{FOS}$)

#### 2.1.1    Historical period 1850-2022

The estimates of global and national fossil $CO_2$ emissions ($E_{FOS}$) include the oxidation of fossil fuels through both combustion (e.g., transport, heating) and chemical oxidation (e.g. carbon anode decomposition in aluminium refining) activities, and the decomposition of carbonates in industrial processes (e.g. the production of cement). We also include $CO_2$ uptake from the cement carbonation process. Several emissions sources are not estimated or not fully covered: coverage of emissions from lime production are not global, and decomposition of carbonates in glass and ceramic production are included only for the "Annex 1" countries of the United Nations Framework Convention on Climate Change (UNFCCC) for lack of activity data. These omissions are considered to be minor. Short-cycle carbon emissions - for example from combustion of biomass - are not included here but are accounted for in the $CO_2$ emissions from land use (see Section 2.2).

Our estimates of fossil $CO_2$ emissions rely on data collection by many other parties. Our goal is to produce the best estimate of this flux, and we therefore use a prioritisation framework to combine data from different sources that have used different methods, while being careful to avoid double counting and undercounting of emissions sources. The CDIAC-FF emissions dataset, derived largely from UN energy data, forms the foundation, and we extend emissions to year Y-1 using energy growth rates reported by the Energy Institute (a dataset formally produced by BP). We then proceed to replace estimates using data from what we consider to be superior sources, for example Annex 1 countries' official submissions to the UNFCCC. All data points are potentially subject to revision, not just the latest year. For full details see Andrew and Peters (2022).

Other estimates of global fossil $CO_2$ emissions exist, and these are compared by Andrew (2020a). The most common reason for differences in estimates of global fossil $CO_2$ emissions is a difference in which emissions sources are included in the datasets. Datasets such as those published by the energy company BP, the US Energy Information Administration, and the International Energy Agency's '$CO_2$ emissions from fuel combustion' are all generally limited to emissions from combustion of fossil fuels. In contrast, datasets such as PRIMAP-hist, CEDS, EDGAR, and GCP's dataset aim to include all sources of fossil $CO_2$ emissions. See Andrew (2020a) for detailed comparisons and discussion.

Cement absorbs $CO_2$ from the atmosphere over its lifetime, a process known as 'cement carbonation'. We estimate this $CO_2$ sink, from 1931, onwards as the average of two studies in the literature (Cao et al., 2020; Guo et al., 2021). Both studies use the same model, developed by Xi et al. (2016), with different parameterisations and input data, with the estimate of Guo and colleagues being a revision of Xi et al. (2016). The trends of the two studies are very similar. Since carbonation is a function of both current and previous cement production, we extend these estimates to 2022 by using the growth rate derived from the smoothed cement emissions (10-year smoothing) fitted to the carbonation data. In the present budget, we always include the cement carbonation carbon sink in the fossil $CO_2$ emission component ($E_{FOS}$).

We use the Kaya Identity for a simple decomposition of $CO_2$ emissions into the key drivers (Raupach et al., 2007). While there are variations (Peters et al., 2017), we focus here on a decomposition of $CO_2$ emissions into population, GDP per person, energy use per GDP, and $CO_2$ emissions per energy. Multiplying these individual





components together returns the $CO_2$ emissions. Using the decomposition, it is possible to attribute the change
in $CO_2$ emissions to the change in each of the drivers. This method gives a first-order understanding of what
causes $CO_2$ emissions to change each year.
**2.1.2    2023 projection**
We provide a projection of global fossil $CO_2$ emissions in 2022 by combining separate projections for China,
USA, EU, India, and for all other countries combined. The methods are different for each of these. For China we
combine monthly fossil fuel production data from the National Bureau of Statistics and trade data from the
Customs Administration, giving us partial data for the growth rates to date of natural gas, petroleum, and
cement, and of the apparent consumption itself for raw coal. We then use a regression model to project full-year
emissions based on historical observations. For the USA our projection is taken directly from the Energy
Information Administration's (EIA) Short-Term Energy Outlook (EIA, 2023), combined with the year-to-date
growth rate of cement clinker production. For the EU we use monthly energy data from Eurostat to derive
estimates of monthly $CO_2$ emissions through July, with coal emissions extended through September using a
statistical relationship with reported electricity generation from coal and other factors. For natural gas we use
Holt-Winters to project the last four months of the year. EU emissions from oil are derived using the EIA's
projection of oil consumption for Europe. EU cement emissions are based on available year-to-date data from
three of the largest producers, Germany, Poland, and Spain. India's projected emissions are derived from
estimates through August (July for coal) using the methods of Andrew (2020b) and extrapolated assuming
seasonal patterns from before 2019. Emissions for the rest of the world are derived using projected growth in
economic production from the IMF (2023) combined with extrapolated changes in emissions intensity of
economic production. More details on the $E_{FOS}$ methodology and its 2023 projection can be found in
Supplement S.1.
**2.2    $CO_2$ emissions from land-use, land-use change and forestry ($E_{LUC}$)**
**2.2.1    Historical period 1850-2022**
The net $CO_2$ flux from land-use, land-use change and forestry ($E_{LUC}$, called land-use change emissions in the
rest of the text) includes $CO_2$ fluxes from deforestation, afforestation, logging and forest degradation (including
harvest activity), shifting cultivation (cycle of cutting forest for agriculture, then abandoning), and regrowth of
forests (following wood harvest or agriculture abandonment). Emissions from peat burning and peat drainage
are added from external datasets, peat drainage being averaged from three spatially explicit independent datasets
(see Supplement S.2.1).
Three bookkeeping approaches (updated estimates each of BLUE (Hansis et al., 2015), OSCAR (Gasser et al.,
2020), and H&C2023 (Houghton and Castanho, 2023)) were used to quantify gross emissions and gross
removals and the resulting net $E_{LUC}$. Uncertainty estimates were derived from the Dynamic Global Vegetation
Models (DGVMs) ensemble for the time period prior to 1960, and using for the recent decades an uncertainty
range of $\pm 0.7$ GtC yr$^{-1}$, which is a semi-quantitative measure for annual and decadal emissions and reflects our



best value judgement that there is at least 68% chance (±1σ) that the true land-use change emission lies within
the given range, for the range of processes considered here.
Our $E_{LUC}$ estimates follow the definition of global carbon cycle models of $CO_2$ fluxes related to land use and
land management and differ from IPCC definitions adopted in National GHG Inventories (NGHGI) for
reporting under the UNFCCC, which additionally generally include, through adoption of the IPCC so-called
managed land proxy approach, the terrestrial fluxes occurring on all land that countries define as managed. This
partly includes fluxes due to environmental change (e.g. atmospheric $CO_2$ increase), which are part of $S_{LAND}$ in
our definition. This causes the global emission estimates to be smaller for NGHGI than for the global carbon
budget definition (Grassi et al., 2018). The same is the case for the Food Agriculture Organization (FAO)
estimates of carbon fluxes on forest land, which include both anthropogenic and natural sources on managed
land (Tubiello et al., 2021). We translate the two definitions to each other, to provide a comparison of the
anthropogenic carbon budget to the official country reporting to the climate convention.
$E_{LUC}$ contains a range of fluxes that are related to Carbon Dioxide Removal (CDR). CDR can be defined as the
set of anthropogenic activities that remove $CO_2$ from the atmosphere and store it in durable form, such as in
forest biomass and soils, long-lived products, or in geological or ocean reservoirs. We quantify vegetation-based
CDR that is implicitly or explicitly captured by land-use fluxes consistent with our updated model estimates
(CDR not based on vegetation is discussed in Section 2.3; IPCC, 2023). We quantify re/afforestation from the
three bookkeeping estimates by separating forest regrowth in shifting cultivation cycles from permanent
increases in forest cover (see Supplement C.2.1). The latter count as CDR, but it should be noted that the
permanence of the storage under climate risks such as fire is increasingly questioned. Other CDR activities
contained in $E_{LUC}$ include the transfer of carbon to harvested wood products (HWP), which is represented by the
bookkeeping models with varying details concerning product usage and their lifetimes; bioenergy with carbon
capture and storage (BECCS); and biochar production. Bookkeeping and TRENDY models currently only
represent BECCS and biochar with regard to the $CO_2$ removal through photosynthesis, but not for the durable
storage. HWP, BECCS, and biochar are typically counted as CDR when the transfer to the durable storage site
occurs and not when the $CO_2$ is removed from the atmosphere, which complicates a direct comparison to the
global carbon budgets approach to quantify annual fluxes to and from the atmosphere. Estimates for CDR
through HWP, BECCS, and biochar are thus not indicated in this budget, but can be found elsewhere (see
Section 3.2.2).

### 2.2.2   2023 Projection

We project the 2023 land-use emissions for BLUE, H&C2023, and OSCAR based on their $E_{LUC}$ estimates for
2022 and adding the change in carbon emissions from peat fires and tropical deforestation and degradation fires
(2023 emissions relative to 2022 emissions) estimated using active fire data (MCD14ML; Giglio et al., 2016).
Peat drainage is assumed to be unaltered as it has low interannual variability. More details on the $E_{LUC}$
methodology can be found in Supplement S.2.





### 2.3 Carbon Dioxide Removal (CDR) not based on vegetation

CDR not based on terrestrial vegetation currently relies on enhanced rock weathering and Direct Air Carbon Capture and Storage (DACCS) projects. The majority of this (58%) derives from a single project: Climeworks' Orca DACCS plant based in Hellisheidi, Iceland. The remainder is generated by 13 small-scale projects including, for example, 500 tons of carbon dioxide sequestered through the spreading of crushed olivine on agricultural areas by Eion Carbon. We use data from the State of CDR Report (Smith et al., 2023), which quantifies all currently deployed CDR methods, including the land-use related activities already covered by Section 2.2. The State of CDR Report (Smith et al., 2023) combines estimates of carbon storage in managed land derived from NGHGI data with project-by-project storage rates obtained through 20 extant CDR databases and registries (status as of mid-year 2022) by Powis et al. (2023). They assessed the data quality on existing CDR projects to be poor, suffering from fragmentation, different reporting standards, limited geographical coverage, and inclusion of a number of pilot plants with uncertain lifespans. As a consequence, these numbers could change substantially from year-to-year in the near-term.

### 2.4 Growth rate in atmospheric $CO_2$ concentration ($G_{ATM}$)

#### 2.4.1 Historical period 1850-2022

The rate of growth of the atmospheric $CO_2$ concentration is provided for years 1959-2022 by the US National Oceanic and Atmospheric Administration Global Monitoring Laboratory (NOAA/GML; Lan et al., 2023), which is updated from Ballantyne et al. (2012) and includes recent revisions to the calibration scale of atmospheric $CO_2$ measurements (Hall et al., 2021). For the 1959-1979 period, the global growth rate is based on measurements of atmospheric $CO_2$ concentration averaged from the Mauna Loa and South Pole stations, as observed by the $CO_2$ Program at Scripps Institution of Oceanography (Keeling et al., 1976). For the 1980-2021 time period, the global growth rate is based on the average of multiple stations selected from the marine boundary layer sites with well-mixed background air (Ballantyne et al., 2012), after fitting a smooth curve through the data for each station as a function of time, and averaging by latitude band (Masarie and Tans, 1995). The annual growth rate is estimated by Lan et al. (2023) from atmospheric $CO_2$ concentration by taking the average of the most recent December-January months corrected for the average seasonal cycle and subtracting this same average one year earlier. The growth rate in units of ppm yr$^{-1}$ is converted to units of GtC yr$^{-1}$ by multiplying by a factor of 2.124 GtC per ppm, assuming instantaneous mixing of $CO_2$ throughout the atmosphere (Ballantyne et al., 2012; Table 1).

Since 2020, NOAA/GML provides estimates of atmospheric $CO_2$ concentrations with respect to a new calibration scale, referred to as WMO-CO2-X2019, in line with a recalibration agreed by the World Meteorological Organization (WMO) Global Atmosphere Watch (GAW) community (Hall et al., 2021). The re-calibrated data were first used to estimate $G_{ATM}$ in the 2021 edition of the global carbon budget (Friedlingstein et al., 2022a). Friedlingstein et al. (2022a) verified that the change of scales from WMO-CO2-X2007 to WMO-CO2-X2019 made a negligible difference to the value of $G_{ATM}$ (-0.06 GtC yr$^{-1}$ during 2010-2019 and -0.01 GtC yr$^{-1}$ during 1959-2019, well within the uncertainty range reported below).





The uncertainty around the atmospheric growth rate is due to four main factors. First, the long-term
reproducibility of reference gas standards (around 0.03 ppm for $1\sigma$ from the 1980s; Lan et al., 2023). Second,
small unexplained systematic analytical errors that may have a duration of several months to two years come
and go. They have been simulated by randomising both the duration and the magnitude (determined from the
existing evidence) in a Monte Carlo procedure. Third, the network composition of the marine boundary layer
with some sites coming or going, gaps in the time series at each site, etc (Lan et al., 2023). The latter uncertainty
was estimated by NOAA/GML with a Monte Carlo method by constructing 100 "alternative" networks (Masarie
and Tans, 1995; NOAA/GML, 2019). The second and third uncertainties, summed in quadrature, add up to
0.085 ppm on average (Lan et al., 2023). Fourth, the uncertainty associated with using the average $CO_2$
concentration from a surface network to approximate the true atmospheric average $CO_2$ concentration (mass-
weighted, in 3 dimensions) as needed to assess the total atmospheric $CO_2$ burden. In reality, $CO_2$ variations
measured at the stations will not exactly track changes in total atmospheric burden, with offsets in magnitude
and phasing due to vertical and horizontal mixing. This effect must be very small on decadal and longer time
scales, when the atmosphere can be considered well mixed. The CO2 increase in the stratosphere lags the
increase (meaning lower concentrations) that we observe in the marine boundary layer, while the continental
boundary layer (where most of the emissions take place) leads the marine boundary layer with higher
concentrations. These effects nearly cancel each other. In addition the growth rate is nearly the same everywhere
(Ballantyne et al, 2012). We therefore maintain an uncertainty around the annual growth rate based on the
multiple stations data set ranges between 0.11 and 0.72 GtC yr$^{-1}$, with a mean of 0.61 GtC yr$^{-1}$ for 1959-1979
and 0.17 GtC yr$^{-1}$ for 1980-2022, when a larger set of stations were available as provided by Lan et al. (2023).
We estimate the uncertainty of the decadal averaged growth rate after 1980 at 0.02 GtC yr$^{-1}$ based on the
calibration and the annual growth rate uncertainty but stretched over a 10-year interval. For years prior to 1980,
we estimate the decadal averaged uncertainty to be 0.07 GtC yr$^{-1}$ based on a factor proportional to the annual
uncertainty prior and after 1980 (0.02 * [0.61/0.17] GtC yr$^{-1}$).
We assign a high confidence to the annual estimates of $G_{ATM}$ because they are based on direct measurements
from multiple and consistent instruments and stations distributed around the world (Ballantyne et al., 2012; Hall
et al., 2021).
To estimate the total carbon accumulated in the atmosphere since 1750 or 1850, we use an atmospheric $CO_2$
concentration of 278.3 ± 3 ppm or 285.1 ± 3 ppm, respectively (Gulev et al., 2021). For the construction of the
cumulative budget shown in Figure 3, we use the fitted estimates of $CO_2$ concentration from Joos and Spahni
(2008) to estimate the annual atmospheric growth rate using the conversion factors shown in Table 1. The
uncertainty of ±3 ppm (converted to ±1$\sigma$) is taken directly from the IPCC's AR5 assessment (Ciais et al., 2013).
Typical uncertainties in the growth rate in atmospheric $CO_2$ concentration from ice core data are equivalent to
±0.1-0.15 GtC yr$^{-1}$ as evaluated from the Law Dome data (Etheridge et al., 1996) for individual 20-year intervals
over the period from 1850 to 1960 (Bruno and Joos, 1997).
**2.4.2    2023 projection**
We provide an assessment of $G_{ATM}$ for 2023 as the average of two methods. As in previous GCB releases, we
use the observed monthly global atmospheric $CO_2$ concentration (GLO) through June 2023 (Lan et al., 2023),





and the bias-adjusted Holt–Winters exponential smoothing with additive seasonality (Chatfield, 1978) to project
to January 2024. The uncertainty is estimated from past variability using the standard deviation of the last 5
years' monthly growth rates. For the first time this year, we also use the multi-model mean and uncertainty of
the 2023 $G_{ATM}$ estimated by the ESMs prediction system (see Section 2.9). We then take the average of the
Holt–Winters and ESMs $G_{ATM}$ estimates, with their respective uncertainty combined quadratically.
Similarly, the projection of the 2023 global average $CO_2$ concentration (in ppm), is calculated as the average of
the estimates from the two methods. For Holt–Winters method, it is the annual average of global concentration
over the 12 months; for the ESMs, it is the observed global average $CO_2$ concentration for 2022 plus the annual
increase in 2023 predicted by the ESMs multi-model mean.
**2.5    Ocean $CO_2$ sink**
**2.5.1    Historical period 1850-2022**
The reported estimate of the global ocean anthropogenic $CO_2$ sink $S_{OCEAN}$ is derived as the average of two
estimates. The first estimate is derived as the mean over an ensemble of ten global ocean biogeochemistry
models (GOBMs, Table 4 and Table S2). The second estimate is obtained as the mean over an ensemble of
seven surface ocean $fCO_2$-observation-based data-products (Table 4 and Table S3). An eighth $fCO_2$-product
(Watson et al., 2020) is shown, but is not included in the ensemble average as it differs from the other products
by adjusting the flux to a cool, salty ocean surface skin (see Supplement S.3.1 for a discussion of the Watson
product). The GOBMs simulate both the natural and anthropogenic $CO_2$ cycles in the ocean. They constrain the
anthropogenic air-sea $CO_2$ flux (the dominant component of $S_{OCEAN}$) by the transport of carbon into the ocean
interior, which is also the controlling factor of present-day ocean carbon uptake in the real world. They cover
the full globe and all seasons and were recently evaluated against surface ocean carbon observations, suggesting
they are suitable to estimate the annual ocean carbon sink (Hauck et al., 2020). The $fCO_2$-products are tightly
linked to observations of $fCO_2$ (fugacity of $CO_2$, which equals $pCO_2$ corrected for the non-ideal behaviour of the
gas; Pfeil et al., 2013), which carry imprints of temporal and spatial variability, but are also sensitive to
uncertainties in gas-exchange parameterizations and data-sparsity (Gloege et al., 2021, Hauck et al., 2023).
Their asset is the assessment of the mean spatial pattern of variability and its seasonality (Hauck et al., 2020,
Gloege et al. 2021, Hauck et al., 2023). We further use two diagnostic ocean models to estimate $S_{OCEAN}$ over the
industrial era (1781-1958).
The global $fCO_2$-based flux estimates were adjusted to remove the pre-industrial ocean source of $CO_2$ to the
atmosphere of $0.65 \pm 0.3$ GtC yr$^{-1}$ from river input to the ocean (Regnier et al., 2022), to satisfy our definition of
$S_{OCEAN}$ (Hauck et al., 2020). The river flux adjustment was distributed over the latitudinal bands using the
regional distribution of Lacroix et al. (2020; North: 0.14 GtC yr$^{-1}$, Tropics: 0.42 GtC yr$^{-1}$, South: 0.09 GtC yr$^{-1}$).
Acknowledging that this distribution is based on only one model, the advantage is that a gridded field is
available and the river flux adjustment can be calculated for the three latitudinal bands and the RECCAP regions
(REgional Carbon Cycle Assessment and Processes (RECCAP2; Ciais et al., 2020). This data set suggests that
more of the riverine outgassing is located in the tropics than in the Southern Ocean, and is thus opposed to the
previously used data set of Aumont et al. (2001). Accordingly, the regional distribution is associated with a



major uncertainty in addition to the large uncertainty around the global estimate (Crisp et al., 2022; Gruber et
al., 2023). Anthropogenic perturbations of river carbon and nutrient transport to the ocean are not considered
(see Section 2.10 and Supplement S.6.3).
We derive $S_{OCEAN}$ from GOBMs by using a simulation (sim A) with historical forcing of climate and
atmospheric $CO_2$, accounting for model biases and drift from a control simulation (sim B) with constant
atmospheric $CO_2$ and normal year climate forcing. A third simulation (sim C) with historical atmospheric $CO_2$
increase and normal year climate forcing is used to attribute the ocean sink to $CO_2$ (sim C minus sim B) and
climate (sim A minus sim C) effects. A fourth simulation (sim D; historical climate forcing and constant
atmospheric $CO_2$) is used to compare the change in anthropogenic carbon inventory in the interior ocean (sim A
minus sim D) to the observational estimate of Gruber et al. (2019) with the same flux components (steady state
and non-steady state anthropogenic carbon flux). The $fCO_2$-products are adjusted with respect to their original
publications to represent the full ice-free ocean area, including coastal zones and marginal seas, when the area
coverage is below 99%. This is done by either area filling following Fay et al. (2021) or a simple scaling
approach. GOBMs and $fCO_2$-products fall within the observational constraints over the 1990s ($2.2 \pm 0.7$ GtC yr$^{-1}$
$^{1}$, Ciais et al., 2013) after applying adjustments.
$S_{OCEAN}$ is calculated as the average of the GOBM ensemble mean and the $fCO_2$-product ensemble mean from
1990 onwards. Prior to 1990, it is calculated as the GOBM ensemble mean plus half of the offset between
GOBMs and $fCO_2$-products ensemble means over 1990-2001.
We assign an uncertainty of $\pm 0.4$ GtC yr$^{-1}$ to the ocean sink based on a combination of random (ensemble
standard deviation) and systematic uncertainties (GOBMs bias in anthropogenic carbon accumulation,
previously reported uncertainties in $fCO_2$-products; see Supplement S.3.4). We assess a medium confidence
level to the annual ocean $CO_2$ sink and its uncertainty because it is based on multiple lines of evidence, it is
consistent with ocean interior carbon estimates (Gruber et al., 2019, see Section 3.6.5) and the interannual
variability in the GOBMs and data-based estimates is largely consistent and can be explained by climate
variability. We refrain from assigning a high confidence because of the systematic deviation between the
GOBM and $fCO_2$-product trends since around 2002. More details on the $S_{OCEAN}$ methodology can be found in
Supplement S.3.

### 2.5.2    2023 Projection

The ocean $CO_2$ sink forecast for the year 2023 is based on the annual historical (Lan et al., 2023) and our
estimated 2023 atmospheric $CO_2$ concentration growth rate, the historical and our estimated 2023 annual global
fossil fuel emissions from this year's carbon budget, and the spring (March, April, May) Oceanic Niño Index
(ONI) (NCEP, 2023). Using a non-linear regression approach, i.e., a feed-forward neural network, atmospheric
$CO_2$, ONI, and the fossil fuel emissions are used as training data to best match the annual ocean $CO_2$ sink (i.e.
combined $S_{OCEAN}$ estimate from GOBMs and data products) from 1959 through 2022 from this year's carbon
budget. Using this relationship, the 2023 $S_{OCEAN}$ can then be estimated from the projected 2022 input data using
the non-linear relationship established during the network training. To avoid overfitting, the neural network was
trained with a variable number of hidden neurons (varying between 2-5) and 20% of the randomly selected



training data were withheld for independent internal testing. Based on the best output performance (tested using
the 20% withheld input data), the best performing number of neurons was selected. In a second step, we trained
the network 10 times using the best number of neurons identified in step 1 and different sets of randomly
selected training data. The mean of the 10 trainings is considered our best forecast, whereas the standard
deviation of the 10 ensembles provides a first order estimate of the forecast uncertainty. This uncertainty is then
combined with the $S_{OCEAN}$ uncertainty (0.4 GtC yr$^{-1}$) to estimate the overall uncertainty of the 2023 projection.
As an additional line of evidence, we also assess the 2023 atmosphere-ocean carbon flux from the ESM
prediction system (see Section 2.9).

**2.6    Land CO$_2$ sink**

**2.6.1    Historical Period 1850-2022**

The terrestrial land sink ($S_{LAND}$) is thought to be due to the combined effects of fertilisation by rising
atmospheric CO$_2$ and N inputs on plant growth, as well as the effects of climate change such as the lengthening
of the growing season in northern temperate and boreal areas. $S_{LAND}$ does not include land sinks directly
resulting from land-use and land-use change (e.g., regrowth of vegetation) as these are part of the land-use flux
($E_{LUC}$), although system boundaries make it difficult to attribute exactly CO$_2$ fluxes on land between $S_{LAND}$ and
$E_{LUC}$ (Erb et al., 2013).
$S_{LAND}$ is estimated from the multi-model mean of 20 DGVMs (Table S1) with an additional comparison of
DGVMs with a data-driven, carbon data model framework (CARDAMOM) (Bloom and Williams, 2015; Bloom
et al., 2016), see Supplement S4. DGVMs simulations include all climate variability and CO$_2$ effects over land.
In addition to the carbon cycle represented in all DGVMs, 14 models also account for the nitrogen cycle and
hence can include the effect of N inputs on $S_{LAND}$. The DGVMs estimate of $S_{LAND}$ does not include the export of
carbon to aquatic systems or its historical perturbation, which is discussed in Supplement S.6.3. See Supplement
S.4.2 for DGVMs evaluation and uncertainty assessment for $S_{LAND}$, using the International Land Model
Benchmarking system (ILAMB; Collier et al., 2018). More details on the $S_{LAND}$ methodology can be found in
Supplement S.4.

**2.6.2    2023 Projection**

Like for the ocean forecast, the land CO$_2$ sink ($S_{LAND}$) forecast is based on the annual historical (Lan et al.,
2023) and our estimated 2023 atmospheric CO$_2$ concentration , historical and our estimated 2023 annual global
fossil fuel emissions from this year's carbon budget, and the summer (June, July, August) ONI (NCEP, 2022).
All training data are again used to best match $S_{LAND}$ from 1959 through 2022 from this year's carbon budget
using a feed-forward neural network. To avoid overfitting, the neural network was trained with a variable
number of hidden neurons (varying between 2-15), larger than for $S_{OCEAN}$ prediction due to the stronger land
carbon interannual variability. As done for $S_{OCEAN}$, a pre-training selects the optimal number of hidden neurons
based on 20% withheld input data, and in a second step, an ensemble of 10 forecasts is produced to provide the
mean forecast plus uncertainty. This uncertainty is then combined with the $S_{LAND}$ uncertainty for 2022 (0.9 GtC
yr$^{-1}$) to estimate the overall uncertainty of the 2023 projection.





**2.7    Atmospheric inversion estimate**
The world-wide network of in-situ atmospheric measurements and satellite derived atmospheric $CO_2$ column
($xCO_2$) observations put a strong constraint on changes in the atmospheric abundance of $CO_2$. This is true
globally (hence our large confidence in $G_{ATM}$), but also regionally in regions with sufficient observational
density found mostly in the extra-tropics. This allows atmospheric inversion methods to constrain the magnitude
and location of the combined total surface $CO_2$ fluxes from all sources, including fossil and land-use change
emissions and land and ocean $CO_2$ fluxes. The inversions assume $E_{FOS}$ to be well known, and they solve for the
spatial and temporal distribution of land and ocean fluxes from the residual gradients of $CO_2$ between stations
that are not explained by fossil fuel emissions. By design, such systems thus close the carbon balance ($B_{IM} = 0$)
and thus provide an additional perspective on the independent estimates of the ocean and land fluxes.
This year's release includes fourteen inversion systems that are described in Table S4, of which thirteen are
included in the ensemble of inverse estimates presented in the text and figures. Each system is rooted in
Bayesian inversion principles but uses different methodologies. These differences concern the selection of
atmospheric $CO_2$ data or $xCO_2$, and the choice of a-priori fluxes to refine. They also differ in spatial and
temporal resolution, assumed correlation structures, and mathematical approach of the models (see references in
Table S4 for details). Importantly, the systems use a variety of transport models, which was demonstrated to be
a driving factor behind differences in atmospheric inversion-based flux estimates, and specifically their
distribution across latitudinal bands (Gaubert et al., 2019; Schuh et al., 2019). Six inversion systems (CAMS-
FT23r1, CMS-flux, GONGGA, THU, COLA, GCASv2) used satellite $xCO_2$ retrievals from GOSAT and/or
OCO-2, scaled to the WMO 2019 calibration scale. Two inversions this year (CMS-Flux, COLA) used these
xCO2 datasets in addition to the in-situ observational $CO_2$ mole fraction records.
The original products delivered by the inverse modellers were modified to facilitate the comparison to the other
elements of the budget, specifically on two accounts: (1) global total fossil fuel emissions including cement
carbonation $CO_2$ uptake, and (2) riverine $CO_2$ transport. Details are given below. We note that with these
adjustments the inverse results no longer represent the net atmosphere-surface exchange over land/ocean areas
as sensed by atmospheric observations. Instead, for land, they become the net uptake of $CO_2$ by vegetation and
soils that is not exported by fluvial systems, similar to the DGVMs estimates. For oceans, they become the net
uptake of anthropogenic $CO_2$, similar to the GOBMs estimates.
The inversion systems prescribe global fossil fuel emissions based on e.g. the GCP's Gridded Fossil Emissions
Dataset versions 2023.1 (GCP-GridFED; Jones et al., 2023), which are updates to GCP-GridFEDv2021
presented by Jones et al. (2021b). GCP-GridFEDv2023 scales gridded estimates of $CO_2$ emissions from
EDGARv4.3.2 (Janssens-Maenhout et al., 2019) within national territories to match national emissions
estimates provided by the GCB for the years 1959-2022, which were compiled following the methodology
described in Section 2.1. Small differences between the systems due to for instance regridding to the transport
model resolution, or use of different fossil fuel emissions, are adjusted in the latitudinal partitioning we present,
to ensure agreement with the estimate of $E_{FOS}$ in this budget. We also note that the ocean fluxes used as prior by
8 out of 14 inversions are part of the suite of the ocean process model or $fCO_2$-products listed in Section 2.5.



Although these fluxes are further adjusted by the atmospheric inversions, it makes the inversion estimates of the
ocean fluxes not completely independent of $S_{OCEAN}$ assessed here.
To facilitate comparisons to the independent $S_{OCEAN}$ and $S_{LAND}$, we used the same corrections for transport and
outgassing of carbon transported from land to ocean, as done for the observation-based estimates of $S_{OCEAN}$ (see
Supplement S.3).
The atmospheric inversions are evaluated using vertical profiles of atmospheric $CO_2$ concentrations (Figure S4).
More than 30 aircraft programs over the globe, either regular programs or repeated surveys over at least 9
months (except for SH programs), have been used to assess system performance (with space-time observational
coverage sparse in the SH and tropics, and denser in NH mid-latitudes; Table S7). The fourteen systems are
compared to the independent aircraft $CO_2$ measurements between 2 and 7 km above sea level between 2001 and
2022. Results are shown in Figure S4 and discussed in Supplement S.5.2. One inversion was flagged for
concerns after quality control with these observations, as well as assessment of their global growth rate. This
makes the number of systems included in the ensemble to be N=13.
With a relatively small ensemble of systems that cover at least one full decade (N=9), and which moreover share
some a-priori fluxes used with one another, or with the process-based models, it is difficult to justify using their
mean and standard deviation as a metric for uncertainty across the ensemble. We therefore report their full range
(min-max) without their mean. More details on the atmospheric inversions methodology can be found in
Supplement S.5.
**2.8        Atmospheric oxygen based estimate**
Long-term atmospheric $O_2$ and $CO_2$ observations allow estimation of the global ocean and land carbon sinks,
due to the coupling of $O_2$ and $CO_2$ with distinct exchange ratios for fossil fuel emissions and land uptake, and
uncoupled $O_2$ and $CO_2$ ocean exchange (Keeling and Manning, 2014). The global ocean and net land carbon
sinks were calculated following methods and constants used in Keeling and Manning (2014), but modified to
also include the effective $O_2$ source from metal refining (Battle et al., 2023), and using a value of 1.05 for the
exchange ratio of the net land sink, following Resplandy et al. (2019). Atmospheric $O_2$ is observed as $\delta(O_2/N_2)$
and combined with $CO_2$ mole fraction observations into Atmospheric Potential Oxygen (APO, Stephens et al.,
1998). The APO observations from 1990 to 2022 were taken from a weighted average of flask records from the
three stations in the Scripps $O_2$ program network (Alert, Canada (ALT), La Jolla, California (LJO), and Cape
Grim, Australia (CGO), weighted per Keeling and Manning (2020). Observed $CO_2$ was taken from the globally
averaged marine surface annual mean growth rate from the NOAA/ESRL Global Greenhouse Gas Reference
Network (Lan et al., 2023). The $O_2$ source from ocean warming is based on ocean heat content from updated
data from NOAA/NCEI (Levitus et al., 2012). The effective $O_2$ source from metal refining is based on
production data from Bray (2020), Flanagan (2021), and Tuck (2022). Uncertainty was determined through a
Monte Carlo approach with 5,000 iterations, using uncertainties prescribed in Keeling and Manning (2014),
including observational uncertainties from Keeling et al. (2007) and autoregressive errors in fossil fuel
emissions (Ballantyne et al., 2015). The reported uncertainty is one standard deviation of the ensemble.



## 2.9  Earth System Models estimate

Reconstructions and predictions from decadal prediction systems based on Earth system models (ESMs) provide a novel line of evidence in assessing the atmosphere-land and atmosphere-ocean carbon fluxes in the past decades and predicting their changes for the current year. The decadal prediction systems based on ESMs used here consist of three sets of simulations: (i) uninitialized freely evolving historical simulations (1850-2014); (ii) assimilation reconstruction incorporating observational data into the model (1980-2022); (iii) initialized prediction simulations for the 1981-2023 period, starting every year from initial states obtained from the above assimilation simulations. The assimilations are designed to reconstruct the actual evolution of the Earth system by assimilating essential fields from data products. The assimilations' states, which are expected to be close to observations, are used to start the initialized prediction simulations used for the current year (2023) global carbon budget. Similar initialized prediction simulations starting every year (Nov. 1st or Jan. 1st) over the 1981-2022 period (i.e., hindcasts) are also performed for predictive skill quantification and for bias correction. More details on the illustration of a decadal prediction system based on an ESM can refer to Figure 1 of Li et al. (2023).

By assimilating physical atmospheric and oceanic data products into the ESMs, the models are able to reproduce the historical variations of the atmosphere-sea $CO_2$ fluxes, atmosphere-land $CO_2$ fluxes, and atmospheric $CO_2$ growth rate (Li et al., 2016, 2019; Lovenduski et al., 2019a,b; Ilyina et al., 2021; Li et al., 2023). Furthermore, the ESM-based predictions have proven their skill in predicting the air-sea $CO_2$ fluxes for up to 6 years, the air-land $CO_2$ fluxes and atmospheric $CO_2$ growth for 2 years (Lovenduski et al., 2019a,b; Ilyina et al., 2021; Li et al., 2023). The reconstructions from the fully coupled model simulations ensure a closed budget within the Earth system, i.e., no budget imbalance term.

Four ESMs, i.e., CanESM5 (Swart et al., 2019; Sospedra-Alfonso et al., 2021), IPSL-CM6A-CO2-LR (Boucher et al., 2020), MIROC-ES2L (Watanabe et al., 2020), and MPI-ESM1-2-LR (Mauritsen et al., 2019; Li et al., 2023), have performed the set of prediction simulations. Each ESM uses a different assimilation method and combination of data products incorporated in the system, more details on the models configuration can be found in Table 4. The ESMs use external forcings from the Coupled Model Intercomparison Project Phase 6 (CMIP6) historical (1980-2014) plus SSP2-4.5 baseline and CovidMIP two year blip scenario (2015-2023) (Eyring et al., 2016; Jones et al., 2021a). The $CO_2$ emissions forcing from 2015-2023 are substituted by GCB-GridFED (v2023.1, Jones et al., 2023) to provide a more realistic forcing. Reconstructions of atmosphere-ocean $CO_2$ fluxes ($S_{OCEAN}$) and atmosphere-land $CO_2$ fluxes ($S_{LAND}$-$E_{LUC}$) for the time period from 1980-2022 are assessed here. Predictions of the atmosphere-ocean $CO_2$ flux, atmosphere-land $CO_2$ flux, and atmospheric $CO_2$ growth for 2023 are calculated based on the predictions at a lead time of 1 year. The predictions are bias-corrected using the 1985-2014 climatology mean of GCB2022 (Friedlingstein et al., 2022), more details on methods can be found in Boer et al. (2016) and Li et al. (2023). The ensemble size of initialized prediction simulations is 10, and the ensemble mean for each individual model is used here. The ESMs are used here to support the assessment of $S_{OCEAN}$ and net atmosphere-land $CO_2$ flux ($S_{LAND}$ - $E_{LUC}$) over the 1980-2022 period, and to provide an estimate of the 2023 projection of $G_{ATM}$.



### 2.10 Processes not included in the global carbon budget

The contribution of anthropogenic CO and $CH_4$ to the global carbon budget is not fully accounted for in Eq. (1) and is described in Supplement S.6.1. The contributions to $CO_2$ emissions of decomposition of carbonates not accounted for is described in Supplement S.6.2. The contribution of anthropogenic changes in river fluxes is conceptually included in Eq. (1) in $S_{OCEAN}$ and in $S_{LAND}$, but it is not represented in the process models used to quantify these fluxes. This effect is discussed in Supplement S.6.3. Similarly, the loss of additional sink capacity from reduced forest cover is missing in the combination of approaches used here to estimate both land fluxes ($E_{LUC}$ and $S_{LAND}$) and its potential effect is discussed and quantified in Supplement S.6.4.

## 3 Results

For each component of the global carbon budget, we present results for three different time periods: the full historical period, from 1850 to 2022, the decades in which we have atmospheric concentration records from Mauna Loa (1960-2022), a specific focus on last year (2022), and the projection for the current year (2023). Subsequently, we assess the estimates of the budget components of the last decades against the top-down constraints from inverse modelling of atmospheric observations, the land/ocean partitioning derived from the atmospheric $O_2$ measurements, and the budget components estimates from the ESMs assimilation simulations. Atmospheric inversions further allow for an assessment of the budget components with a regional breakdown of land and ocean sinks.

### 3.1 Fossil $CO_2$ Emissions

#### 3.1.1 Historical period 1850-2022

Cumulative fossil $CO_2$ emissions for 1850-2022 were $477 \pm 25$ GtC, including the cement carbonation sink (Figure 3, Table 8, with all cumulative numbers rounded to the nearest 5GtC). In this period, 46% of global fossil $CO_2$ emissions came from coal, 35% from oil, 15% from natural gas, 3% from decomposition of carbonates, and 1% from flaring. In 1850, the UK stood for 62% of global fossil $CO_2$ emissions. In 1891 the combined cumulative emissions of the current members of the European Union reached and subsequently surpassed the level of the UK. Since 1917 US cumulative emissions have been the largest. Over the entire period 1850-2022, US cumulative emissions amounted to 115GtC (24% of world total), the EU's to 80 GtC (17%), and China's to 70 GtC (15%).

In addition to the estimates of fossil $CO_2$ emissions that we provide here (see Methods), there are three global datasets with long time series that include all sources of fossil $CO_2$ emissions: CDIAC-FF (Gilfillan and Marland, 2021), CEDS version v_2021_04_21 (Hoesly et al., 2018; O'Rourke et al., 2021) and PRIMAP-hist version 2.4.2 (Gütschow et al., 2016; Gütschow and Pflüger, 2023), although these datasets are not entirely independent from each other (Andrew, 2020a). CDIAC-FF has the lowest cumulative emissions over 1750-2018 at 440 GtC, GCP has 444 GtC, CEDS 445 GtC, PRIMAP-hist TP 453 GtC, and PRIMAP-hist CR 452 GtC. CDIAC-FF excludes emissions from lime production. CEDS has higher emissions from international shipping in recent years, while PRIMAP-hist has higher fugitive emissions than the other datasets. However, in general these four datasets are in relative agreement as to total historical global emissions of fossil $CO_2$.



### 3.1.2 Recent period 1960-2022

Global fossil $CO_2$ emissions, $E_{FOS}$ (including the cement carbonation sink), have increased every decade from an average of $3.0 \pm 0.2$ GtC yr$^{-1}$ for the decade of the 1960s to an average of $9.6 \pm 0.5$ GtC yr$^{-1}$ during 2013-2022 (Table 7, Figure 2 and Figure 5). The growth rate in these emissions decreased between the 1960s and the 1990s, from 4.3% yr$^{-1}$ in the 1960s (1960-1969), 3.2% yr$^{-1}$ in the 1970s (1970-1979), 1.6% yr$^{-1}$ in the 1980s (1980-1989), to 1.0% yr$^{-1}$ in the 1990s (1990-1999). After this period, the growth rate began increasing again in the 2000s at an average growth rate of 2.8% yr$^{-1}$, decreasing to 0.5% yr$^{-1}$ for the last decade (2013-2022). China's emissions increased by +1.6% yr$^{-1}$ on average over the last 10 years dominating the global trend, and India's emissions increased by +3.5% yr$^{-1}$, while emissions decreased in EU27 by $-1.7$% yr$^{-1}$, and in the USA by $-1.0$% yr$^{-1}$. Figure 6 illustrates the spatial distribution of fossil fuel emissions for the 2013-2022 period.

$E_{FOS}$ reported here includes the uptake of $CO_2$ by cement via carbonation which has increased with increasing stocks of cement products, from an average of 20 MtC yr$^{-1}$ (0.02 GtC yr$^{-1}$) in the 1960s to an average of 206 MtC yr$^{-1}$ (0.21 GtC yr$^{-1}$) during 2013-2022 (Figure 5).

### 3.1.3 Final year 2022

Global fossil $CO_2$ emissions were slightly higher, 0.85%, in 2022 than in 2021, with an increase of less than 0.1 GtC to reach $9.9 \pm 0.5$ GtC (including the 0.2 GtC cement carbonation sink) in 2022 (Figure 5), distributed among coal (41%), oil (32%), natural gas (21%), cement (4%), flaring (1%), and others (1%). Compared to the previous year, 2022 emissions from coal and oil increased by 1.6% and 3.3% respectively, while emissions from gas and cement respectively decreased by 2.2% and 5.7%. All growth rates presented are adjusted for the leap year, unless stated otherwise.

In 2022, the largest absolute contributions to global fossil $CO_2$ emissions were from China (31%), the USA (14%), India (8%), and the EU27 (7%). These four regions account for 59% of global fossil $CO_2$ emissions, while the rest of the world contributed 41%, including international aviation and marine bunker fuels (2.6% of the total). Growth rates for these countries from 2021 to 2022 were 0.9% (China), 1% (USA), -1.9% (EU27), and 5.8% (India), with +0.6% for the rest of the world. The per-capita fossil $CO_2$ emissions in 2022 were 1.3 tC person$^{-1}$ yr$^{-1}$ for the globe, and were 4.1 (USA), 2.2 (China), 1.7 (EU27) and 0.5 (India) tC person$^{-1}$ yr$^{-1}$ for the four highest emitters (Figure 5).

### 3.1.4 Year 2023 Projection

Globally, we estimate that global fossil $CO_2$ emissions (including cement carbonation) will grow by 1.2% in 2023 (0.2% to 2.3%) to 10.0 GtC (36.8 GtCO$_2$), exceeding the pre-COVID19 2019 emission levels of 9.9 GtC (36.3 GtCO$_2$). Global increase in 2023 emissions per fuel types are projected to be +1.1% (range -0.2% to 2.4%) for coal, +1.8% (range 0.8% to 2.9%) for oil, +0.3% (range -0.6% to 1.3%) for natural gas, and 1.8% (range 0.2% to 3.4%) for cement.



821 For China, projected fossil emissions in 2023 are expected to increase by 4% (range 1.9% to 6.2%) compared

822 with 2022 emissions, bringing 2023 emissions for China around 3.2 GtC $yr^{-1}$ (11.9 $GtCO_2$ $yr^{-1}$). Changes in fuel

823 specific projections for China are 3.5% for coal, 7.7% for oil, 6.4% natural gas, and 0.2% for cement.

824 For the USA, the Energy Information Administration (EIA) emissions projection for 2023 combined with

825 cement clinker data from USGS gives an decrease of 3.4% (range -5.9% to -0.9%) compared to 2022, bringing

826 USA 2023 emissions to around 1.3 GtC $yr^{-1}$ (4.9 $GtCO_2$ $yr^{-1}$). This is based on separate projections for coal -

827 19.9%, oil -0.7%, natural gas +1.7%, and cement -3.2%.

828 For the European Union, our projection for 2023 is for a decrease of 7.1% (range -9.6% to -4.6%) over 2022,

829 with 2023 emissions around 0.7 GtC $yr^{-1}$ (2.6 $GtCO_2$ $yr^{-1}$). This is based on separate projections for coal of -

830 19.6%, oil -0.9%, natural gas -6.6%, and cement unchanged.

831 For India, our projection for 2023 is an increase of 8% (range of 7.9% to 8.0%) over 2022, with 2023 emissions

832 around 0.8 GtC $yr^{-1}$ (3.1 $GtCO_2$ $yr^{-1}$). This is based on separate projections for coal of +9.2%, oil +5.2%, natural

833 gas +4.4%, and cement +8.1%.

834 For the rest of the world, the expected growth rate for 2023 is 0.9% (range -0.8% to 2.6%) with 2023 emissions

835 around 4.2 GtC $yr^{-1}$ (15.2 $GtCO_2$ $yr^{-1}$). The fuel-specific projected 2023 growth rates for the rest of the world

836 are: +1% for coal, +1.5% for oil, -0.3% for natural gas, +2.6% for cement.

837 **3.2  Emissions from Land Use Changes**

838 **3.2.1  Historical period 1850-2022**

839 Cumulative $CO_2$ emissions from land-use changes ($E_{LUC}$) for 1850-2022 were $220 \pm 65$ GtC (Table 8; Figure 3;

840 Figure 15). The cumulative emissions from $E_{LUC}$ show a large spread among individual estimates of 150 GtC

841 (H&C2023), 290 GtC (BLUE), and 215 GtC (OSCAR) for the three bookkeeping models and a similar wide

842 estimate of $210 \pm 65$ GtC for the DGVMs (all cumulative numbers are rounded to the nearest 5 GtC). These

843 estimates are broadly consistent with indirect constraints from vegetation biomass observations, giving

844 cumulative emissions of $155 \pm 50$ GtC over the 1901-2012 period (Li et al., 2017). However, given the large

845 spread, a best estimate is difficult to ascertain.

846 **3.2.2  Recent period 1960-2022**

847 In contrast to growing fossil emissions, $CO_2$ emissions from land-use, land-use change, and forestry remained

848 relatively constant over the 1960-1999 period. Since the 1990s they have shown a slight decrease of about 0.1

849 GtC per decade, reaching $1.3 \pm 0.7$ GtC $yr^{-1}$ for the 2013-2022 period (Table 7), but with large spread across

850 estimates (Table 5, Figure 7). Different from the bookkeeping average, the DGVMs average grows slightly

851 larger over the 1970-2022 period and shows no sign of decreasing emissions in the recent decades (Table 5,

852 Figure 7). This is, however, expected as DGVM-based estimates include the loss of additional sink capacity,

853 which grows with time, while the bookkeeping estimates do not (Supplement S.6.4).



We separate net E$_{LUC}$ into five component fluxes to gain further insight into the drivers of net emissions:
deforestation, forest (re-)growth, wood harvest and other forest management, peat drainage and peat fires, and
all other transitions (Figure 7c; Sec. C.2.1). We further decompose the deforestation and the forest (re-)growth
term into contributions from shifting cultivation vs permanent forest cover changes (Figure 7d). Averaged over
the 2013-2022 period and over the three bookkeeping estimates, fluxes from deforestation amount to 1.9 [1.5 to
2.4] GtC yr$^{-1}$ (Table 5), of which 1.1 [1.0, 1.2] GtC yr$^{-1}$ are from permanent deforestation. Fluxes from forest
(re-)growth amount to -1.3 [-1.5, -0.9] GtC yr$^{-1}$ (Table 5), of which -0.5 [-0.8 to-0.2] GtC yr$^{-1}$ are from
re/afforestation and the remainder from forest regrowth in shifting cultivation cycles. Emissions from wood
harvest and other forest management (0.2 [0.0, 0.6] GtC yr$^{-1}$), peat drainage and peat fires (0.3 [0.3, 0.3] GtC yr$^{-}$
$^{1}$) and the net flux from other transitions (0.1 [0.0, 0.3] GtC yr$^{-1}$) are substantially less important globally (Table
5). However, the small net flux from wood harvest and other forest management contains substantial gross
fluxes that largely compensate each other (see Figure S7): 1.3 [0.9, 2.0] GtC yr$^{-1}$ emissions result from the
decomposition of slash and the decay of wood products and -1.1 [-1.3, -0.8] GtC yr$^{-1}$ removals result from
regrowth after wood harvesting. This split into component fluxes clarifies the potentials for emission reduction
and carbon dioxide removal: the emissions from permanent deforestation - the largest of our component fluxes -
could be halted (largely) without compromising carbon uptake by forests, contributing substantially to emissions
reduction. By contrast, reducing wood harvesting would have limited potential to reduce emissions as it would
be associated with less forest regrowth; removals and emissions cannot be decoupled here on long timescales. A
similar conclusion applies to removals and emissions from shifting cultivation, which we have therefore
separated out. Carbon Dioxide Removal (CDR) in forests could instead be increased by permanently increasing
the forest cover through re/afforestation. Our estimate of about -0.5 [-0.8, -0.2] GtC yr$^{-1}$ (of which about two
thirds are located in non-Annex-I countries, in particular in China) removed on average each year during 2013-
2022 by re/afforestation is very similar to independent estimates that were derived from NGHGIs for 2022.
Re/afforestation constitutes the vast majority of all current CDR (Powis et al., 2023). Though they cannot be
compared directly to annual fluxes from the atmosphere, CDR through transfers between non-atmospheric
reservoirs such as in durable HWPs, biochar or BECCS comprise much smaller amounts of carbon. 61 MtC yr$^{-1}$
have been estimated to be transferred to HWPs in 2022, and BECCS projects have been estimated to store 0.5
MtC yr$^{-1}$ in geological projects worldwide (Powis et al., 2023). "Blue carbon", i.e. coastal wetland management
such as restoration of mangrove forests, saltmarshes and seagrass meadows, though at the interface of land and
ocean carbon fluxes, are counted towards the land-use sector as well. Currently, bookkeeping models do not
include blue carbon; however, current CDR deployment in coastal wetlands is small globally.
The small declining trend of E$_{LUC}$ over the last three decades is a result of total deforestation emissions showing
no clear trend, while forest regrowth has provided steadily increasing removals. Since the processes behind
gross removals, foremost forest regrowth and soil recovery, are all slow, while gross emissions include a large
instantaneous component, short-term changes in land-use dynamics, such as a temporary decrease in
deforestation, influences gross emissions dynamics more than gross removals dynamics, which rather are a
response to longer-term dynamics. Component fluxes often differ more across the three bookkeeping estimates
than the net flux, which is expected due to different process representation; in particular, treatment of shifting
cultivation, which increases both gross emissions and removals, differs across models, but also net and gross



wood harvest fluxes show high uncertainty. By contrast, models agree relatively well for emissions from
permanent deforestation emissions and removals by re/afforestation.
Overall, highest land-use emissions occur in the tropical regions of all three continents. The top three emitters
(both cumulatively 1959-2022 and on average over 2013-2022) are Brazil (in particular the Amazon Arc of
Deforestation), Indonesia and the Democratic Republic of the Congo, with these 3 countries contributing 0.7
GtC yr$^{-1}$ or 55% of the global net land-use emissions (average over 2013-2022) (Figure 6b). This is related to
massive expansion of cropland, particularly in the last few decades in Latin America, Southeast Asia, and sub-
Saharan Africa (Hong et al., 2021), to a substantial part for export of agricultural products (Pendrill et al., 2019).
Emission intensity is high in many tropical countries, particularly of Southeast Asia, due to high rates of land
conversion in regions of carbon-dense and often still pristine, undegraded natural forests (Hong et al., 2021).
Emissions are further increased by peat fires in equatorial Asia (GFED4s, van der Werf et al., 2017). Uptake due
to land-use change occurs, particularly in Europe, partly related to expanding forest area as a consequence of the
forest transition in the 19$^{th}$ and 20$^{th}$ century and subsequent regrowth of forest (Figure 6b) (Mather 2001;
McGrath et al., 2015).
While the mentioned patterns are robust and supported by independent literature, we acknowledge that model
spread is substantially larger on regional than global levels, as has been shown for bookkeeping models (Bastos
et al., 2021) as well as DGVMs (Obermeier et al., 2021). Assessments for individual regions will be performed
as part of REgional Carbon Cycle Assessment and Processes (RECCAP2; Ciais et al., 2020) or already exist for
selected regions (e.g., for Europe by Petrescu et al., 2020, for Brazil by Rosan et al., 2021, for 8 selected
countries/regions in comparison to inventory data by Schwingshackl et al., 2022).
National GHG inventory data (NGHGI) under the LULUCF sector or data submitted by countries to FAOSTAT
differ from the global models' definition of $E_{LUC}$. In the NGHGI reporting, the natural fluxes ($S_{LAND}$) are
counted towards $E_{LUC}$ when they occur on managed land (Grassi et al., 2018). In order to compare our results to
the NGHGI approach, we perform a translation of our $E_{LUC}$ estimates by subtracting $S_{LAND}$ in managed forest
from the DGVMs simulations (following Grassi et al., 2021) from the bookkeeping $E_{LUC}$ estimate (see
Supplement S.2.3). For the 2013-2022 period, we estimate that 2.0 GtC yr$^{-1}$ of $S_{LAND}$ occurred in managed
forests. Subtracting this value from $E_{LUC}$ changes $E_{LUC}$ from being a source of 1.3 GtC yr$^{-1}$ to a sink of 0.8 GtC
yr$^{-1}$, very similar to the NGHGI estimate that yields a sink of 0.7 GtC yr$^{-1}$ (Table 9). The translation approach
has been shown to be generally applicable also on country-level (Grassi et al., 2023; Schwingshackl et al.,
2022). Country-level analysis suggests, e.g., that the bookkeeping method estimates higher deforestation
emissions than the national report in Indonesia, but less $CO_2$ removal by afforestation than the national report in
China. The fraction of the natural $CO_2$ sinks that the NGHGI estimates include differs substantially across
countries, related to varying proportions of managed vs total forest areas (Schwingshackl et al., 2022). By
comparing $E_{LUC}$ and NGHGI on the basis of the component fluxes used above, we find that our estimates
reproduce very closely the NGHGI estimates for emissions from permanent deforestation (1.1 GtC yr$^{-1}$ averaged
over 2013-2022). Forest fluxes, that is, (re-)growth from re/afforestation plus the net flux from wood harvesting
and other forest management, constitute a large sink in the NGHGI (-1.9 GtC yr$^{-1}$ averaged over 2013-2022),
since they also include $S_{LAND}$ in managed forests. Summing up the bookkeeping estimates of (re-)growth from
re/afforestation and the net flux from wood harvesting and other forest management and adding $S_{LAND}$ in



managed forests yields a flux of -2.3 GtC yr$^{-1}$ (averaged over 2013-2022), which compares well with the
NGHGI estimate. Emissions from organic soils in NGHGI are similar to the estimates based on the bookkeeping
approach and the external peat drainage and burning datasets. The net flux from other transitions is small in both
NGHGI and bookkeeping estimates, but a difference in sign (small source in bookkeeping estimates, small sink
in NGHGI) creates a notable difference between NGHGI and bookkeeping estimates. Though estimates between
NGHGI, FAOSTAT and the translated budget estimates still differ in value and need further analysis, the
approach suggested by Grassi et al. (2023), which we adopt here, provides a feasible way to relate the global
models' and NGHGI approach to each other and thus link the anthropogenic carbon budget estimates of land
CO$_2$ fluxes directly to the Global Stocktake, as part of UNFCCC Paris Agreement.

### 3.2.3    Final year 2022

The global CO$_2$ emissions from land-use change are estimated as 1.2 ± 0.7 GtC in 2022, similar to the 2020 and
2021 estimates. However, confidence in the annual change remains low. Effects of the COVID-19 pandemic on
land-use change have turned out to be country-specific as global market mechanisms, national economics and
changes in household income all could act to curb or enhance deforestation (Wunder et al., 2021). Concerns
about enhanced deforestation due to weakened environmental protection and monitoring in tropical countries
(Brancalion et al., 2020, Vale et al., 2021) have been confirmed only for some countries (Cespedes et al., 2023).
For example, a recent study suggests slightly increased deforestation rates for the Democratic Republic of
Congo linked in particular to post-pandemic economic recovery in the mining sector, while deforestation trends
in Brazil seem to have been unaffected. Land use dynamics may be further altered by the Russian invasion of
Ukraine, but scientific evidence related to international dependencies (like a shift to tropical palm oil to alleviate
dependencies on sunflower oil) so far is very limited and recent changes will not be reflected by the land-use
forcing applied in the global models. High food prices, which preceded but were exacerbated by the war (FAO,
2022), are generally linked to higher deforestation (Angelsen and Kaimowitz, 1999). A new wave of cropland
abandonment in the conflict region may increase the substantial Eastern European carbon sink due to land-use
changes, but sanctions being placed on trade may also incentivise domestic agricultural production, thus leading
to recultivation of abandoned areas in Russia (Winkler et al., 2023).

### 3.2.4    Year 2023 Projection

In Indonesia, peat fire emissions are below average (12 Tg C through September 29 2023) despite El Niño
conditions, which in general lead to more fires. Tropical deforestation and degradation fires in Indonesia are
around average (13 Tg C through September 29 2023), but higher than in the previous year, which had a
relatively wet dry season (GFED4.1s, van der Werf et al., 2017; see also
https://www.geo.vu.nl/~gwerf/GFED/GFED4/tables/GFED4.1s_C.txt). In South America, emissions from
tropical deforestation and degradation fires are among the lowest over the last decades (64 Tg C through
September 29 2023). Effects of the El Niño in the Amazon, such as droughts, are not expected before 2024.
Disentangling the degree to which interannual variability in rainfall patterns and stronger environmental
protection measures in both Indonesia after their 2015 high fire season and in Brazil after the change in
government in Brazil play a role in this is an important research topic. Cumulative fire emission estimates



through September 29 2023 are 155 Tg C for global deforestation and degradation fires and 12 Tg C for
peatland fires in Indonesia (https://www.geo.vu.nl/~gwerf/GFED/GFED4/tables/GFED4.1s_C.txt).
Based on these estimates, we expect $E_{LUC}$ emissions of around 1.1 GtC (4.1 GtCO$_2$) in 2023. Our preliminary
estimate of $E_{LUC}$ for 2023 is substantially lower than the 2013-2022 average, which saw years of anomalously
dry conditions in Indonesia and high deforestation fires in South America (Friedlingstein et al., 2022b). Note
that although our extrapolation includes tropical deforestation and degradation fires, degradation attributable to
selective logging, edge-effects or fragmentation is not captured. Further, deforestation and fires in deforestation
zones may become more disconnected, partly due to changes in legislation in some regions. For example, Van
Wees et al. (2021) found that the contribution from fires to forest loss decreased in the Amazon and in Indonesia
over the period of 2003-2018.
**3.3     CDR not based on vegetation**
Besides the CDR through land-use (Sec. 3.2), the atmosphere to geosphere flux of carbon resulting from carbon
dioxide removal (CDR) activity is currently 0.003 MtC/yr, with 0.002 MtC/yr of DACCS and 0.001 MtC/yr of
enhanced weathering projects. This represents an offset of about 0.03% of current fossil fuel emissions.
**3.4     Total anthropogenic emissions**
Cumulative anthropogenic CO$_2$ emissions for 1850-2022 totalled 695 ± 70 GtC (2550 ± 260 GtCO$_2$), of which
70% (485 GtC) occurred since 1960 and 33% (235 GtC) since 2000 (Table 7 and 8). Total anthropogenic
emissions more than doubled over the last 60 years, from 4.6 ± 0.7 GtC yr$^{-1}$ for the decade of the 1960s to an
average of 10.9 ± 0.8 GtC yr$^{-1}$ during 2013-2022, and reaching 11.1 ± 0.9 GtC (40.7 ± 3.3 GtCO$_2$) in 2022. For
2023, we project global total anthropogenic CO$_2$ emissions from fossil and land use changes to be also around
11.2 GtC (40.9 GtCO$_2$). All values here include the cement carbonation sink (currently about 0.2 GtC yr$^{-1}$).
During the historical period 1850-2022, 31% of historical emissions were from land use change and 69% from
fossil emissions. However, fossil emissions have grown significantly since 1960 while land use changes have
not, and consequently the contributions of land use change to total anthropogenic emissions were smaller during
recent periods (18% during the period 1960-2022 and down to 12% over the 2013-2022 period).
**3.5     Atmospheric CO$_2$**
**3.5.1     Historical period 1850-2022**
Atmospheric CO$_2$ concentration was approximately 278 parts per million (ppm) in 1750, reaching 300 ppm in
the 1910s, 350 ppm in the late 1980s, and reaching 417.07 ± 0.1 ppm in 2022 (Lan et al., 2023; Figure 1). The
mass of carbon in the atmosphere increased by 48% from 590 GtC in 1750 to 886 GtC in 2022. Current CO$_2$
concentrations in the atmosphere are unprecedented in the last 2 million years and the current rate of
atmospheric CO$_2$ increase is at least 10 times faster than at any other time during the last 800,000 years
(Canadell et al., 2021).



### 3.5.2 Recent period 1960-2022

The growth rate in atmospheric $CO_2$ level increased from $1.7 \pm 0.07$ GtC yr$^{-1}$ in the 1960s to $5.2 \pm 0.02$ GtC yr$^{-1}$ during 2013-2022 with important decadal variations (Table 7, Figure 3 and Figure 4). During the last decade (2013-2022), the growth rate in atmospheric $CO_2$ concentration continued to increase, albeit with large interannual variability (Figure 4).

The airborne fraction (AF), defined as the ratio of atmospheric $CO_2$ growth rate to total anthropogenic emissions:

$$AF = G_{ATM} / (E_{FOS} + E_{LUC}) \tag{2}$$

provides a diagnostic of the relative strength of the land and ocean carbon sinks in removing part of the anthropogenic $CO_2$ perturbation. The evolution of AF over the last 60 years shows no significant trend, remaining at around 44%, albeit showing a large interannual and decadal variability driven by the year-to-year variability in $G_{ATM}$ (Figure 9). The observed stability of the airborne fraction over the 1960-2020 period indicates that the ocean and land $CO_2$ sinks have been removing on average about 56% of the anthropogenic emissions (see Sections 3.6.2 and 3.7.2).

### 3.5.3 Final year 2022

The growth rate in atmospheric $CO_2$ concentration was $4.6 \pm 0.2$ GtC ($2.18 \pm 0.08$ ppm) in 2022 (Figure 4; Lan et al., 2023), below the 2021 growth rate ($5.2 \pm 0.2$ GtC) or the 2013-2022 average ($5.2 \pm 0.02$ GtC).

### 3.5.4 Year 2023 Projection

The 2023 growth in atmospheric $CO_2$ concentration ($G_{ATM}$) is projected to be about 4.0 GtC (1.89 ppm). This is the average of the Holt–Winters method (3.7 GtC, 1.73 ppm) and ESMs the multi-model mean (4.4 GtC, 2.05 ppm). The 2023 atmospheric $CO_2$ concentration, averaged over the year, is expected to reach the level of 419.2 ppm, 51% over the pre-industrial level.

## 3.6 Ocean Sink

### 3.6.1 Historical period 1850-2022

Cumulated since 1850, the ocean sink adds up to $180 \pm 35$ GtC, with more than two thirds of this amount (125 GtC) being taken up by the global ocean since 1960. Over the historical period, the ocean sink increased in pace with the anthropogenic emissions exponential increase (Figure 3). Since 1850, the ocean has removed 26% of total anthropogenic emissions.

### 3.6.2 Recent period 1960-2022

The ocean $CO_2$ sink increased from $1.1 \pm 0.4$ GtC yr$^{-1}$ in the 1960s to $2.8 \pm 0.4$ GtC yr$^{-1}$ during 2013-2022 (Table 7), with interannual variations of the order of a few tenths of GtC yr$^{-1}$ (Figure 10). The ocean-borne fraction ($S_{OCEAN}/(E_{FOS}+E_{LUC})$) has been remarkably constant around 25% on average (Figure 9c), with variations



around this mean illustrating the decadal variability of the ocean carbon sink. So far, there is no indication of a
decrease in the ocean-borne fraction from 1960 to 2022. The increase of the ocean sink is primarily driven by
the increased atmospheric $CO_2$ concentration, with the strongest $CO_2$ induced signal in the North Atlantic and
the Southern Ocean (Figure 11a). The effect of climate change is much weaker, reducing the ocean sink globally
by $0.16 \pm 0.04$ GtC yr$^{-1}$ (-6.7% of $S_{OCEAN}$) during 2013-2022 (all models simulate a weakening of the ocean sink
by climate change, range -4.3 to -10.3%), and does not show clear spatial patterns across the GOBMs ensemble
(Figure 11b). This is the combined effect of change and variability in all atmospheric forcing fields, previously
attributed, in one model, to wind and temperature changes (LeQuéré et al., 2010).
The global net air-sea $CO_2$ flux is a residual of large natural and anthropogenic $CO_2$ fluxes into and out of the
ocean with distinct regional and seasonal variations (Figure 6 and B1). Natural fluxes dominate on regional
scales, but largely cancel out when integrated globally (Gruber et al., 2009). Mid-latitudes in all basins and the
high-latitude North Atlantic dominate the ocean $CO_2$ uptake where low temperatures and high wind speeds
facilitate $CO_2$ uptake at the surface (Takahashi et al., 2009). In these regions, formation of mode, intermediate
and deep-water masses transport anthropogenic carbon into the ocean interior, thus allowing for continued $CO_2$
uptake at the surface. Outgassing of natural $CO_2$ occurs mostly in the tropics, especially in the equatorial
upwelling region, and to a lesser extent in the North Pacific and polar Southern Ocean, mirroring a well-
established understanding of regional patterns of air-sea $CO_2$ exchange (e.g., Takahashi et al., 2009, Gruber et
al., 2009). These patterns are also noticeable in the Surface Ocean CO2 Atlas (SOCAT) dataset, where an ocean
$f$CO$_2$ value above the atmospheric level indicates outgassing (Figure S1). This map further illustrates the data-
sparsity in the Indian Ocean and the southern hemisphere in general.
Interannual variability of the ocean carbon sink is driven by climate variability with a first-order effect from a
stronger ocean sink during large El Niño events (e.g., 1997-1998) (Figure 10; Rödenbeck et al., 2014, Hauck et
al., 2020; McKinley et al. 2017). The GOBMs show the same patterns of decadal variability as the mean of the
$f$CO$_2$-products, with a stagnation of the ocean sink in the 1990s and a strengthening since the early 2000s
(Figure 10; Le Quéré et al., 2007; Landschützer et al., 2015, 2016; DeVries et al., 2017; Hauck et al., 2020;
McKinley et al., 2020, Gruber et al., 2023). Different explanations have been proposed for this decadal
variability, ranging from the ocean's response to changes in atmospheric wind and pressure systems (e.g., Le
Quéré et al., 2007, Keppler and Landschützer, 2019), including variations in upper ocean overturning circulation
(DeVries et al., 2017) to the eruption of Mount Pinatubo and its effects on sea surface temperature and slowed
atmospheric $CO_2$ growth rate in the 1990s (McKinley et al., 2020). The main origin of the decadal variability is
a matter of debate with a number of studies initially pointing to the Southern Ocean (see review in Canadell et
al., 2021), but also contributions from the North Atlantic and North Pacific (Landschützer et al., 2016, DeVries
et al., 2019), or a global signal (McKinley et al., 2020) were proposed.
Although all individual GOBMs and $f$CO$_2$-products fall within the observational constraint, the ensemble means
of GOBMs, and $f$CO$_2$-products adjusted for the riverine flux diverge over time with a mean offset increasing
from 0.30 GtC yr$^{-1}$ in the 1990s to 0.57 GtC yr$^{-1}$ in the decade 2013-2022 and reaching 0.61 GtC yr$^{-1}$ in 2022.
The $S_{OCEAN}$ positive trend over time diverges by a factor two since 2002 (GOBMs: $0.24 \pm 0.07$ GtC yr$^{-1}$ per
decade, $f$CO$_2$-products: $0.48 \pm 0.11$ GtC yr$^{-1}$ per decade, $S_{OCEAN}$: 0.36 GtC yr$^{-1}$ per decade) and by a factor of 2.5
since 2010 (GOBMs: $0.16 \pm 0.15$ GtC yr$^{-1}$ per decade, $f$CO$_2$-products: $0.42 \pm 0.18$ GtC yr$^{-1}$ per decade$^,$ $S_{OCEAN}$:



0.29 GtC yr$^{-1}$ per decade). The $f$CO$_2$-product estimate is slightly different compared to Friedlingstein et al.
(2022b) as a result of an updated submission of the NIES-ML3 product (previously NIES-NN), however the
difference in the integrated mean flux is small.
The discrepancy between the two types of estimates stems from a larger S$_{OCEAN}$ trend in the northern and
southern extra-tropics since around 2002 (Figure 13). Note that the discrepancy in the mean flux, which was
located in the Southern Ocean in previous versions of the GCB, has been reduced due to the choice of the
regional river flux adjustment (Lacroix et al., 2020 instead of Aumont et al., 2001). This comes at the expense of
a new discrepancy in the mean S$_{OCEAN}$ of about 0.2 GtC yr$^{-1}$ in the tropics. Likely explanations for the
discrepancy in the trends in the high-latitudes are data sparsity and uneven data distribution (Bushinsky et al.,
2019, Gloege et al., 2021, Hauck et al., 2023). In particular, two $f$CO$_2$-products that are part of the GCB
ensemble were shown to overestimate the Southern Ocean CO$_2$ flux trend by 50 and 130% based on current
sampling in a model subsampling experiment (Hauck et al., 2023). Another likely contributor to the discrepancy
between GOBMs and $f$CO$_2$-products are model biases (as indicated by the large model spread in the South,
Figure 13, and the larger model-data $f$CO$_2$ mismatch, Figure S2).
In previous GCB releases, the ocean sink 1959-1989 was only estimated by GOBMs due to the absence of $f$CO$_2$
observations. Now, the first data-based estimates extending back to 1957/58 are becoming available (Jena-MLS,
Rödenbeck et al., 2022, LDEO-HPD, Bennington et al., 2022; Gloege et al., 2022). These are based on a multi-
linear regression of $p$CO$_2$ with environmental predictors (Rödenbeck et al., 2022) or on model-data $p$CO$_2$ misfits
and their relation to environmental predictors (Bennington et al., 2022). The Jena-MLS and LDEO-HPD
estimates fall well within the range of GOBM estimates and have a correlation of 0.99 and 0.98 respectively
with S$_{OCEAN}$ for the period 1959-2022 (and 0.98 and 0.97 for the 1959-1989 period). They agree well on the
mean S$_{OCEAN}$ estimate since 1977 with a slightly higher amplitude of variability (Figure 10). Until 1976, Jena-
MLS and LDEO-HPD are respectively about 0.25 GtCyr$^{-1}$ and about 0.1 GtCyr$^{-1}$ below the central S$_{OCEAN}$
estimate. The agreement especially on phasing of variability is impressive in both products, and the
discrepancies in the mean flux 1959-1976 could be explained by an overestimated trend of Jena-MLS
(Rödenbeck et al., 2022). Bennington et al. (2022) report a larger flux into the pre-1990 ocean than in Jena-
MLS, although lower than S$_{OCEAN}$.
The reported S$_{OCEAN}$ estimate from GOBMs and $f$CO$_2$-products is 2.2 ± 0.4 GtC yr$^{-1}$ over the period 1994 to
2007, which is in excellent agreement with the ocean interior estimate of 2.2 ± 0.4 GtC yr$^{-1}$, which accounts for
the climate effect on the natural CO$_2$ flux of −0.4 ± 0.24 GtC yr$^{-1}$ (Gruber et al., 2019) to match the definition of
S$_{OCEAN}$ used here (Hauck et al., 2020). This comparison depends critically on the estimate of the climate effect
on the natural CO$_2$ flux, which is smaller from the GOBMs (-0.1 GtC yr$^{-1}$) than in Gruber et al. (2019).
Uncertainties of these two estimates would also overlap when using the GOBM estimate of the climate effect on
the natural CO$_2$ flux.
During 2010-2016, the ocean CO$_2$ sink appears to have intensified in line with the expected increase from
atmospheric CO$_2$ (McKinley et al., 2020). This effect is slightly stronger in the $f$CO$_2$-products (Figure 10, ocean
sink 2016 minus 2010, GOBMs: +0.42 ± 0.10 GtC yr$^{-1}$, $f$CO$_2$-products: +0.48 ± 0.10 GtC yr$^{-1}$). The reduction of
-0.14 GtC yr$^{-1}$ (range: -0.39 to +0.01 GtC yr$^{-1}$) in the ocean CO$_2$ sink in 2017 is consistent with the return to



normal conditions after the El Niño in 2015/16, which caused an enhanced sink in previous years. After an
increasing $S_{OCEAN}$ in 2018 and 2019, 2017, the GOBM and $f$CO$_2$-product ensemble means suggest a decrease of
$S_{OCEAN}$, related to the triple La Niña event 2020-2023.

### 3.6.3    Final year 2022

The estimated ocean CO$_2$ sink is 2.8 ± 0.4 GtC for 2022. This is a small decrease of 0.05 GtC compared to 2021,
in line with the expected sink weakening from persistent La Niña conditions. GOBM and $f$CO$_2$-product
estimates consistently result in a near-stagnation of $S_{OCEAN}$ (GOBMs: -0.01 ±0.05 GtC, $f$CO$_2$-products: -0.09
±0.10 GtC). Four models and six $f$CO$_2$-products show a decrease in $S_{OCEAN}$ (GOBMs down to -0.09 GtC, $f$CO$_2$-
products down to -0.25 GtC), while one model shows no change and five models and two $f$CO$_2$-products show
an increase in $S_{OCEAN}$ (GOBMs up to 0.07 GtC, $f$CO$_2$-products up to 0.15 GtC; Figure 10). The $f$CO$_2$-products
have a larger uncertainty at the end of the reconstructed time series (tail effect, e.g., Watson et al., 2020).
Specifically, the $f$CO$_2$-products' estimate of the last year is regularly adjusted in the following release owing to
the tail effect and an incrementally increasing data availability. While the monthly grid cells covered may have a
lag of only about a year (Figure 10 inset), the values within grid cells may change with 1-5 years lag (see
absolute number of observations plotted in previous GCB releases).

### 3.6.4    Year 2023 Projection

Using a feed-forward neural network method (see Section 2.5.2) we project an ocean sink of 2.9 GtC for 2023.
This is slightly higher than for the year 2022 and could mark a reversal of the decreasing $S_{OCEAN}$ sink trend of
the past three years, due to the transition from persisting La Niña conditions to emerging El Niño conditions in
2023. The new set of ESMs predictions support this estimate with a 2023 ocean sink of around 3.1 [2.9, 3.2]
GtC.

### 3.6.5    Ocean Models Evaluation

The process-based model evaluation draws a generally positive picture with GOBMs scattered around the
observational values for Southern Ocean sea-surface salinity, Southern Ocean stratification index and surface
ocean Revelle factor (Section C3.3 and Table S10). However, the Atlantic Meridional Overturning Circulation
at 26°N is underestimated by 8 out of 10 GOBMs. It is planned to derive skill scores for the GOBMs in future
releases based on these metrics.
The model simulations allow to separate the anthropogenic carbon component (steady state and non-steady
state, sim D - sim A) and to compare the model flux and DIC inventory change directly to the interior ocean
estimate of Gruber et al. (2019) without further assumptions (Table S10). The GOBMs ensemble average of
anthropogenic carbon inventory changes 1994-2007 amounts to 2.4 GtC yr$^{-1}$ and is thus lower than the 2.6 ± 0.3
GtC yr$^{-1}$ estimated by Gruber et al. (2019) although within the uncertainty. Only four models with the highest
sink estimate fall within the range reported by Gruber et al. (2019). This suggests that the majority of the
GOBMs underestimate anthropogenic carbon uptake by 10-20%. Analysis of Earth System Models indicate that
an underestimation by about 10% may be due to biases in ocean carbon transport and mixing from the surface
mixed layer to the ocean interior (Goris et al., 2018, Terhaar et al., 2021, Bourgeois et al., 2022, Terhaar et al.,



2022), biases in the chemical buffer capacity (Revelle factor) of the ocean (Vaittinada Ayar et al., 2022; Terhaar
et al., 2022) and partly due to a late starting date of the simulations (mirrored in atmospheric $CO_2$ chosen for the
preindustrial control simulation, Table S2, Bronselaer et al., 2017, Terhaar et al., 2022). Interestingly, and in
contrast to the uncertainties in the surface $CO_2$ flux, we find the largest mismatch in interior ocean carbon
accumulation in the tropics (96% of the mismatch), with minor contributions from the north (3%) and the south
(<1%). These numbers deviate slightly from GCB2021 because of submission of the ACCESS model with a
high anthropogenic carbon accumulation, particularly in the Southern Ocean. The large discrepancy in
accumulation in the tropics highlights the role of interior ocean carbon redistribution for those inventories
(Khatiwala et al., 2009, DeVries et al., 2023).
The evaluation of the ocean estimates with the $f$CO$_2$ observations from the SOCAT v2023 dataset for the period
1990-2022 shows an RMSE from annually detrended data of 0.4 to 2.4 µatm for the seven $f$CO$_2$-products over
the globe (Figure S2). The GOBMs RMSEs are larger and range from 2.9 to 5.4 µatm. The RMSEs are
generally larger at high latitudes compared to the tropics, for both the $f$CO$_2$-products and the GOBMs. The
$f$CO$_2$-products have RMSEs of 0.3 to 2.8 µatm in the tropics, 0.7 to 2.3 µatm in the north, and 0.7 to 2.8 µatm in
the south. Note that the $f$CO$_2$-products are based on the SOCAT v2023 database, hence the SOCAT is not an
independent dataset for the evaluation of the $f$CO$_2$-products. The GOBMs RMSEs are more spread across
regions, ranging from 2.5 to 5.0 µatm in the tropics, 3.0 to 7.2 µatm in the North, and 3.7 to 8.5 µatm in the
South. The higher RMSEs occur in regions with stronger climate variability, such as the northern and southern
high latitudes (poleward of the subtropical gyres). The upper range of the model RMSEs have increased
somewhat relative to Friedlingstein et al. (2022b).
**3.7    Land Sink**
**3.7.1    Historical period 1850-2022**
Cumulated since 1850, the terrestrial $CO_2$ sink amounts to 225 ± 55 GtC, 32% of total anthropogenic emissions.
Over the historical period, the sink increased in pace with the anthropogenic emissions exponential increase
(Figure 3).
**3.7.2    Recent period 1960-2022**
The terrestrial $CO_2$ sink $S_{LAND}$ increased from 1.3 ± 0.5 GtC yr$^{-1}$ in the 1960s to 3.3 ± 0.8 GtC yr$^{-1}$ during 2013-
2022, with important interannual variations of up to 2 GtC yr$^{-1}$ generally showing a decreased land sink during
El Niño events (Figure 8), responsible for the corresponding enhanced growth rate in atmospheric $CO_2$
concentration. The larger land $CO_2$ sink during 2013-2022 compared to the 1960s is reproduced by all the
DGVMs in response to the increase in both atmospheric $CO_2$, nitrogen deposition, and the changes in climate,
and is consistent with constraints from the other budget terms (Table 5).
Over the period 1960 to present the increase in the global terrestrial $CO_2$ sink is largely attributed to the $CO_2$
fertilisation effect (Prentice et al., 2001, Piao et al., 2009, Schimel et al., 2015) and increased nitrogen
deposition (Huntzinger et al., 2017, O'Sullivan et al., 2019), directly stimulating plant photosynthesis and
increased plant water use in water limited systems, with a small negative contribution of climate change (Figure



11). There is a range of evidence to support a positive terrestrial carbon sink in response to increasing
atmospheric $CO_2$, albeit with uncertain magnitude (Walker et al., 2021). As expected from theory, the greatest
$CO_2$ effect is simulated in the tropical forest regions, associated with warm temperatures and long growing
seasons (Hickler et al., 2008) (Figure 11a). However, evidence from tropical intact forest plots indicate an
overall decline in the land sink across Amazonia (1985-2011), attributed to enhanced mortality offsetting
productivity gains (Brienen et al., 2015, Hubau et al., 2020). During 2013-2022 the land sink is positive in all
regions (Figure 6) with the exception of eastern Brazil, Bolivia, Paraguay, northern Venezuela, Southwest USA,
central Europe and Central Asia, North and South Africa, and eastern Australia, where the negative effects of
climate variability and change (i.e. reduced rainfall and/or increased temperature) counterbalance $CO_2$ effects.
This is clearly visible on Figure 11 where the effects of $CO_2$ (Figure 11a) and climate (Figure 11b) as simulated
by the DGVMs are isolated. The negative effect of climate is the strongest in most of South America, Central
America, Southwest US, Central Europe, western Sahel, southern Africa, Southeast Asia and southern China,
and eastern Australia (Figure 11b). Globally, over the 2013-2022 period, climate change reduces the land sink
by $0.68 \pm 0.62$ GtC yr$^{-1}$ (20% of $S_{LAND}$).
Most DGVMs have similar $S_{LAND}$ averaged over 2013-2022, and 14/20 models fall within the 1σ range of the
residual land sink [2.0-3.8 GtC yr$^{-1}$] (see Table 5), and all but one model are within the 2σ range [1.1-4.7 GtC yr$^{-1}$
]. The ED model is an outlier, with a land sink estimate of 5.7 GtC yr$^{-1}$, driven by a strong $CO_2$ fertilisation
effect (6.6 GtC yr$^{-1}$ in the $CO_2$ only (S1) simulation), that is offset by correspondingly high land-use emissions.
There are no direct global observations of the land sink, or the $CO_2$ fertilisation effect, and so we are not yet in a
position to rule out models based on component fluxes if the net land sink ($S_{LAND}$-$E_{LUC}$) is within the
observational uncertainty provided by atmospheric $O_2$ measurements (Table 5). Overall, therefore the spread
among models for the estimate of $S_{LAND}$ over the last decade has increased this year (0.8 GtC yr$^{-1}$) compared to
GCB2022 (0.6 GtC yr$^{-1}$).
Furthermore, DGVMs were compared against a data-constrained intermediate complexity model of the land
carbon cycle (CARDAMOM) (Bloom and Williams, 2015; Bloom et al., 2016). Results suggest good
correspondence between approaches at the interannual timescales, but divergence in the recent trend with
CARDAMOM simulating a stronger trend than the DGVMs (Figure S8).
Since 2020 the globe has experienced La Niña conditions which would be expected to lead to an increased land
carbon sink. A clear peak in the global land sink is not evident in $S_{LAND}$, and we find that a La Niña- driven
increase in tropical land sink is offset by a reduced high latitude extra-tropical land sink, which may be linked to
the land response to recent climate extremes. A notable difference from GCB2022 (2012-2021 $S_{LAND}$ mean) is
the reduced carbon losses across tropical drylands. Further, central Europe has switched from a sink of carbon to
a source, with the summer heatwave of 2022 (and associated drought and wildfire) causing widespread losses
(Peters et al., 2023). In the past years several regions experienced record-setting fire events. While global
burned area has declined over the past decades mostly due to declining fire activity in savannas (Andela et al.,
2017), forest fire emissions are rising and have the potential to counter the negative fire trend in savannas
(Zheng et al., 2021). Noteworthy events include the 2019-2020 Black Summer event in Australia (emissions of
roughly 0.2 GtC; van der Velde et al., 2021) and Siberia in 2021 where emissions approached 0.4 GtC or three

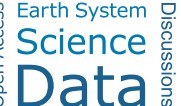

times the 1997-2020 average according to GFED4s. While other regions, including Western US and
Mediterranean Europe, also experienced intense fire seasons in 2021 their emissions are substantially lower.
Despite these regional negative effects of climate change on $S_{LAND}$, the efficiency of land to remove
anthropogenic $CO_2$ emissions has remained broadly constant over the last six decades, with a land-borne
fraction ($S_{LAND}/(E_{FOS}+E_{LUC})$) of around 30% (Figure 9b).

### 3.7.3   Final year 2022

The terrestrial $CO_2$ sink from the DGVMs ensemble was $3.8 \pm 0.8$ GtC in 2022, above the decadal average of
$3.3 \pm 0.8$ GtC yr$^{-1}$ (Figure 4, Table 7), and slightly above the 2021 sink of $3.5 \pm 1.0$ GtC, likely driven by the
persistent La Niña conditions. We note that the DGVMs estimate for 2022 is similar to the $3.7 \pm 1.0$ GtC yr$^{-1}$
estimate from the residual sink from the global budget ($E_{FOS}+E_{LUC}-G_{ATM}-S_{OCEAN}$) (Table 5).

### 3.7.4   Year 2023 Projection

Using a feed-forward neural network method we project a land sink of 3.0 GtC for 2023, 0.8 GtC smaller than
the 2022 estimate. As for the ocean sink, we attribute this to the emerging El Niño conditions in 2023, leading to
a reduced land sink. The ESMs do not provide an additional estimate of $S_{LAND}$ as they only simulate the net
atmosphere-land carbon flux ($S_{LAND}-E_{LUC}$).

### 3.7.5   Land Models Evaluation

The evaluation of the DGVMs shows generally high skill scores across models for runoff, and to a lesser extent
for vegetation biomass, GPP, and ecosystem respiration. These conclusions are supported by a more
comprehensive analysis of DGVM performance in comparison with benchmark data (Seiler et al., 2022). A
relative comparison of DGVM performance (Figure S3) suggests several DGVMs (CABLE-POP, CLASSIC,
OCN, ORCHIDEE) may outperform others at multiple carbon and water cycle benchmarks. However, results
from Seiler et al., 2022, also show how DGVM differences are often of similar magnitude compared with the
range across observational datasets.

### 3.8   Partitioning the carbon sinks

### 3.8.1   Global sinks and spread of estimates

In the period 2013-2022, the bottom-up view of global net ocean and land carbon sinks provided by the GCB,
$S_{OCEAN}$ for the ocean and $S_{LAND}-E_{LUC}$ for the land, agrees closely with the top-down global carbon sinks
delivered by the atmospheric inversions. This is shown in Figure 12, which visualises the individual decadal
mean atmosphere-land and atmosphere-ocean fluxes from each, along with the constraints on their sum offered
by the global fossil $CO_2$ emissions flux minus the atmospheric growth rate ($E_{FOS} - G_{ATM}$, $4.5 \pm 0.5$ Gt C yr$^{-1}$,
Table 7, shown as diagonal line on Figure 12). The GCB estimate for net atmosphere-to-surface flux ($S_{OCEAN}$ +
$S_{LAND} - E_{LUC}$) during 2013-2022 is $4.9 \pm 1.2$ Gt C yr$^{-1}$ (Table 7), with the difference to the diagonal representing
the budget imbalance ($B_{IM}$) of 0.4 GtC yr$^{-1}$ discussed in Section 3.9. By virtue of the inversion methodology, the



imbalance of the top-down estimates is < 0.1 GtC yr$^{-1}$ and thus scatter across the diagonal, inverse models
trading land for ocean fluxes in their solution. The independent constraint on the net atmosphere-to-surface flux
based on atmospheric $O_2$ is 4.4 ± 1.4 GtC yr$^{-1}$ over the 2013-2022 period (orange symbol on Figure 12), while
the ESMs estimate for the net atmosphere-to-surface flux over that period is 5.0 [4.2, 5.5] Gt C yr$^{-1}$, consistent
with the GCB estimate (Tables 5 and 6).
The distributions based on the individual models and data products reveal substantial spread but converge near
the decadal means quoted in Tables 5 to 7. Sink estimates for $S_{OCEAN}$ and from inverse systems are mostly non-
Gaussian, while the ensemble of DGVMs appears more normally distributed justifying the use of a multi-model
mean and standard deviation for their errors in the budget. Noteworthy is that the tails of the distributions
provided by the land and ocean bottom-up estimates would not agree with the global constraint provided by the
fossil fuel emissions and the observed atmospheric $CO_2$ growth rate. This illustrates the power of the
atmospheric joint constraint from $G_{ATM}$ and the global $CO_2$ observation network it derives from.
**3.8.1.1   Net atmosphere-to-land fluxes**
The GCB net atmosphere-to-land fluxes ($S_{LAND} – E_{LUC}$), calculated as the difference between $S_{LAND}$ from the
DGVMs and $E_{LUC}$ from the bookkeeping models, amounts to a 2.1 ± 1.1 GtC yr$^{-1}$ sink during 2013-2022 (Table
5). Estimates of net atmosphere-to-land fluxes ($S_{LAND} – E_{LUC}$) from the DGVMs alone (1.7 ± 0.6 GtC yr$^{-1}$, Table
5, green symbol on Figure 12) are slightly lower, within the uncertainty of the GCB estimate and also with the
global carbon budget constraint from the ocean sink ($E_{FOS} – G_{ATM} – S_{OCEAN}$, 1.6 ± 0.6 GtC yr$^{-1}$; Table 7). For the
last decade (2013-2022), the inversions estimate the net atmosphere-to-land uptake to be 1.6 [0.5, 2.3] GtC yr$^{-1}$,
similar to the DGVMs estimates (purple symbol on Figure 12). The ESMs estimate for the net atmosphere-to-
land uptake during 2013-2022 is 2.4 [1.8, 3.3] GtC yr$^{-1}$, consistent with the GCB and DGVMs estimates of
$S_{LAND} – E_{LUC}$ (Figure 13 top row). The independent constraint based on atmospheric $O_2$ is significantly lower,
1.1 ± 1.3 GtC yr$^{-1}$, although its relatively high uncertainty range overlaps with the central estimates from other
approaches.
**3.8.1.2   Net atmosphere-to-ocean fluxes**
For the 2013-2022 period, the GOBMs (2.6 ± 0.4 GtC yr$^{-1}$) produce a lower estimate for the ocean sink than the
$f$CO$_2$-products (3.1 [2.6, 3.3] GtC yr$^{-1}$), which shows up in Figure 12 as separate peaks in the distribution from
the GOBMs (dark blue symbols) and from the $f$CO$_2$-products (light blue symbols). Atmospheric inversions (3.0
[2.4, 4.1] GtC yr$^{-1}$) suggest an ocean uptake more in line with the $f$CO$_2$-products for the recent decade (Table 7) ,
although the inversions range includes both the GOBMs and $f$CO$_2$-products estimates (Figure 13 top row). The
ESMs 2.6 [2.2, 3.4] GtC yr$^{-1}$ suggest a moderate estimate for the ocean carbon sink, comparable to the GOBMs
estimate with regard to mean and spread. Conversely, the independent constraint based on atmospheric $O_2$
suggests a larger ocean sink (3.3 ± 0.6 GtC yr$^{-1}$), more consistent with the $f$CO$_2$-products and atmospheric
inversions. We caution that the riverine transport of carbon taken up on land and outgassing from the ocean is a
substantial (0.65 ± 0.3 GtC yr$^{-1}$) and uncertain term (Crisp et al., 2022; Gruber et al., 2023; DeVries et al., 2023)
that separates the GOBMs, ESMs and oxygen-based estimates on the one hand from the $f$CO$_2$-products and
atmospheric inversions on the other hand. However, the high ocean sink estimate based on atmospheric oxygen



that is not subject to river flux adjustment, provides another line of evidence that most GOBMs and ESMs
underestimate the ocean sink.

### 3.8.2 Regional partitioning

Figure 13 shows the latitudinal partitioning of the global atmosphere-to-ocean ($S_{OCEAN}$), atmosphere-to-land
($S_{LAND} - E_{LUC}$), and their sum ($S_{OCEAN} + S_{LAND} - E_{LUC}$) according to the estimates from GOBMs and ocean
$f$CO$_2$-products ($S_{OCEAN}$), DGVMs ($S_{LAND} - E_{LUC}$), and from atmospheric inversions ($S_{OCEAN}$ and $S_{LAND} - E_{LUC}$).

#### 3.8.2.1 North

Despite being one of the most densely observed and studied regions of our globe, annual mean carbon sink
estimates in the northern extra-tropics (north of 30°N) continue to differ. The atmospheric inversions suggest an
atmosphere-to-surface sink ($S_{OCEAN} + S_{LAND} - E_{LUC}$) for 2013-2022 of 2.8[1.7 to 3.3] GtC yr$^{-1}$, which is higher
than the process models' estimate of 2.2 ± 0.4 GtC yr$^{-1}$ (Figure 13). The GOBMs (1.2 ± 0.2 GtC yr$^{-1}$), $f$CO$_2$-
products (1.3[1.2-1.4] GtC yr$^{-1}$), and inversion systems (1.2[0.7 to 1.4] GtC yr$^{-1}$) produce consistent estimates of
the ocean sink. Thus, the difference mainly arises from the net land flux ($S_{LAND} - E_{LUC}$) estimate, which is 1.0 ±
0.4 GtC yr$^{-1}$ in the DGVMs compared to 1.6[0.4 to 2.6] GtC yr$^{-1}$ in the atmospheric inversions (Figure 13,
second row). We note that the range among inversions driven by OCO-2 satellite data is smaller though (1.6 -
2.2 GtC yr$^{-1}$ N=6), supporting the notion that northern extra-tropics land uptake was larger than suggested by the
DGVMs at least in the 2015-2022 period covered by this data product.
Discrepancies in the northern land fluxes conforms with persistent issues surrounding the quantification of the
drivers of the global net land CO$_2$ flux (Arneth et al., 2017; Huntzinger et al., 2017; O'Sullivan et al., 2022) and
the distribution of atmosphere-to-land fluxes between the tropics and high northern latitudes (Baccini et al.,
2017; Schimel et al., 2015; Stephens et al., 2007; Ciais et al., 2019; Gaubert et al., 2019).
In the northern extra-tropics, the process models, inversions, and $f$CO$_2$-products consistently suggest that most
of the variability stems from the land (Figure 13). Inversions generally estimate similar interannual variations
(IAV) over land to DGVMs (0.28-0.35 vs 0.8-0.64 GtC yr$^{-1}$, averaged over 1990-2022), and they have higher
IAV in ocean fluxes (0.05-0.10 GtC yr$^{-1}$) relative to GOBMs (0.02-0.06 GtC yr$^{-1}$, Figure S2), and $f$CO$_2$-
products (0.03-0.10 GtC yr$^{-1}$).

#### 3.8.2.2 Tropics

In the tropics (30°S-30°N), both the atmospheric inversions and process models estimate a net carbon balance
($S_{OCEAN} + S_{LAND} - E_{LUC}$) that is close to neutral over the past decade. The GOBMs (-0.03 ± 0.24 GtC yr$^{-1}$), $f$CO$_2$-
products (0.2 [0.2, 0.3] GtC yr$^{-1}$), and inversion systems (-0.3 [-0.1, 0.8] GtC yr$^{-1}$) all indicate an approximately
neutral tropical ocean flux (see Figure S1 for spatial patterns). DGVMs indicate a net land sink ($S_{LAND} - E_{LUC}$) of
0.6 ±0.4 GtC yr$^{-1}$, whereas the inversion systems indicate a net land flux of 0.03 [-0.8, 1.1] GtC yr$^{-1}$, though with
high uncertainty (Figure 13, third row).



The tropical lands are the origin of most of the atmospheric $CO_2$ interannual variability (Ahlström et al., 2015),
consistently among the process models and inversions (Figure 13). The interannual variability in the tropics is
similar among the ocean $f$CO₂-products (0.07-0.16 GtC yr-1) and the GOBMs (0.07-0.16 GtC yr$^{-1}$,
Figure S2), which is the highest ocean sink variability of all regions. The DGVMs and inversions indicate that
atmosphere-to-land $CO_2$ fluxes are more variable than atmosphere-to-ocean $CO_2$ fluxes in the tropics, with
interannual variability of 0.35 to 1.61 and 0.77-0.92 GtC yr$^{-1}$ for DGVMs and inversions, respectively.

### 3.8.2.3 South

In the southern extra-tropics (south of 30°S), the atmospheric inversions suggest a net atmosphere-to-surface
sink ($S_{OCEAN}$+$S_{LAND}$-$E_{LUC}$) for 2013-2022 of 1.5 [1.2, 1.9] GtC yr⁻¹, slightly higher than the process models'
estimate of 1.5 ± 0.4 GtC yr⁻¹ (Figure 13). An approximately neutral net land flux ($S_{LAND}$-$E_{LUC}$) for the southern
extra-tropics is estimated by both the DGVMs (0.05 ± 0.07 GtC yr⁻¹) and the inversion systems (sink of 0.02 [-
0.2, 0.2] GtC yr⁻¹). This means nearly all carbon uptake is due to oceanic sinks south of 30°S. The Southern
Ocean flux in the $f$CO₂-products (1.6[ 1.3, 1.7 GtC] yr⁻¹) and inversion estimates (1.5 [1.3, 1.9] GtCyr-1) is
slightly higher than in the GOBMs (1.4 ± 0.3 GtC yr⁻¹) (Figure 13, bottom row). This discrepancy in the mean
flux is smaller this year than in previous releases due to the change in data set of the regional distribution of the
river flux adjustment applied to $f$CO₂-products and inverse systems to isolate the anthropogenic $S_{OCEAN}$ flux.
The data set used (Lacroix et al., 2020) has less river-induced carbon outgassing in the Southern Ocean than the
previously used data set (Aumont et al., 2001). Nevertheless, the time-series of atmospheric inversions and
$f$CO₂-products diverge from the GOBMs. A substantial overestimation of the trends in the $f$CO₂-products could
be explained by sparse and unevenly distributed observations, especially in wintertime (Figure S1; Hauck et al.,
2023; Gloege et al., 2021). Model biases may contribute as well, with biases in mode water formation,
stratification, and the chemical buffer capacity known to play a role in Earth System Models (Terhaar et al.,
2021, Bourgeois et al., 2022, Terhaar et al., 2022).
The interannual variability in the southern extra-tropics is low because of the dominance of ocean areas with
low variability compared to land areas. The split between land ($S_{LAND}$-$E_{LUC}$) and ocean ($S_{OCEAN}$) shows a
substantial contribution to variability in the south coming from the land, with no consistency between the
DGVMs and the inversions or among inversions. This is expected due to the difficulty of separating exactly the
land and oceanic fluxes when viewed from atmospheric observations alone. The $S_{OCEAN}$ interannual variability
was found to be higher in the $f$CO₂-products (0.04-0.18 GtC yr$^{-1}$) compared to GOBMs (0.03 to 0.06 GtC yr$^{-1}$)
in 1990-2022 (Figure S2). Model subsampling experiments recently illustrated that $f$CO₂-products may
overestimate decadal variability in the Southern Ocean carbon sink by 30% and the trend since 2000 by 50-
130% due to data sparsity, based on one and two $f$CO₂-products with strong variability (Gloege et al., 2021,
Hauck et al., 2023).

### 3.8.2.4 RECCAP2 regions

Aligning with the RECCAP-2 initiative (Ciais et al., 2022; Poulter et al., 2022; DeVries et al., 2023), we
provide an overview of $E_{LUC}$, $S_{LAND}$, Net land ($S_{LAND}$ - $E_{LUC}$), and $S_{OCEAN}$ fluxes for 10 land regions, and 5 ocean
regions, averaged over the period 2013-2022. The DGVMs and inversions suggest a positive net land sink in all





regions, except for South America and Africa, where the inversions indicate a small net source of respectively -
0.1 [-0.5, 0.3] GtC yr$^{-1}$ and -0.3 [-0.6, -0.1] GtC yr$^{-1}$, compared to a small sink of 0.1±0.3 GtC yr$^{-1}$ and
0.3±0.2 GtC yr$^{-1}$ for the DGVMs. However, for South America, there is substantial uncertainty in both products
(ensembles span zero). For the DGVMs, this is driven by uncertainty in both $S_{LAND}$ (0.6±0.5 GtC yr$^{-1}$) and $E_{LUC}$
(0.4±0.2 GtC yr$^{-1}$). The bookkeeping models also suggest an $E_{LUC}$ source of around 0.4 GtC yr$^{-1}$ in South
America and Africa, in line with the DGVMs estimates. Bookkeeping models and DGVMs similarly estimate a
loss of 0.4 GtC yr$^{-1}$ in Southeast Asia, with DGVMs suggesting a near neutral net land sink (0.03±0.12 GtC
yr$^{-1}$). This contrasts the inversion estimate of a 0.2 [-0.3,0.6] GtC yr$^{-1}$ sink, although the ensemble spread is
substantial. The inversions suggest the largest net land sinks are located in North America (0.5 [-0.1,0.8] GtC
yr$^{-1}$), Russia (0.7 [0.5,1.1] GtC yr$^{-1}$), and East Asia (0.3 [0.0,0.9] GtC yr$^{-1}$). This agrees well with the DGVMs
in North America (0.4±0.2 GtC yr$^{-1}$), which indicate a large natural land sink ($S_{LAND}$) of 0.6±0.2 GtC yr$^{-1}$,
being slightly reduced by land-use related carbon losses (0.2±0.1 GtC yr$^{-1}$). The DGVMs suggest a smaller net
land sink in Russia compared to inversions (0.4±0.2 GtC yr$^{-1}$), and a similar net sink in East Asia (0.2±0.1 GtC
yr$^{-1}$).
There is generally a higher level of agreement in regional $S_{OCEAN}$ estimates between the different data streams
(GOBMs, $f$CO$_2$-products and atmospheric inversions) on decadal scale, compared to the land flux estimates. All
data streams agree that the largest contribution to $S_{OCEAN}$ stems from the Southern Ocean, with important
contributions also from the vast ocean basins in the Atlantic and Pacific oceans. In the Southern Ocean, GOBMs
suggest a sink of 1.0±0.3 GtC yr$^{-1}$, in line with the $f$CO$_2$-products (1.1 [0.9,1.2] GtC yr$^{-1}$) and atmospheric
inversions (1.0 [0.8,1.4] GtC yr$^{-1}$). There is similar agreement in the Pacific ocean, with GOBMs, $f$CO$_2$-
products, and atmospheric inversions indicating a sink of 0.5±0.1 GtC yr$^{-1}$, 0.7 [0.5,0.9] GtC yr$^{-1}$, and 0.6
[0.2,1.0] GtC yr$^{-1}$, respectively. However, in the Atlantic ocean, GOBMs simulate a sink of 0.5±0.1 GtC yr$^{-1}$,
noticeably lower than both the $f$CO$_2$-products (0.8 [0.7,0.9] GtC yr$^{-1}$) and atmospheric inversions (0.8 [0.5,1.2]
GtC yr$^{-1}$). It is important to note the $f$CO$_2$-products and atmospheric inversions have a substantial and uncertain
river flux adjustment in the Atlantic ocean (0.3 GtC yr$^{-1}$) that also leads to a mean offset between GOBMs and
$f$CO$_2$-products/inversions in the latitude band of the tropics (Figure 13). The Indian Ocean due its smaller size
and the Arctic Ocean due to its size and sea-ice cover that prevents air-sea gas-exchange are responsible for
smaller but non negligible $S_{OCEAN}$ fluxes (Indian Ocean: (0.3 [0.2,0.4] GtC yr$^{-1}$, 0.3 [0.3,0.4] GtC yr$^{-1}$, and 0.4
[0.3,0.6] GtC yr$^{-1}$ for GOBMs, $f$CO$_2$-products, and atmospheric inversions, respectively, and Arctic Ocean: (0.1
[0.1,0.1] GtC yr$^{-1}$, 0.2 [0.2,0.2] GtC yr$^{-1}$, and 0.1 [0.1,0.1] GtC yr$^{-1}$ for GOBMs, $f$CO$_2$-products, and
atmospheric inversions, respectively). Note that the $S_{OCEAN}$ numbers presented here deviate from numbers
reported in RECCAP-2 where the net air-sea CO$_2$ flux is reported (i.e. without river flux adjustment for $f$CO$_2$-
products and inversions, and with river flux adjustment subtracted from GOBMs in most chapters, or comparing
unadjusted data sets with discussion of uncertain regional riverine fluxes as major uncertainty, e.g. Sarma et al.,
2023, DeVries et al., 2023).



### 3.8.2.5 Tropical vs northern land uptake

A continuing conundrum is the partitioning of the global atmosphere-land flux between the northern hemisphere land and the tropical land (Stephens et al., 2017; Pan et al., 2011; Gaubert et al., 2019). It is of importance because each region has its own history of land-use change, climate drivers, and impact of increasing atmospheric $CO_2$ and nitrogen deposition. Quantifying the magnitude of each sink is a prerequisite to understanding how each individual driver impacts the tropical and mid/high-latitude carbon balance.

We define the North-South (N-S) difference as net atmosphere-land flux north of 30°N minus the net atmosphere-land flux south of 30°N. For the inversions, the N-S difference ranges from -0.5 GtC yr$^{-1}$ to +3.0 GtC yr$^{-1}$ across this year's inversion ensemble, but with a clear cluster of solutions driven by the OCO-2 satellite product with a NH land sink of 1.6-2.2 GtC yr$^{-1}$, along with a tropical land flux of -0.6 to +0.2 GtC yr$^{-1}$, and a dipole between +1.4 and +2.8 GtC yr$^{-1}$ for the period 2015-2022. Whether this tighter clustering relative to the surface-observation based inversions is driven by (a) additional information on tropical fluxes delivered by tropical retrievals contained in OCO-2, (b) a tighter constraint on the NH land sink from that same product, or (c) a reduced sensitivity to vertical transport differences between models when using $CO_2$ column integrals, requires further investigation.

In the ensemble of DGVMs the N-S difference is $0.5 \pm 0.6$ GtC yr$^{-1}$, a much narrower range than the one from atmospheric inversions. Five DGVMs have a N-S difference larger than 1.0 GtC yr$^{-1}$, compared to only two from last year's ensemble. This is still only 25% of DGVMs, compared to most inversion systems simulating a difference at least this large. The smaller spread across DGVMs than across inversions is to be expected as there is no correlation between Northern and Tropical land sinks in the DGVMs as opposed to the inversions where the sum of the two regions being well-constrained by atmospheric observations leads to an anti-correlation between these two regions. This atmospheric N-S gradient could be used as an additional way to evaluate tropical and NH uptake in DGVMs, if their fluxes were combined with multiple transport models. Vice versa, the much smaller spread in the N-S difference between the DGVMs could help to scrutinise the inverse systems further. For example, a large northern land sink and a tropical land source in an inversion would suggest a large sensitivity to $CO_2$ fertilisation (the dominant factor driving the land sinks) for Northern ecosystems, which would be not mirrored by tropical ecosystems. Such a combination could be hard to reconcile with the process understanding gained from the DGVM ensembles and independent measurements (e.g. Free Air $CO_2$ Enrichment experiments).

### 3.8.3 Forest Fires in 2023

Fire emissions so far in 2023 have been above the average of recent decades, due to an extreme wildfire season in North America. Figure S9 shows global and regional emissions estimates for the period 1st Jan-30th September in each year 2003-2023. Estimates derive from two global fire emissions products: the global fire emissions database (GFED, version 4.1s; van der Werf et al., 2017), and; the global fire assimilation system (GFAS, operated by the Copernicus Atmosphere Service; Di Giuseppe et al., 2018). The two products estimate that global emissions from fires were 1.5-1.8 GtC yr$^{-1}$ during January-September 2023. These estimates are 13-



15% above the 2013-2022 average for the same months (1.3-1.6 GtC yr$^{-1}$) and 7-9% above the 2003-2022
average (1.4-1.6 GtC yr$^{-1}$).
The above-average global fire emissions during January-September 2023 have occurred despite below-average
fire emissions from major source regions. On average during 2013-2022, 72-79% of global fire emissions
through September occur in the tropics (0.9-1.3 GtC yr$^{-1}$) and around half of global fire emissions through
September occur in Africa (0.6-0.8 GtC yr$^{-1}$). This year, through September, fire emissions in the tropics (0.7-
0.9 GtC yr$^{-1}$) were 7-23% below the 2013-2022 average and fire emissions in Africa (0.5-0.7 GtC yr$^{-1}$) were 7-
17% below the 2013-2022 average.
In contrast, fire emissions from the Northern extra-tropics so far in 2023 have exceeded the values of all
previous years 2003-2022. Northern extra-tropical emissions during January-September 2023 (0.6-0.8 GtC yr$^{-1}$)
were 80-160% above the average for the same months in the past decade (0.3 GtC yr$^{-1}$ for both global fire
emissions products). Fire emissions in North America alone (0.5-0.7 GtC yr$^{-1}$) were 220-380% above the
average of the past decade (0.1 GtC yr$^{-1}$ for both products). In both products, North America was the only
RECCAP2 region with above-average fire C emissions for January-September in 2023.
While the fire emission fluxes presented above point towards a highly unusual Northern Hemisphere fire season
so far in 2023, we caution that the fluxes presented should not be compared directly with other fluxes of the
budget (e.g. $S_{LAND}$ or $E_{LUC}$) due to incompatibilities between the observable fire emission fluxes and what is
quantified in the $S_{LAND}$ and $E_{LUC}$ components of the budget. The fire emission estimates from global fire
products relate to all fire types that can be observed in Earth Observations (Giglio et al., 2018; Randerson et al.,
2012; Kaiser et al., 2012), including (i) fires occurring as part of natural disturbance-recovery cycles that would
also have occurred in the pre-industrial period (Yue et al., 2016; Keeley and Pausas, 2019; Zou et al., 2019), (ii)
fires occurring above and beyond natural disturbance-recovery cycle due to changes in climate, $CO_2$ and N
fertilisation and to an increased frequency of extreme drought and heatwave events (Abatzoglou et al., 2019;
Jones et al., 2022; Zheng et al., 2021; Burton et al., 2023), and (iii) fires occurring in relation to land use and
land use change, such as deforestation fires and agricultural fires (van der Werf et al., 2010; Magi et al., 2012).
In the context of the global carbon budget, only the portion of fire emissions associated with (ii) should be
included in the $S_{LAND}$ component, and fire emissions associated with (iii) should already be accounted for in the
$E_{LUC}$ component. Emissions associated with (i) should not be included in the global carbon budget. It is not
currently possible to derive specific estimates for fluxes (i), (ii), and (iii) using global fire emission products
such as GFED or GFAS. In addition, the fire emissions estimates from global fire emissions products represent
a gross flux of carbon to the atmosphere, whereas the $S_{LAND}$ component of the budget is a net flux that should
also include post-fire recovery fluxes. Even if emissions from fires of type (ii) could be separated from those of
type (i), these fluxes may be partially or wholly offset in subsequent years by post-fire fluxes as vegetation
recovers, sequestering carbon from the atmosphere to the terrestrial biosphere (Yue et al., 2016).



### 3.9 Closing the Global Carbon Cycle

#### 3.9.1 Partitioning of Cumulative Emissions and Sink Fluxes

The global carbon budget over the historical period (1850-2021) is shown in Figure 3.

Emissions during the period 1850-2022 amounted to $695 \pm 70$ GtC and were partitioned among the atmosphere $(280 \pm 5$ GtC; 40%), ocean $(180 \pm 35$ GtC; 26%), and land $(225 \pm 55$ GtC; 32%). The cumulative land sink is almost equal to the cumulative land-use emissions $(220 \pm 70$ GtC), making the global land nearly neutral over the whole 1850-2022 period.

The use of nearly independent estimates for the individual terms of the global carbon budget shows a cumulative budget imbalance of 15 GtC (2% of total emissions) during 1850-2022 (Figure 3, Table 8), which, if correct, suggests that emissions could be slightly too high by the same proportion (2%) or that the combined land and ocean sinks are slightly underestimated (by about 3%), although these are well within the uncertainty range of each component of the budget. Nevertheless, part of the imbalance could originate from the estimation of significant increase in $E_{FOS}$ and $E_{LUC}$ between the mid 1920s and the mid 1960s which is unmatched by a similar growth in atmospheric $CO_2$ concentration as recorded in ice cores (Figure 3). However, the known loss of additional sink capacity of 30-40 GtC (over the 1850-2020 period) due to reduced forest cover has not been accounted for in our method and would exacerbate the budget imbalance (see Section 2.10 and Supplement S.6.4).

For the more recent 1960-2022 period where direct atmospheric $CO_2$ measurements are available, total emissions $(E_{FOS} + E_{LUC})$ amounted to $485 \pm 50$ GtC, of which $395 \pm 20$ GtC (82%) were caused by fossil $CO_2$ emissions, and $90 \pm 45$ GtC (18%) by land-use change (Table 8). The total emissions were partitioned among the atmosphere $(215 \pm 5$ GtC; 44%), ocean $(125 \pm 25$ GtC; 25%), and the land $(150 \pm 35$ GtC; 31%), with a near zero (-5 GtC) unattributed budget imbalance. All components except land-use change emissions have significantly grown since 1960, with important interannual variability in the growth rate in atmospheric $CO_2$ concentration and in the land $CO_2$ sink (Figure 4), and some decadal variability in all terms (Table 7). Differences with previous budget releases are documented in Figure S5.

The global carbon budget averaged over the last decade (2013-2022) is shown in Figure 2, Figure 14 (right panel) and Table 7. For this period, 88% of the total emissions $(E_{FOS} + E_{LUC})$ were from fossil $CO_2$ emissions $(E_{FOS})$, and 12% from land-use change $(E_{LUC})$. The total emissions were partitioned among the atmosphere (47%), ocean (26%) and land (31%), with a small unattributed budget imbalance (~4%). For single years, the budget imbalance can be larger (Figure 4). For 2022, the combination of our estimated sources $(11.1 \pm 0.9$ GtC yr$^{-1})$ and sinks $(11.2 \pm 0.9$ GtC yr$^{-1})$ leads to a $B_{IM}$ of -0.09 GtC, suggesting a near closure of the global carbon budget, although there is relatively high uncertainty on $B_{IM}$ $(\pm 1.3$ GtC for 2022) as this is calculated as the residual of the five budget terms.





### 3.9.2    Trend and Variability in the Carbon Budget Imbalance

The carbon budget imbalance ($B_{IM}$; Eq. 1, Figure 4) quantifies the mismatch between the estimated total
emissions and the estimated changes in the atmosphere, land, and ocean reservoirs. The budget imbalance from
1960 to 2022 is very small (-3.0 GtC over the period, i.e. average of 0.05 GtC yr$^{-1}$) and shows no trend over the
full time series (Figure 4e). The process models (GOBMs and DGVMs) and data-products have been selected to
match observational constraints in the 1990s, but no further constraints have been applied to their representation
of trend and variability. Therefore, the near-zero mean and trend in the budget imbalance is seen as evidence of
a coherent community understanding of the emissions and their partitioning on those time scales (Figure 4).
However, the budget imbalance shows substantial variability of the order of ±1 GtC yr$^{-1}$, particularly over semi-
decadal time scales, although most of the variability is within the uncertainty of the estimates. The positive
carbon imbalance during the 1960s, and early 1990s, indicates that either the emissions were overestimated, or
the sinks were underestimated during these periods. The reverse is true for the 1970s, and to a lesser extent for
the 1980s and 2013-2022 period (Figure 4, Table 7).

We cannot attribute the cause of the variability in the budget imbalance with our analysis, we only note that the
budget imbalance is unlikely to be explained by errors or biases in the emissions alone because of its large semi-
decadal variability component, a variability that is atypical of emissions and has not changed in the past 60 years
despite a near tripling in emissions (Figure 4). Errors in $S_{LAND}$ and $S_{OCEAN}$ are more likely to be the main cause
for the budget imbalance, especially on interannual to semi-decadal timescales. For example, underestimation of
the $S_{LAND}$ by DGVMs has been reported following the eruption of Mount Pinatubo in 1991 possibly due to
missing responses to changes in diffuse radiation (Mercado et al., 2009). Although since GCB2021 we
accounted for aerosol effects on solar radiation quantity and quality (diffuse vs direct), most DGVMs only used
the former as input (i.e., total solar radiation) (Table S1). Thus, the ensemble mean may not capture the full
effects of volcanic eruptions, i.e. associated with high light scattering sulphate aerosols, on the land carbon sink
(O'Sullivan et al., 2021). DGVMs are suspected to overestimate the land sink in response to the wet decade of
the 1970s (Sitch et al., 2008). Quasi-decadal variability in the ocean sink has also been reported, with all
methods agreeing on a smaller than expected ocean $CO_2$ sink in the 1990s and a larger than expected sink in the
2000s (Figure 10; Landschützer et al., 2016, DeVries et al., 2019, Hauck et al., 2020, McKinley et al., 2020,
Gruber et al., 2023) and the climate-driven variability could be substantial but is not well constrained (DeVries
et al., 2023, Müller et al., 2023). Errors in sink estimates could also be driven by errors in the climatic forcing
data, particularly precipitation for $S_{LAND}$ and wind for $S_{OCEAN}$. Also, the $B_{IM}$ shows substantial departure from
zero on yearly time scales (Figure 4e), highlighting unresolved variability of the carbon cycle, likely in the land
sink ($S_{LAND}$), given its large year to year variability (Figure 4d and 8).

Both the budget imbalance ($B_{IM}$, Table 7) and the residual land sink from the global budget ($E_{FOS}+E_{LUC}-G_{ATM}-$
$S_{OCEAN}$, Table 5) include an error term due to the inconsistencies that arises from combining $E_{LUC}$ from
bookkeeping models with $S_{LAND}$ from DGVMs, most notably the loss of additional sink capacity (see Section
2.10 and Supplement S.6.4). Other differences include a better accounting of land use changes practices and
processes in bookkeeping models than in DGVMs, or the bookkeeping models error of having present-day
observed carbon densities fixed in the past. That the budget imbalance shows no clear trend towards larger



values over time is an indication that these inconsistencies probably play a minor role compared to other errors
in $S_{LAND}$ or $S_{OCEAN}$.
Although the budget imbalance is near zero for the recent decades, it could be due to a compensation of errors.
We cannot exclude an overestimation of $CO_2$ emissions, particularly from land-use change, given their large
uncertainty, as has been suggested elsewhere (Piao et al., 2018), combined with an underestimate of the sinks. A
larger DGVM ($S_{LAND}$-$E_{LUC}$) over the extra-tropics would reconcile model results with inversion estimates for
fluxes in the total land during the past decade (Figure 13; Table 5). Likewise, a larger $S_{OCEAN}$ is also possible
given the higher estimates from the $f$CO$_2$-products (see Section 3.6.2, Figure 10 and Figure 13), the
underestimation of interior ocean anthropogenic carbon accumulation in the GOBMs (Section 3.6.5), and the
recently suggested upward adjustments of the ocean carbon sink in Earth System Models (Terhaar et al., 2022),
and in $f$CO$_2$-products, here related to a potential temperature bias and skin effects (Watson et al., 2020; Dong et
al., 2022; Figure 10). If $S_{OCEAN}$ were to be based on $f$CO$_2$-products alone, with all $f$CO$_2$-products including this
adjustment, this would result in a 2013-2022 $S_{OCEAN}$ of 3.7 GtC yr$^{-1}$ (Dong et al., 2022) or >3.9 GtC yr$^{-1}$
(Watson et al., 2020), i.e., outside of the range supported by the atmospheric inversions and with an implied
negative $B_{IM}$ of more than -1 GtC yr$^{-1}$ indicating that a closure of the budget could only be achieved with either
anthropogenic emissions being significantly larger and/or the net land sink being substantially smaller than
estimated here. A recent model study suggests that the skin effect is smaller (about 0.1 GtC yr$^{-1}$ or 5%) due to
feedbacks with surface carbon concentration (Bellenger et al., 2023), which would nevertheless lead to a larger
$S_{OCEAN}$ even in the GOBMs. More integrated use of observations in the Global Carbon Budget, either on their
own or for further constraining model results, should help resolve some of the budget imbalance (Peters et al.,

1557     2017).

**4     Tracking progress towards mitigation targets**
The average growth in global fossil $CO_2$ emissions peaked at nearly +3% per year during the 2000s, driven by
the rapid growth in emissions in China. In the last decade, however, the global growth rate has slowly declined,
reaching a low +0.5% per year over 2013-2022. While this slowdown in global fossil $CO_2$ emissions growth is
welcome, global fossil $CO_2$ emissions continue to grow, far from the rapid emission decreases needed to be
consistent with the temperature goals of the Paris Agreement.
Since the 1990s, the average growth rate of fossil $CO_2$ emissions has continuously declined across the group of
developed countries of the Organisation for Economic Co-operation and Development (OECD), with emissions
peaking in around 2005 and now declining at around 1% yr$^{-1}$ (Le Quéré et al., 2021). In the decade 2013-2022,
territorial fossil $CO_2$ emissions decreased significantly (at the 95% confidence level) in 18 countries/economies
whose economies grew significantly (also at the 95% confidence level): Belgium, Brazil, Croatia, Czechia,
Denmark, Estonia, Finland, France, Germany, Greece, Hong Kong, Israel, Italy, Jamaica, Japan, Luxembourg,
Netherlands, Norway, Portugal, Romania, Slovenia, South Africa, Sweden, Switzerland, United Kingdom,
USA, Zimbabwe (updated from Le Quéré et al., 2019). Altogether, these 18 countries emitted 1.9 GtC yr$^{-1}$ (7.1
GtCO$_2$ yr$^{-1}$) on average over the last decade, about 20% of world $CO_2$ fossil emissions. Figure 16 shows that the
emission declines in the USA and the EU27 are primarily driven by slightly weaker economic growth in the last
decade compared to the 1990s, sustained declines in energy per GDP (though, weakening in the USA), and



sustained declines in $CO_2$ emissions per unit energy (decarbonisation) with a slight acceleration in the US in the
last decade.
In contrast, fossil $CO_2$ emissions continue to grow in non-OECD countries, although the growth rate has slowed
from almost 6% $yr^{-1}$ during the 2000s to less than 2% $yr^{-1}$ in the last decade. Representing 47% of non-OECD
emissions in 2022, a large part of this slowdown is due to China, which has seen emissions growth decline from
9% $yr^{-1}$ in the 2000s to 2.2% $yr^{-1}$ in the last decade. Excluding China, non-OECD emissions grew at 3.1% $yr^{-1}$ in
the 2000s compared to 1.5% $yr^{-1}$ in the last decade. Figure 16 shows that China has had weaker economic
growth in the 2000s compared to the 2010s and a higher decarbonisation rate from 2005 to 2015 comparable to
the highs in the 1990s, though the decarbonisation rate has slowed considerably since 2016. India and the rest of
the world have strong economic growth that is not offset by decarbonisation or declines in energy per GDP,
driving up fossil $CO_2$ emissions. Despite the high deployment of renewables in some countries (e.g., India),
fossil energy sources continue to grow to meet growing energy demand (Le Quéré et al., 2019).
Globally, fossil $CO_2$ emissions growth is slowing, and this is due in part to the emergence of climate policy
(Eskander and Fankhauser 2020; Le Quere et al 2019) and technological change, which is leading to a shift from
coal to gas and growth in renewable energies, and reduced expansion of coal capacity. At the aggregated global
level, decarbonisation shows a strong and growing signal in the last decade, with smaller contributions from
lower economic growth and declines in energy per GDP. Despite the slowing growth in global fossil $CO_2$
emissions, emissions are still growing, far from the reductions needed to meet the ambitious climate goals of the
UNFCCC Paris agreement.
This year we updated the remaining carbon budget (RCB) based on two studies, the IPCC AR6 (Canadell et al,
2021) as used in GCB2022, and a recent revision of the IPCC AR6 estimates (Forster et al 2023). We update the
RCB assessed by the IPCC AR6 (Canadell et al., 2021), accounting for the 2020 to 2023 estimated emissions
from fossil fuel combustion ($E_{FOS}$) and land use changes ($E_{LUC}$). From January 2024, the IPCC AR6 RCB (50%
likelihood) for limiting global warming to 1.5°C, 1.7°C and 2°C is estimated to amount to 95, 190, and 325 GtC
(340, 690, 1190 GtCO₂). The Forster et al. (2023) study proposed a significantly lower RCB than IPCC AR6,
with the largest reduction being due to an update of the climate emulator (MAGICC) used to estimate the
warming contribution of non-$CO_2$ agents, and to the warming (i.e. emissions) that occurred over the 2020-2022
period. We update the Forster et al., budget accounting for the 2023 estimated emissions from fossil fuel
combustion ($E_{FOS}$) and land use changes ($E_{LUC}$). From January 2024, the Forster et al., (2023) RCB (50%
likelihood) for limiting global warming to 1.5°C, 1.7°C and 2°C is estimated to amount to 55, 155, and 305 GtC
(210, 560, 1110 GtCO₂), significantly smaller than the updated IPCC AR6 estimate. Both the original IPCC
AR6 and Forster et al. (2023) estimates include an uncertainty due to the climate response to cumulative $CO_2$
emissions, which is reflected through the percent likelihood of exceeding the given temperature threshold, an
additional uncertainty of 220GtCO₂ due to alternative non-$CO_2$ emission scenarios, and other sources of
uncertainties (see Canadell et al., 2021). The two sets of estimates overlap when considering all uncertainties.
The IPCC AR6 estimates have the advantage of a consensus building approach, while the Forster et al. (2023)
estimates include significant update estimates but without the backing of the IPCC yet. Here, we take the
average of our update of both IPCC AR6 and Forster et al. (2023) estimates, giving a remaining carbon (50%
likelihood) for limiting global warming to 1.5°C, 1.7°C and 2°C of respectively 75, 175, and 315 GtC (275, 625,



GtCO$_2$) starting from January 2024. We emphasise the large uncertainties, particularly when close to the
global warming limit of 1.5°C. These 1.5°C, 1.7°C and 2°C average remaining carbon budgets correspond
respectively to about 7, 15 and 28 years from the beginning of 2024, at the 2023 level of total anthropogenic
CO$_2$ emissions. Reaching net-zero CO$_2$ emissions by 2050 entails cutting total anthropogenic CO$_2$ emissions by
about 0.4 GtC (1.5 GtCO$_2$) each year on average, comparable to the decrease in E$_{FOS}$ observed in 2020 during
the COVID-19 pandemic. However, this would lead to cumulative emissions over 2024-2050 of 150 GtC (550
GtCO$_2$), well above the remaining carbon budget of 75 GtC to limit global warming to 1.5°C, but still below the
remaining budget of 175 GtC to limit warming to 1.7°C (in phase with the "well below 2°C" ambition of the
Paris Agreement). Even reaching net zero CO$_2$ globally by 2040, which would require annual emissions cuts of
0.7 GtC (2.4 GtCO$_2$) on average, would still exceed the remaining carbon budget, with 95 GtC (350 GtCO$_2$)
cumulative emissions over 2024-2050, unless the global emissions trajectory becomes net negative (i.e. more
anthropogenic CO$_2$ sinks than emissions) after 2040.
**5     Discussion**
Each year when the global carbon budget is published, each flux component is updated for all previous years to
consider corrections that are the result of further scrutiny and verification of the underlying data in the primary
input data sets. Annual estimates may be updated with improvements in data quality and timeliness (e.g., to
eliminate the need for extrapolation of forcing data such as land-use). Of all terms in the global budget, only the
fossil CO$_2$ emissions and the growth rate in atmospheric CO$_2$ concentration are based primarily on empirical
inputs supporting annual estimates in this carbon budget. The carbon budget imbalance, yet an imperfect
measure, provides a strong indication of the limitations in observations, in understanding and representing
processes in models, and/or in the integration of the carbon budget components.
The persistent unexplained variability in the carbon budget imbalance limits our ability to verify reported
emissions (Peters et al., 2017) and suggests we do not yet have a complete understanding of the underlying
carbon cycle dynamics on annual to decadal timescales. Resolving most of this unexplained variability should
be possible through different and complementary approaches. First, as intended with our annual updates, the
imbalance as an error term should be reduced by improvements of individual components of the global carbon
budget that follow from improving the underlying data and statistics and by improving the models through the
resolution of some of the key uncertainties detailed in Table 10. Second, additional clues to the origin and
processes responsible for the variability in the budget imbalance could be obtained through a closer scrutiny of
carbon variability in light of other Earth system data (e.g., heat balance, water balance), and the use of a wider
range of biogeochemical observations to better understand the land-ocean partitioning of the carbon imbalance
such as the constraint from atmospheric oxygen included this year. Finally, additional information could also be
obtained through better inclusion of process knowledge at the regional level, and through the introduction of
inferred fluxes such as those based on satellite xCO$_2$ retrievals. The limit of the resolution of the carbon budget
imbalance is yet unclear, but most certainly not yet reached given the possibilities for improvements that lie
ahead.
Estimates of global fossil CO$_2$ emissions from different datasets are in relatively good agreement when the
different system boundaries of these datasets are considered (Andrew, 2020a). But while estimates of E$_{FOS}$ are

derived from reported activity data requiring much fewer complex transformations than some other components of the budget, uncertainties remain, and one reason for the apparently low variation between datasets is precisely the reliance on the same underlying reported energy data. The budget excludes some sources of fossil $CO_2$ emissions, which available evidence suggests are relatively small (<1%). We have added emissions from lime production in China and the US, but these are still absent in most other non-Annex I countries, and before 1990 in other Annex I countries.

Estimates of $E_{LUC}$ suffer from a range of intertwined issues, including the poor quality of historical land-cover and land-use change maps, the rudimentary representation of management processes in most models, and the confusion in methodologies and boundary conditions used across methods (e.g., Arneth et al., 2017; Pongratz et al., 2014, see also Supplement S.6.4 on the loss of sink capacity; Bastos et al., 2021). Uncertainties in current and historical carbon stocks in soils and vegetation also add uncertainty in the $E_{LUC}$ estimates. Unless a major effort to resolve these issues is made, little progress is expected in the resolution of $E_{LUC}$. This is particularly concerning given the growing importance of $E_{LUC}$ for climate mitigation strategies, and the large issues in the quantification of the cumulative emissions over the historical period that arise from large uncertainties in $E_{LUC}$.

By adding the DGVMs estimates of $CO_2$ fluxes due to environmental change from countries' managed forest areas (part of $S_{LAND}$ in this budget) to the budget $E_{LUC}$ estimate, we successfully reconciled the large gap between our $E_{LUC}$ estimate and the land use flux from NGHGIs using the approach described in Grassi et al. (2021) for future scenarios and in Grassi et al. (2023) using data from the Global Carbon Budget 2021. The updated data presented here can be used as potential adjustment in the policy context, e.g., to help assess the collective countries' progress towards the goal of the Paris Agreement and avoiding double-accounting for the sink in managed forests. In the absence of this adjustment, collective progress would hence appear better than it is (Grassi et al., 2021). The application of this adjustment is also recommended in the UNFCCC Synthesis report for the first Global Stocktake (UNFCCC, 2022) whenever a comparison between LULUCF fluxes reported by countries and the global emission estimates of the IPCC is conducted. However, this adjustment should be seen as a short-term and pragmatic fix based on existing data, rather than a definitive solution to bridge the differences between global models and national inventories. Additional steps are needed to understand and reconcile the remaining differences, some of which are relevant at the country level (Grassi, et al., 2023, Schwingshackl, et al., 2022).

The comparison of GOBMs, $f$CO$_2$-products, and inversions highlights substantial discrepancy in the temporal evolution of $S_{OCEAN}$ in the Southern Ocean and northern high-latitudes (Figure 13, Hauck et al., 2023) and in the mean $S_{OCEAN}$ in the tropics. A large part of the uncertainty in the mean fluxes stems from the regional distribution of the river flux adjustment term. The current distribution simulates the largest share of the outgassing to occur in the tropics (Lacroix et al., 2020) in contrast to the regional distribution previously used with the largest riverine outgassing flux south of 20°S (Aumont et al., 2001). The long-standing sparse data coverage of $f$CO$_2$ observations in the Southern compared to the Northern Hemisphere (e.g., Takahashi et al., 2009) continues to exist (Bakker et al., 2016, 2022, Figure S1) and to lead to substantially higher uncertainty in the $S_{OCEAN}$ estimate for the Southern Hemisphere (Watson et al., 2020, Gloege et al., 2021, Hauck et al., 2023). This discrepancy, which also hampers model improvement, points to the need for increased high-quality $f$CO$_2$ observations especially in the Southern Ocean. At the same time, model uncertainty is illustrated by the large





spread of individual GOBM estimates (indicated by shading in Figure 13) and highlights the need for model
improvement. The diverging trends in S$_{OCEAN}$ from different methods is a matter of concern. Recent and on-
going work suggests that the $f$CO$_2$-products may overestimate the trend (Hauck et al., 2023), though many
products remain to be tested, whereas evidence is accumulating that GOBMs likely underestimate the mean flux
(Section 3.6.2, Terhaar et al., 2022, DeVries et al., 2023, Müller et al., 2023). The independent constraint from
atmospheric oxygen measurements is consistent within errors with the relatively larger ocean sink in the $f$CO$_2$-
products. The assessment of the net land-atmosphere exchange from DGVMs and atmospheric inversions also
shows substantial discrepancy, particularly for the estimate of the net land flux over the northern extra-tropic.
This discrepancy highlights the difficulty to quantify complex processes (CO$_2$ fertilisation, nitrogen deposition
and fertilisers, climate change and variability, land management, etc.) that collectively determine the net land
CO$_2$ flux. Resolving the differences in the Northern Hemisphere land sink will require the consideration and
inclusion of larger volumes of observations.
We provide metrics for the evaluation of the ocean and land models and the atmospheric inversions (Figures B2
to B4, Table S10). These metrics expand the use of observations in the global carbon budget, helping 1) to
support improvements in the ocean and land carbon models that produce the sink estimates, and 2) to constrain
the representation of key underlying processes in the models and to allocate the regional partitioning of the CO$_2$
fluxes. The introduction of process-based metrics targeted to evaluate the simulation of S$_{OCEAN}$ in the ocean
biogeochemistry models is an important addition to the evaluation based on ocean carbon observations. This is
an initial step towards the introduction of a broader range of observations and more stringent model evaluation
that we hope will support continued improvements in the annual estimates of the global carbon budget.
We assessed before that a sustained decrease of –1% in global emissions could be detected at the 66%
likelihood level after a decade only (Peters et al., 2017). Similarly, a change in behaviour of the land and/or
ocean carbon sink would take as long to detect, and much longer if it emerges more slowly. To continue
reducing the carbon imbalance on annual to decadal time scales, regionalising the carbon budget, and integrating
multiple variables are powerful ways to shorten the detection limit and ensure the research community can
rapidly identify issues of concern in the evolution of the global carbon cycle under the current rapid and
unprecedented changing environmental conditions.
**6        Conclusions**
The estimation of global CO$_2$ emissions and sinks is a major effort by the carbon cycle research community that
requires a careful compilation and synthesis of measurements, statistical estimates, and model results. The
delivery of an annual carbon budget serves two purposes. First, there is a large demand for up-to-date
information on the state of the anthropogenic perturbation of the climate system and its underpinning causes. A
broad stakeholder community relies on the data sets associated with the annual carbon budget including
scientists, policy makers, businesses, journalists, and non-governmental organisations engaged in adapting to
and mitigating human-driven climate change. Second, over the last decades we have seen unprecedented
changes in the human and biophysical environments (e.g., changes in the growth of fossil fuel emissions, impact
of COVID-19 pandemic, Earth's warming, and strength of the carbon sinks), which call for frequent
assessments of the state of the planet, a better quantification of the causes of changes in the contemporary global



carbon cycle, and an improved capacity to anticipate its evolution in the future. Building this scientific
understanding to meet the extraordinary climate mitigation challenge requires frequent, robust, transparent, and
traceable data sets and methods that can be scrutinised and replicated. This paper via 'living data' helps to keep
track of new budget updates.
**Data availability**
The data presented here are made available in the belief that their wide dissemination will lead to greater
understanding and new scientific insights of how the carbon cycle works, how humans are altering it, and how
we can mitigate the resulting human-driven climate change. Full contact details and information on how to cite
the data shown here are given at the top of each page in the accompanying database and summarised in Table 2.
The accompanying database includes three Excel files organised in the following spreadsheets:
File Global_Carbon_Budget_2023v0.1.xlsx includes the following:
1.  Summary
2.  The global carbon budget (1959-2022);
3.  The historical global carbon budget (1750-2022);
4.  Global $CO_2$ emissions from fossil fuels and cement production by fuel type, and the per-capita emissions
1744        (1850-2022);

5.  $CO_2$ emissions from land-use change from the individual bookkeeping models (1959-2022);
6.  Ocean $CO_2$ sink from the individual global ocean biogeochemistry models and $f$$CO_2$-products (1959-
1747        2022);

7.  Terrestrial $CO_2$ sink from the individual DGVMs (1959-2022);
8.  Cement carbonation $CO_2$ sink (1959-2022).
File National_Fossil_Carbon_Emissions_2023v0.1.xlsx includes the following:
1.  Summary
2.  Territorial country $CO_2$ emissions from fossil fuels and cement production (1850-2022);
3.  Consumption country $CO_2$ emissions from fossil fuels and cement production and emissions transfer from
1754        the international trade of goods and services (1990-2020) using CDIAC/UNFCCC data as reference;

4.  Emissions transfers (Consumption minus territorial emissions; 1990-2020);
5.  Country definitions.
File National_LandUseChange_Carbon_Emissions_2023v0.1xlsx includes the following:



1. Summary
2. Territorial country $CO_2$ emissions from Land Use Change (1850-2022) from three bookkeeping models;
All three spreadsheets are published by the Integrated Carbon Observation System (ICOS) Carbon Portal and
are available at https://doi.org/10.18160/GCP-2023 (Friedlingstein et al., 2023). National emissions data are also
available on Zenodo (Andrew and Peters, 2022), from the Global Carbon Atlas
(http://www.globalcarbonatlas.org/, last access: 27 September 2023) and from Our World in Data
(https://ourworldindata.org/co2-emissions, last access: 27 September 2023).
**Author contributions**
PF, MO, MWJ, RMA, DCEB, JH, PL, CLQ, ITL, GPP, WP, JP, CSc, and SSi designed the study, conducted the
analysis, and wrote the paper with input from JGC, PCi and RBJ. RMA, GPP and JIK produced the fossil $CO_2$
emissions and their uncertainties and analysed the emissions data. MH and GMa provided fossil fuel emission
data. JP, TGa, CSc and RAH provided the bookkeeping land-use change emissions with synthesis by JP and
CSc. FJo provided peat drainage emission estimates. SSm and CMP provided the estimates of non-vegetation
CDR fluxes. LBo, MCh, ÖG, NG, TI, TJ, LR, JS, RS, and HiT provided an update of the global ocean
biogeochemical models, TTTC, DF, LG, YI, AJ, GMc, ChR, and JZ provided an update of the ocean $f$CO2-data
products, with synthesis on both streams by JH, PL and NMa. SRA, LBa, NRB, MB, MCr, KE, WE, RAF, TGk,
AK, NL, DRM, SN, AO, AMO, TO, MEP, DP, KP, GR, AJS, CSw, ST, BT, EvO, RW, and CWR provided
ocean fCO$_2$ measurements for the year 2022, with synthesis by DCEB and KMO. PA, DB, SF, JG, HJ, AKJ,
EK, DK, JK, GMe, LM, PM, MO, BP, TLS, QS, HTi, WY, XYua, XYue, and SZ provided an update of the
Dynamic Global Vegetation Models, with synthesis by SSi and MO. HL, RSA, MW, and PCa provided
estimates of land and ocean sinks from Earth System Models, as well as a projection of the atmospheric growth
rate for 2023. FC, ITL, NC, LF, ARJ, FJi, JL, ZJin, ZLiu, YN, CR, DY, and BZ provided an updated
atmospheric inversion, WP, FC, and ITL developed the protocol and produced the synthesis and evaluation of
the atmospheric inversions. RMA provided projections of the 2023 fossil emissions and atmospheric CO2
growth rate. PL provided the predictions of the 2023 ocean and land sinks. IBMB, LPC, GCH, KKG, TMR, and
GRvdW provided forcing data for land-use change. FT and GG provided data for the land-use change NGHGI
harmonisation. RK provided key atmospheric CO2 data. EJM and RFK provided the atmospheric oxygen
constraint on surface net carbon sinks. XL, PPT and MWJ provided the historical atmospheric CO2
concentration and growth rate. MO and NB produced the aerosol diffuse radiative forcing for the DGVMs. IH
provided the climate forcing data for the DGVMs. ER provided the evaluation of the DGVMs. MWJ provided



the emissions prior for use in the inversion systems. XD provided seasonal emissions data for most recent years
for the emission prior. PF, MO and MWJ coordinated the effort, revised all figures, tables, text and numbers to
ensure the update was clear from the 2022 edition and in line with the globalcarbonatlas.org.
**Competing interests.**
At least one of the (co-)authors is a member of the editorial board of Earth System Science Data
**Acknowledgements**
We thank all people and institutions who provided the data used in this global carbon budget 2023 and the Global
Carbon Project members for their input throughout the development of this publication. We thank Nigel Hawtin
for producing Figure 2 and Figure 15. We thank Alex Vermeulen and Hanna Ritchie for respectively hosting the
Global Carbon Budget datasets on the ICOS portal and the Our World in Data website. We thank Ian G. C. Ashton,
Sebastian Brune, Fatemeh Cheginig, Sam Ditkovsky, Christian Ethé, Amanda R. Fay, Lonneke Goddijn-Murphy,
T. Holding, Yawen Kong, Fabrice Lacroix, Yi Liu, Damian Loher, Naiqing Pan, Paridhi Rustogi, Shijie Shu, J.
D. Shutler, Richard Sims, Phillip Townsend, Jing Wang, Andrew J. Watson, and David K. Woolf for their
involvement in the development, use and analysis of the models and data-products used here. We thank Toste
Tanhua, Marcos Fontela, Claire Lo Monaco and Nicolas Metzl who contributed to the provision of surface ocean
$CO_2$ observations for the year 2022 (see Table S6). We also thank Stephen D. Jones of the Ocean Thematic Centre
of the EU Integrated Carbon Observation System (ICOS) Research Infrastructure, Eugene Burger of NOAA's
Pacific Marine Environmental Laboratory and Alex Kozyr of NOAA's National Centers for Environmental
Information, for their contribution to surface ocean $CO_2$ data and metadata management. This is PMEL
contribution 5550. We thank the scientists, institutions, and funding agencies responsible for the collection and
quality control of the data in SOCAT as well as the International Ocean Carbon Coordination Project (IOCCP),
the Surface Ocean Lower Atmosphere Study (SOLAS) and the Integrated Marine Biosphere Research (IMBeR)
program for their support. We thank data providers ObsPack GLOBALVIEWplus v8.0 and NRT v8.1 for
atmospheric $CO_2$ observations. We thank Fortunat Joos, Samar Khatiwala and Timothy DeVries for providing
historical atmospheric and ocean data. Ingrid T Luijkx and Wouter Peters thank the CarbonTracker Europe team
at Wageningen University, including Remco de Kok, Joram Hooghiem, Linda Kooijmans and Auke van der
Woude. Daniel Kennedy thanks all the scientists, software engineers, and administrators who contributed to the
development of CESM2. Josefine Ghattas thanks the whole ORCHIDEE group. Ian Harris thanks the Japan
Meteorological Agency (JMA) for producing the Japanese 55-year Reanalysis (JRA-55). Reinel Sospedra-



Alfonso thanks Barbara Winter, Woosung Lee, and William J. Merryfield for their contribution to the preparation
and production of CanESM5 runs. Patricia Cadule thanks Olivier Torres, Juliette Mignot, Didier Swingedouw,
and Laurent Bopp for contributions to the IPSL-CM6-LR-ESMCO2 simulations. Yosuke Niwa thanks CSIRO,
EC, EMPA, FMI, IPEN, JMA, LSCE, NCAR, NIES, NILU, NIWA, NOAA, SIO, and TU/NIPR for providing
data for NISMON-CO2. Zhe Jin thanks Xiangjun Tian, Yilong Wang, Hongqin Zhang, Min Zhao, Tao Wang,
Jinzhi Ding and Shilong Piao for their contributions to the GONGGA inversion system. Bo Zheng thanks Yawen
Kong for running the THU inversion system. Frédéric Chevallier thanks Zoé Lloret who maintained the
atmospheric transport model for the CAMS inversions. Frédéric Chevallier and Thi-Tuyet-Trang Chau thank
Marion Gehlen for her contribution to the CMEMS-LSCE-FFNNv2 product. Lian Fang thanks Paul I. Palmer and
acknowledges ongoing support from the National Centre for Earth Observation. Junjie Liu thanks the Jet
Propulsion Laboratory, California Institute of Technology. Zhiqiang Liu thanks Ning Zeng, Yun Liu, Eugenia
Kalnay, and Gassem Asrar for their contributions to the COLA system. Fei Jiang acknowledges ongoing support
from Frontiers Science Center for Critical Earth Material Cycling, Nanjing University. Andy Jacobson thanks the
team at NOAA GML, Boulder, Colorado, USA, who provided the CarbonTracker CT2022 and CT-NRT.v2023-
3 results from the website at http://carbontracker.noaa.gov. Meike Becker and Are Olsen thank Sparebanken Vest
/ Agenda Vestlandet for their support for the observations on the Statsraad Lehmkuhl. Margot Cronin thanks
Anthony English, Clynt Gregory and Gordon Furey (P&O Maritime Services), Tobias Steinhoff and Aodhan
Fitzgerald (Marine Institute) for their support. Wiley Evans and Katie Pocock thank the Tula Foundation for
funding support. Thanos Gkritzalis and the VLIZ ICOS team are thankful to the crew of the research vessel Simon
Stevin for all the support and help they provide. Data providers Nicolas Metzl and Claire LoMonaco thank the
French Institut National des Sciences de l'Univers (INSU), Institut Polaire français, Paul-Emile Victor(IPEV),
Observatoire des sciences de l'univers Ecce-Terra (OSU at Sorbonne Université), Institut de recherche français
sur les ressources marines (IFREMER), French Oceanographic Fleet (FOF) for the Marion Dufresne data set
(http://dx.doi.org/10.17600/18001858). Bronte Tilbrook and Erik van Ooijen thank Australia's Integrated Marine
Observing System (IMOS) for sourcing of $CO_2$ data. IMOS is enabled by the National Collaborative Research
Infrastructure Strategy (NCRIS). FAOSTAT is funded by FAO member states through their contributions to the
FAO Regular Programme, data contributions by national experts are greatly acknowledged. The views expressed
in this paper are the authors' only and do not necessarily reflect those of FAO. Finally, we thank all funders who
have supported the individual and joint contributions to this work (see details below), as well as the reviewers of
this manuscript and previous versions, and the many researchers who have provided feedback.



**Financial and computing support**
This research was supported by the following sources of funding: Integrated Marine Observing System (IMOS)
[Australia]; ICOS Flanders [Belgium]; Research Foundation Flanders (grant no. I001821N) [Belgium]; Tula
Foundation [Canada]; Chinese Academy of Science Project for Young Scientists in Basic Research (Grant No.
YSBR-037) [China]; National Key R&D Program of China (Grant No: 2020YFA0607504); National Natural
Science Foundation (grant no. 42141020, 42275128) [China]; National Natural Science Foundation (grant no.
42275128) [China]; National Natural Science Foundation (grant no. 41921005) [China]; Scientific Research Start-
up Funds (grant no. QD2021024C) from Tsinghua Shenzhen International Graduate School [China]; Second
Tibetan Plateau Scientific Expedition and Research Program (Grant: 2022QZKK0101) [China]; Young Elite
Scientists Sponsorship Program by CAST (grant no. YESS20200135) [China]; Copernicus Atmosphere
Monitoring Service, implemented by ECMWF (Grant: CAMS2 55) [EC]; Copernicus Marine Environment
Monitoring Service, implemented by MOi (Grant: CAMS2 55) [EC]; H2020 (Horizon 2020) 4C (grant no.
821003) [European Commission, EC]; H2020 ESM2025 – Earth System Models for the Future (grant no.
101003536) [European Commission, EC]; H2020 EuroSea (grant no. 862626) [EC]; H2020 GEORGE (grant no.
101094716) [EC]; H2020 JERICO-S3 (grant no. 871153) [EC]; ICOS France [France]; Institut de Recherche pour
le Développement (IRD) [France]; Federal Ministry of Education and Research (BMBF) (grant no. 03F0885AL1)
[Germany]; Federal Ministry of Education and Research, collaborative project C-SCOPE (Towards Marine
Carbon Observations 2.0: Socializing, COnnecting, Perfecting and Expanding, project no. 03F0877A) [Germany];
Helmholtz Association ATMO programme [Germany]; Helmholtz Association of German Research Centres
(project MOSES; Modular Observation Solutions for Earth Systems) [Germany]; ICOS (Integrated Carbon
Observation System) Germany [Germany]; Ludwig-Maximilians-University Munich, Department of Geography
[Germany]; Marine Institute [Ireland]; Arctic Challenge for Sustainability phase II project (ArCS-II; grant no. JP-
MXD1420318865) [Japan]; Environment Research and Technology Development Fund (grant no. JP-
MEERF21S20800) [Japan]; Fundamental Research Funds for the Central Universities (Grant No:
090414380031); Global Environmental Research Coordination System, Ministry of the Environment (grant no.
E2252) [Japan]; Japan Meteorological Agency [Japan]; Ministry of Education, Culture, Sports, Science and
Technology, MEXT program for the advanced studies of climate change projection (SENTAN) (grant numbers
JPMXD0722680395, JPMXD0722681344) [Japan]; Ministry of Environment, Environmental Restoration and
Conservation Agency, Environment Research and Technology Development Fund (grant no.
JPMEERF21S20810) [Japan]; Research Council of Norway (N-ICOS-2, grant no. 296012) [Norway]; Swiss



National Science Foundation (grant no. 200020-200511) [Switzerland]; National Centre for Atmospheric Science
[UK]; NERC (Natural Environment Research Council) Independent Research Fellowship (NE/V01417X/1) [UK];
NERC (NE/R016518/1) [UK]; Royal Society (grant no. RP\R1\191063) [UK]; UK Research and Innovation
(UKRI) for Horizon Europe (GreenFeedBack, grant no. 10040851) [UK]; NASA Carbon Monitoring System
program (80NSSC21K1059) [USA]; NASA OCO Science team program (80NM0018F0583) [USA]; National
Center for Atmospheric Research (NCAR) cooperative agreement (NSF No. 1852977) [USA]; National Oceanic
and Atmospheric Administration (NOAA) cooperative agreement (NA22OAR4320151) [USA]; National Science
Foundation (NSF- 831361857) [USA]; NOAA (NA20OAR4320278) [USA]; NOAA Cooperative Agreement,
Cooperative Institute for Climate, Ocean, & Ecosystem Studies (CIOCES; NA20OAR4320271) [USA]; NOAA
Cooperative Agreement, Cooperative Institute for Marine and Atmospheric Studies (CIMAS) / University of
Miami (NA20OAR4320472) [USA]; NOAA Global Ocean Monitoring and Observing Program (grant no.
100018302, 100007298, NA-03-AR4320179) [USA]; NOAA Ocean Acidification Program [USA]; National
Science Foundation (OPP-1922922) [USA].
We also acknowledge support from the following computing facilities: Adapter Allocation Scheme from the
National Computational Infrastructure (NCI) [Australia]; High-Performance Computing Center (HPCC) of
Nanjing University [China]; GENCI -TGCC (A0130102201, A0130106328, A0140107732 and A0130107403)
[France], CCRT awarded by CEA/DRF (CCRT2023-p24cheva) [France]; HPC cluster Aether at the University
of Bremen, financed by DFG within the scope of the Excellence Initiative [Germany], the State of Baden-
Württemberg, through bwHPC [Germany]; Earth Simulator (ES4) at JAMSTEC [Japan], JAMSTEC's Super
Computer system [Japan], NIES supercomputer system [Japan], NIES (SX-Aurora) and MRI (FUJITSU Server
PRIMERGY CX2550M5) [Japan]; ADA HPC cluster at the University of East Anglia [UK], UK CEDA
JASMIN supercomputer [UK]; Cheyenne NCAR HPC resources managed by CISL (doi:10.5065/D6RX99HX)
[USA].




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





**Tables**


| Unit 1 | Unit 2 | Conversion | Source |
|---|---|---|---|
| GtC (gigatonnes of carbon) | ppm (parts per million) (a) | 2.124 (b) | Ballantyne et al. (2012) |
| GtC (gigatonnes of carbon) | PgC (petagrams of carbon) | 1 | SI unit conversion |
| GtCO2 (gigatonnes of carbon dioxide) | GtC (gigatonnes of carbon) | 3.664 | 44.01/12.011 in mass equivalent |
| GtC (gigatonnes of carbon) | MtC (megatonnes of carbon) | 1000 | SI unit conversion |

(a) Measurements of atmospheric CO2 concentration have units of dry-air mole fraction. 'ppm' is an abbreviation for micromole/mol, dry air.

(b) The use of a factor of 2.124 assumes that all the atmosphere is well mixed within one year. In reality, only the troposphere is well mixed and the growth rate of CO2 concentration in the less well-mixed stratosphere is not measured by sites from the NOAA network. Using a factor of 2.124 makes the approximation that the growth rate of CO2 concentration in the stratosphere equals that of the troposphere on a yearly basis.

**Table 1.** Factors used to convert carbon in various units (by convention, Unit 1 = Unit 2 × conversion).





| Component | Primary reference |
|---|---|
| Global fossil CO2 emissions (EFOS), total and by fuel type | Updated from Andrew and Peters (2022) |
| National territorial fossil CO2 emissions (EFOS) | Gilfillan and Marland (2021), UNFCCC (2022) |
| National consumption-based fossil CO2 emissions (EFOS) by country (consumption) | Peters et al. (2011a) updated as described in this paper |
| Net land-use change flux (ELUC) | This paper (see Table 4 for individual model references). |
| Growth rate in atmospheric CO2 concentration (GATM) | Lan et al. (2023) |
| Ocean and land CO2 sinks (SOCEAN and SLAND) | This paper (see Table 4 for individual model and data products references). |

**Table 2.** How to cite the individual components of the global carbon budget presented here.




| Publication year | Fossil fuel emissions | | LUC emissions | Reservoirs | | | Other changes |
|---|---|---|---|---|---|---|---|
| | Global | Country (territorial) | | Atmosphere | Ocean | Land | |
| 2019<br><br>Friedlingstein et al. (2019) GCB2019 | Global emissions calculated as sum of all countries plus bunkers, rather than taken directly from CDIAC. | | Average of two bookkeeping models; use of 15 DGVMs | Use of three atmospheric inversions | Based on nine models | Based on 16 models | |
| 2020<br><br>Friedlingstein et al. (2020) GCB2020 | Cement carbonation now included in the EFOS estimate, reducing EFOS by about 0.2GtC yr-1 for the last decade | India's emissions from Andrew (2020: India); Corrections to Netherland Antilles and Aruba and Soviet emissions before 1950 as per Andrew (2020: CO2); China's coal emissions in 2019 derived from official statistics, emissions now shown for EU27 instead of EU28. Projection for 2020 based on assessment of four approaches. | Average of three bookkeeping models; use of 17 DGVMs. Estimate of gross land use sources and sinks provided | Use of six atmospheric inversions | Based on nine models. River flux revised and partitioned NH, Tropics, SH | Based on 17 models | |
| 2021<br><br>Friedlingstein et al. (2022a) GCB2021 | Projections are no longer an assessment of four approaches. | Official data included for a number of additional countries, new estimates for South Korea, added emissions from lime | ELUC estimate compared to the estimates adopted in national GHG inventories (NGHGI) | | Average of means of eight models and means of seven data-products. Current year prediction of SOCEAN using a feed-forward | Current year prediction of SLAND using a feed-forward neural network method | |



| | | | | neural network method | | |
|---|---|---|---|---|---|---|
| 2022<br><br>Friedlingstein et al. (2022) GCB2022 | | production in China. | ELUC provided at country level. Revised components decomposition of ELUC fluxes. Revision of LUC maps for Brazil. New datasets for peat drainage. | Use of nine atmospheric inversions | Average of means of ten models and means of seven data-products | Based on 16 models. Revision of LUC maps for Brazil. | |
| 2023<br><br>This study | | | Refined components decomposition of ELUC. Revision of LUC maps for Indonesia. Use of updated peat drainage estimates. | Use of 14 atmospheric inversions. Additional use of 4 Earth System Models to estimate current year CO2 | Additional use of 4 Earth System Models and atmospheric oxygen method to assess SOCEAN. Regional distribution of river flux adjustment revised. | Based on 20 models. Additional use of 4 Earth System Models and atmospheric oxygen method to assess the net atmosphere-land flux. | Inclusion of an estimate of Carbon Dioxide Removal (CDR) |

**Table 3.** Main methodological changes in the global carbon budget since 2019. Methodological changes
introduced in one year are kept for the following years unless noted. Empty cells mean there were no
methodological changes introduced that year. Table S8 lists methodological changes from the first global carbon
budget publication up to 2018.





| Model/data name | Reference | Change from Global Carbon Budget 2022 (Friedlingstein et al., 2022b) |
|---|---|---|
| ***Bookkeeping models for land-use change emissions*** | | |
| BLUE | Hansis et al. (2015) | No change to model, but simulations performed with LUH2-GCB2023 forcing. Update in added peat drainage emissions. |
| H&C2023 | Houghton and Castanho (2023) | H&C2023 replaces the formerly used H&N2017 model. Minor bug fix in fuel harvest estimates. Update in added peat drainage emissions. |
| OSCAR | Gasser et al. (2020) | No change to model, but land-use forcing changed to LUH2-GCB2023 and FRA2020 (extrapolated to 2022). Constraining based on GCB2022 data for SLAND over 1960-2021. Update in added peat drainage emissions. |
| ***Dynamic global vegetation models*** | | |
| CABLE-POP | Haverd et al. (2018) | Improved representation of nitrogen retranslocation and plant uptake, minor bug fixes, parameter changes |
| CLASSIC | Melton et al. (2020), Asaadi et al. (2018) | Bug fixes, correct allocation of leaves after summer solstice for latitudes higher than 45°N, improved phenology for several PFTs |
| CLM5.0 | Lawrence et al. (2019) | No change. |
| DLEM | Tian et al. (2011, 2015) | No change. |
| *EDv3* | Moorcroft et al. (2001), Ma et al. (2022) | New this year. |
| *ELM* | Yang et al.(2023), Burrows et al.(2020) | New this year. |
| IBIS | Yuan et al. (2014) | Changes in parameterisation and new module of soil nitrogen dynamics (Ma et al., 2022) |
| ISAM | Jain et al. (2013), Meiyappan et al. (2015), Shu et al. (2020) | Vertically resolved soil biogeochemistry (carbon and nitrogen) module, following Shu et al. (2020), |
| ISBA-CTRIP | Delire et al. (2020) | No change. |
| JSBACH | Mauritsen et al. (2019), Reick et al. (2021) | No change. |
| JULES-ES | Wiltshire et al. (2021), Sellar et al. (2019), Burton et al. (2019) | Minor bug fixes. (Using JULES v6.3, suite u-co002) |
| LPJ-GUESS | Smith et al. (2014) | Minor bug fixes. |

| | | |
|---|---|---|
| LPJml | Schaphoff et al., 2018, von Bloh et al., 2018, Lutz et al., 2019 (tillage), Heinke et al., 2023 (livestock grazing) | New this year. |
| LPJwsl | Poulter et al. (2011) (d) | No change. |
| LPX-Bern | Lienert and Joos (2018) | No change. |
| OCN | Zaehle and Friend (2010), Zaehle et al. (2011) | Minor bug fixes |
| ORCHIDEEv3 | Krinner et al. (2005), Zaehle and Friend (2010), Vuichard et al. (2019) | Small update for leaf senescence (ORCHIDEE - V3; revision 8119) |
| SDGVM | Woodward and Lomas (2004), Walker et al. (2017) | implement gross land-use transitions, to track carbon from wood & crop harvest, and to track primary & secondary vegetation. |
| VISIT | Ito and Inatomi (2012), Kato et al. (2013) | No change. |
| YIBs | Yue and Unger (2015) | Inclusion of process-based water cycle from Noah-MP (Niu et al., 2011) |
| ***Intermediate complexity land carbon cycle model*** | | |
| CARDAMOM | Bloom et al. (2016), Smallman et al. (2021) | New this year |
| ***Global ocean biogeochemistry models*** | | |
| NEMO3.6-PISCESv2-gas (CNRM) | Berthet et al. (2019), Séférian et al. (2019) | No change. |
| FESOM-2.1-REcoM2 | Gürses et al. (2023) | No change |
| NEMO-PISCES (IPSL) | Aumont et al. (2015) | No change. |
| MOM6-COBALT (Princeton) | Liao et al. (2020) | No change |
| MRI-ESM2-2 | Nakano et al. (2011) | The ocean model has been updated to MRI.COMv5 (Sakamoto et al. 2023). The distribution of background vertical diffusivity is changed to the one proposed by Kawasaki et al. (2021). Model was spup-up with a preindustrial xCO2 of 278 ppm. |
| MICOM-HAMOCC (NorESM-OCv1.2) | Schwinger et al. (2016) | No change. |
| NEMO-PlankTOM12 | Wright et al. (2021) | Minor bug fixes, switch to ERA5 forcing, salinity restoring |
| CESM-ETHZ | Doney et al. (2009) | Model was spup-up with a preindustrial xCO2 of 278 ppm. |



| MPIOM-HAMOCC6 | Lacroix et al. (2021) | No change. |
|---|---|---|
| ACCESS (CSIRO) | Law et al. (2017) | Minor bug fixes, extended spinup since last participation 2020. |
| **fCO2-products** | | |
| CMEMS-LSCE-FFNNv2 | Chau et al. (2022) | Update to SOCATv2023 measurements and time period 1985-2022. The mapping approach by Chau et al (2022) has been upgraded by increasing spatial resolution from 1° to 0.25°. |
| JMA-MLR | Iida et al. (2021) | Updated to SOCATv2023 |
| LDEO-HPD | Gloege et al. (2022), Bennington et al. (2022) | Updated with SOCATv2023. Updated with current GCB2023 models and extending back in time using Bennington et al. (2022) method. |
| MPI-SOMFFN | Landschützer et al. (2016) | update to SOCATv2023. Since GCB2022, fluxes cover open ocean and coastal domains as well as the Arctic Ocean extension. |
| NIES-ML3 | Zeng et al. (2022) | New this year |
| OS-ETHZ-GRaCER | Gregor et al. (2021) | Updated to SOCATv2023 |
| Jena-MLS | Rödenbeck et al. (2014, 2022) | update to SOCATv2023 measurements, time period extended to 1957-2022 |
| UOEx-Watson | Watson et al. (2020) | Updated to SOCAT v2023. fCO2(sw) corrected to CCI SST v2.1 (Merchant et al. 2019) instead of OI SST v2.1. Updated interpolation datasets to CCI SST v2.1, CMEMS SSS and MLD (Jean-Michel et al. 2021). Monthly cool skin difference calculated using NOAA COARE 3.5 (Edson et al. 2013). CO2 flux computed using FluxEngine (Holding et al., 2019; Shutler et al., 2016). |
| **Atmospheric inversions** | | |
| Jena CarboScope | Rödenbeck et al. (2003, 2018) | Extension to 2022, re-addition of a 2.5-year relaxation term. |
| CAMS | Chevallier et al. (2005), Remaud et al. (2018) | Increase of the 3D resolution (4.5 times more 3D cells than the previous submission); extension to year 2022; update of the prior fluxes. |
| CarbonTracker Europe (CTE) | van der Laan-Luijkx et al. (2017) | Extension to 2022, update of prior fluxes. |
| NISMON-CO2 | Niwa et al. (2020, 2022) | Prior terrestrial fluxes include minor fluxes (BVOC and CH4) in addition to GPP, RE and LUC. |
| CT-NOAA | Peters et al. (2005), Jacobson et al. (2023a, 2023b) | New this year. |
| CMS-Flux | Liu et al. (2021) | Update of OCO-2 observations and prior fluxes. |



| | | |
|---|---|---|
| CAMS-Satellite | Chevallier et al. (2005), Remaud et al. (2018) | Increase of the 3D resolution, extension to year 2022 and the first months of 2023; removal of the pre-OCO-2 period (2010-2014 with GOSAT); update of the prior fluxes. |
| GONGGA | Jin et al. (2023) | Update of OCO-2 observations and prior fluxes. |
| THU | Kong et al. (2022) | Updates to the OCO-2 product and the fossil fuel data. |
| COLA | Liu et al. (2022) | New this year. |
| GCASv2 | Jiang et al. (2021, 2022) | New this year. |
| UoE in-situ | Feng et al. (2009), Feng et al. (2016), Palmer et al. (2019) | Update of the inversion system by using new version of GEOS-Chem |
| IAPCAS | Feng et al. (2016), Yang et al. (2021) | New this year. |
| MIROC4-ACTM | Chandra et al. (2022) | New this year |
| ***Earth System Models*** | | |
| CanESM5 | Swart et al. (2019), Sospedra-Alfonso et al. (2021) | New this year. |
| IPSL-CM6a-CO2-LR | Boucher et al. (2020) | New this year. |
| MIROC-ES2L | Watanabe et al. (2020) | New this year. |
| MPI-ESM1-2-LR | Mauritsen et al. (2019), Li et al. (2023) | New this year. |


**Table 4.** References for the process models, bookkeeping models, ocean data products, and atmospheric
inversions. All models and products are updated with new data to the end of year 2022, and the atmospheric
forcing for the DGVMs has been updated as described in Section C.2.2 and C.4.1.





|  |  | 1960s | 1970s | 1980s | 1990s | 2000s | 2013-2022 | 2022 |
|---|---|---|---|---|---|---|---|---|
| Land-use change emissions (ELUC) | Bookkeeping (BK) Net flux (1a) | 1.5±0.7 | 1.3±0.7 | 1.4±0.7 | 1.6±0.7 | 1.4±0.7 | 1.3±0.7 | 1.2±0.7 |
|  | BK - deforestation (total) | 1.7 [1.3,2.1] | 1.6 [1.2,1.9] | 1.7 [1.3,2.1] | 1.9 [1.6,2.2] | 2 [1.6,2.4] | 1.9 [1.5,2.4] | 1.9 [1.4,2.5] |
|  | BK - forest regrowth (total) | -0.8 [-1.1,-0.6] | -0.9 [-1.1,-0.7] | -0.9 [-1.1,-0.7] | -1 [-1.2,-0.7] | -1.1 [-1.3,-0.8] | -1.3 [-1.5,-0.9] | -1.3 [-1.6,-1] |
|  | BK - other transitions | 0.4 [0.3,0.4] | 0.2 [0.1,0.3] | 0.2 [0.2,0.3] | 0.1 [0,0.2] | 0.1 [0,0.2] | 0.1 [0,0.3] | 0.1 [0,0.2] |
|  | BK - peat drainage & peat fires | 0.2 [0.1,0.2] | 0.2 [0.1,0.2] | 0.2 [0.2,0.3] | 0.3 [0.3,0.3] | 0.3 [0.2,0.3] | 0.3 [0.3,0.3] | 0.2 [0.2,0.3] |
|  | BK - wood harvest & forest management | 0.2 [-0.2,0.6] | 0.2 [-0.2,0.6] | 0.2 [-0.2,0.6] | 0.2 [-0.1,0.6] | 0.2 [-0.1,0.6] | 0.2 [0,0.6] | 0.2 [0,0.7] |
|  | DGVMs-net flux (1b) | 1.5±0.5 | 1.3±0.5 | 1.6±0.6 | 1.8±0.6 | 1.8±0.7 | 1.7±0.6 | 1.7±0.6 |
| Terrestrial sink (SLAND) | Residual sink from global budget $(E_{FOS}+E_{LUC}(1a)-G_{ATM}-S_{OCEAN})$ (2a) | 1.7±0.8 | 1.8±0.8 | 1.7±0.9 | 2.7±0.9 | 2.9±0.9 | 2.9±0.9 | 3.7±1 |
|  | DGVMs (2b) | 1.3±0.5 | 2±0.7 | 1.9±0.8 | 2.5±0.6 | 2.9±0.7 | 3.3±0.8 | 3.8±0.8 |
| Net land fluxes (SLAND-ELUC) | GCB2023 Budget (2b-1a) | -0.2±0.8 | 0.8±1 | 0.5±1 | 0.9±0.9 | 1.4±1 | 2.1±1.1 | 2.6±1.1 |
|  | Atmospheric $O_2$ | --- | --- | --- | 1.2±1 | 1.1±1.1 | 1.1±1.3 | - |
|  | DGVMs-net (2b-1b) | -0.2±0.4 | 0.7±0.7 | 0.3±0.6 | 0.7±0.5 | 1.1±0.4 | 1.7±0.6 | 2.1±0.6 |
|  | Inversions[+] | - [-,-] | - [-,-] | 0.5 [0.4,0.6] (2) | 0.9 [0.6,1.3] (3) | 1.3 [0.7,2] (4) | 1.6 [0.5,2.3] (8) | 2.7 [1.4-3.8] (13) |
|  | ESMs | --- | --- | 0.6 [0.1,1] | 1.7 [1.3,2] | 2 [1.4,2.7] | 2.4 [1.8,3.3] | 3.9 [2.8-5.5] |


[+]Estimates are adjusted for the pre-industrial influence of river fluxes, for the cement carbonation sink, and adjusted to common $E_{FOS}$ (Sect. 2.7). The ranges given include varying numbers (in parentheses) of inversions in each decade (Table S4).

**Table 5.** Comparison of results from the bookkeeping method and budget residuals with results from the DGVMs, as well as additional estimates from atmospheric oxygen, atmospheric inversions and Earth System Models (ESMs) for different periods, the last decade, and the last year available. All values are in GtCyr−1. See Figure 7 for explanation of the bookkeeping component fluxes. The DGVM uncertainties represent ±1σ of the decadal or annual (for 2022) estimates from the individual DGVMs: for the inverse systems the mean and range of available results is given. All values are rounded to the nearest 0.1 GtC and therefore columns do not necessarily add to zero.




| Product | 1960s | 1970s | 1980s | 1990s | 2000s | 2013-2022 | 2022 |
|---|---|---|---|---|---|---|---|
| $fCO_2$-products | --- | --- | --- | 2.3 [2,2.9] | 2.4 [2.2,2.7] | 3.1 [2.6,3.3] | 3.1 [2.5,3.3] |
| GOBMs | 1±0.3 | 1.2±0.3 | 1.7±0.3 | 2±0.3 | 2.1±0.4 | 2.6±0.4 | 2.5±0.4 |
| GCB2023 Budget | 1.1±0.4 | 1.4±0.4 | 1.9±0.4 | 2.1±0.4 | 2.3±0.4 | 2.8±0.4 | 2.8±0.4 |
| Atmospheric $O_2$ | --- | --- | --- | 2±0.7 | 2.6±0.6 | 3.3±0.6 | - |
| Inversions | - [-,-] | - [-,-] | 1.7 [1.6,1.8] (2) | 2.2 [1.9,2.5] (3) | 2.4 [1.8,3.1] (4) | 3 [2.4,4.1] (8) | 3 [2.2-4.2] (13) |
| ESMs | --- | --- | 1.6 [0.7,2.4] | 1.8 [1.1,2.5] | 2.1 [1.5,2.8] | 2.6 [2.2,3.4] | 2.7 [2.3-3.5] |


**Table 6:** Comparison of results for the ocean sink from the $f$CO2-products, from global ocean biogeochemistry
models (GOBMs), the best estimate for GCB2023 as calculated from fCO2-products and GOBMs that is used in
the budget Table 7, as well as additional estimates from atmospheric oxygen, atmospheric inversions and Earth
System Models (ESMs) for different periods, the last decade, and the last year available. All values are in
GtCyr−1. Uncertainties represent ±1σ of the estimates from the GOBMs (N>10) and range of ensemble
members is given for ensembles with N<10 ($f$CO2-products, inversions, ESMs). The uncertainty of the
GCB2023 budget estimate is based on expert judgement (Section 2 and Supplementary S1 to S4) and for
oxygen it is the standard deviation of a Monte Carlo ensemble (Section 2.8).



| | | 1960s | 1970s | 1980s | 1990s | 2000s | 2013-2022 | 2022 | 2023 (Projection) |
|---|---|---|---|---|---|---|---|---|---|
| Total emissions (EFOS + ELUC) | Fossil CO2 emissions (EFOS)* | 3±0.2 | 4.7±0.2 | 5.5±0.3 | 6.4±0.3 | 7.8±0.4 | 9.6±0.5 | 9.9±0.5 | 10±0.5 |
| | Land-use change emissions (ELUC) | 1.5±0.7 | 1.3±0.7 | 1.4±0.7 | 1.6±0.7 | 1.4±0.7 | 1.3±0.7 | 1.2±0.7 | 1.1±0.7 |
| | Total emissions | 4.6±0.7 | 6±0.7 | 6.9±0.8 | 7.9±0.8 | 9.2±0.8 | 10.9±0.8 | 11.1±0.9 | 11.2±0.9 |
| Partitioning | Growth rate in atmos CO2 (GATM) | 1.7±0.07 | 2.8±0.07 | 3.4±0.02 | 3.1±0.02 | 4±0.02 | 5.2±0.02 | 4.6±0.2 | 4±0.4 |
| | Ocean sink (SOCEAN) | 1.1±0.4 | 1.4±0.4 | 1.9±0.4 | 2.1±0.4 | 2.3±0.4 | 2.8±0.4 | 2.8±0.4 | 2.9±0.4 |
| | Terrestrial sink (SLAND) | 1.3±0.5 | 2±0.7 | 1.9±0.8 | 2.5±0.6 | 2.9±0.7 | 3.3±0.8 | 3.8±0.8 | 3±1 |
| Budget Imbalance | BIM=EFOS+ELUC-(GATM+SOCEAN+SLAND) | 0.4 | -0.2 | -0.2 | 0.2 | 0 | -0.4 | -0.1 | 1.2 |

*Fossil emissions excluding the cement carbonation sink amount to 3±0.2 GtC/yr, 4.7±0.2 GtC/yr, 5.5±0.3 GtC/yr, 6.4±0.3 GtC/yr, 7.9±0.4 GtC/yr, and 9.8±0.5 GtC/yr for the decades 1960s to 2010s respectively and to 10.2±0.5 GtC/yr for 2022, and 10.3±0.5 GtC/yr for 2023.


**Table 7:** Decadal mean in the five components of the anthropogenic CO2 budget for different periods, and last
year available. All values are in GtC yr-1, and uncertainties are reported as ±1σ. Fossil $CO_2$ emissions include
cement carbonation. The table also shows the budget imbalance ($B_{IM}$), which provides a measure of the
discrepancies among the nearly independent estimates. A positive imbalance means the emissions are
overestimated and/or the sinks are too small. All values are rounded to the nearest 0.1 GtC and therefore
columns do not necessarily add to zero.





| | | 1750-2022 | 1850-2014 | 1850-2022 | 1960-2022 | 1850-2023 |
|---|---|---|---|---|---|---|
| Emissions | Fossil CO2 emissions (EFOS) | 480±25 | 400±20 | 475±25 | 395±20 | 485±25 |
| | Land-use change emissions (ELUC) | 250±75 | 210±65 | 220±65 | 90±45 | 220±65 |
| | Total emissions | 730±80 | 610±65 | 695±70 | 485±50 | 705±70 |
| Partitioning | Growth rate in atmos CO2 (GATM) | 300±5 | 235±5 | 280±5 | 215±5 | 280±5 |
| | Ocean sink (SOCEAN) | 190±40 | 155±30 | 180±35 | 125±25 | 180±35 |
| | Terrestrial sink (SLAND) | 245±60 | 200±50 | 225±55 | 150±35 | 225±55 |
| Budget imbalance | BIM=EFOS+ELUC-(GATM+SOCEAN+SLAND) | -5 | 20 | 15 | -5 | 15 |


**Table 8.** Cumulative $CO_2$ for different time periods in gigatonnes of carbon (GtC). Fossil $CO_2$ emissions
include cement carbonation. The budget imbalance ($B_{IM}$) provides a measure of the discrepancies among the
nearly independent estimates. All values are rounded to the nearest 5 GtC and therefore columns do not
necessarily add to zero. Uncertainties are reported as follows: $E_{FOS}$ is 5% of cumulative emissions; $E_{LUC}$ prior to
1959 is $1\sigma$ spread from the DGVMs, $E_{LUC}$ post-1959 is 0.7*number of years (where 0.7 GtC/yr is the
uncertainty on the annual $E_{LUC}$ flux estimate); $G_{ATM}$ uncertainty is held constant at 5 GtC for all time periods;
$S_{OCEAN}$ uncertainty is 20% of the cumulative sink (20% relates to the annual uncertainty of 0.4 GtC/yr, which is
~20% of the current ocean sink); and $S_{LAND}$ is the $1\sigma$ spread from the DGVMs estimates.




|  | 2003-2012 | 2013-2022 |
|---|---|---|
| ELUC from bookkeeping estimates (from Table 5) | 1.4 | 1.3 |
| SLAND on non-intact forest from DGVMs | 1.9 | 2.0 |
| ELUC subtract SLAND on non-intact forests | -0.5 | -0.8 |
| National Greenhouse Gas Inventories | -0.4 | -0.7 |

**Table 9:** Translation of global carbon cycle models' land flux definitions to the definition of the LULUCF net
flux used in national Greenhouse Gas Inventories reported to UNFCCC. See Sec. C.2.3 and Table S9 for detail
on methodology and comparison to other datasets. Units are GtC yr-1.




| Source of uncertainty | Time scale (years) | Location | Evidence |
|---|---|---|---|
| **Fossil CO2 emissions (EFOS; Section 2.1)** | | | |
| energy statistics | annual to decadal | global, but mainly China & major developing countries | (Korsbakken et al., 2016, Guan et al., 2012) |
| carbon content of coal | annual to decadal | global, but mainly China & major developing countries | (Liu et al., 2015) |
| system boundary | annual to decadal | all countries | (Andrew, 2020a) |
| **Net land-use change flux (ELUC; section 2.2)** | | | |
| land-cover and land-use change statistics | continuous | global; in particular tropics | (Houghton et al., 2012, Gasser et al., 2020, Ganzenmüller et al., 2022, Yu et al. 2022) |
| sub-grid-scale transitions | annual to decadal | global | (Wilkenskjeld et al., 2014) |
| vegetation biomass | annual to decadal | global; in particular tropics | (Houghton et al., 2012, Bastos et al., 2021) |
| forest degradation (fire, selective logging) | annual to decadal | tropics | (Aragão et al., 2018, Qin et al., 2021) |
| wood and crop harvest | annual to decadal | global; SE Asia | (Arneth et al., 2017, Erb et al., 2018) |
| peat burning | multi-decadal trend | global | (van der Werf et al., 2010, 2017) |
| loss of additional sink capacity | multi-decadal trend | global | (Pongratz et al, 2014, Gasser et al, 2020; Obermeier et al., 2021) |
| **Atmospheric growth rate (GATM; section 2.4)** no demonstrated uncertainties larger than ±0.3 GtC yr-1 . The uncertainties in GATM have been estimated as ±0.2 GtC yr-1, although the conversion of the growth rate into a global annual flux assuming instantaneous mixing throughout the atmosphere introduces additional errors that have not yet been quantified. | | | |
| **Ocean sink (SOCEAN; section 2.5)** | | | |
| sparsity in surface fCO2 observations | mean, decadal variability and trend | global, in particular southern hemisphere | (Gloege et al., 2021, Denvil-Sommer et al., 2021, Hauck et al., 2023) |



| riverine carbon outgassing and its anthropogenic perturbation | annual to decadal | global, in particular partitioning between Tropics and South | (Aumont et al., 2001, Lacroix et al., 2020, Cris et al., 2022) |
|---|---|---|---|
| Models underestimate interior ocean anthropogenic carbon storage | annual to decadal | global | (Friedlingstein et al., 2021, this study, DeVries et al., 2023, see also Terhaar et al., 2022) |
| near-surface temperature and salinity gradients | mean on all time-scales | global | (Watson et al., 2020, Dong et al., 2022, Bellenger et al., 2023) |
| Land sink (SLAND; section 2.6) | | | |
| strength of CO2 fertilisation | multi-decadal trend | global | (Wenzel et al., 2016; Walker et al., 2021) |
| response to variability in temperature and rainfall | annual to decadal | global; in particular tropics | (Cox et al., 2013; Jung et al., 2017; Humphrey et al., 2018; 2021) |
| nutrient limitation and supply | annual to decadal | global | (Zaehle et al., 2014) |
| carbon allocation and tissue turnover rates | annual to decadal | global | (De Kauwe et al., 2014; O'Sullivan et al., 2022) |
| tree mortality | annual | global in particular tropics | (Hubau et al., 2021; Brienen et al., 2020) |
| response to diffuse radiation | annual | global | (Mercado et al., 2009; O'Sullivan et al., 2021) |


**Table 10.** Major known sources of uncertainties in each component of the Global Carbon Budget, defined as
input data or processes that have a demonstrated effect of at least ±0.3 GtC yr-1.




**Figures and Captions**

**Atmospheric CO$_2$ Concentration**

— NOAA/ESRL (Dlugokencky and Tans, 2023)
— Scripps Institution of Oceanography (Keeling et al., 1976)


**Figure 1.** Surface average atmospheric CO$_2$ concentration (ppm). Since 1980, monthly data are from
NOAA/GML (Lan et al., 2023) and are based on an average of direct atmospheric CO$_2$ measurements from
multiple stations in the marine boundary layer (Masarie and Tans, 1995). The 1958-1979 monthly data are from
the Scripps Institution of Oceanography, based on an average of direct atmospheric CO$_2$ measurements from the
Mauna Loa and South Pole stations (Keeling et al., 1976). To account for the difference of mean CO$_2$ and
seasonality between the NOAA/GML and the Scripps station networks used here, the Scripps surface average
(from two stations) was de-seasonalised and adjusted to match the NOAA/GML surface average (from multiple
stations) by adding the mean difference of 0.667 ppm, calculated here from overlapping data during 1980-2012.


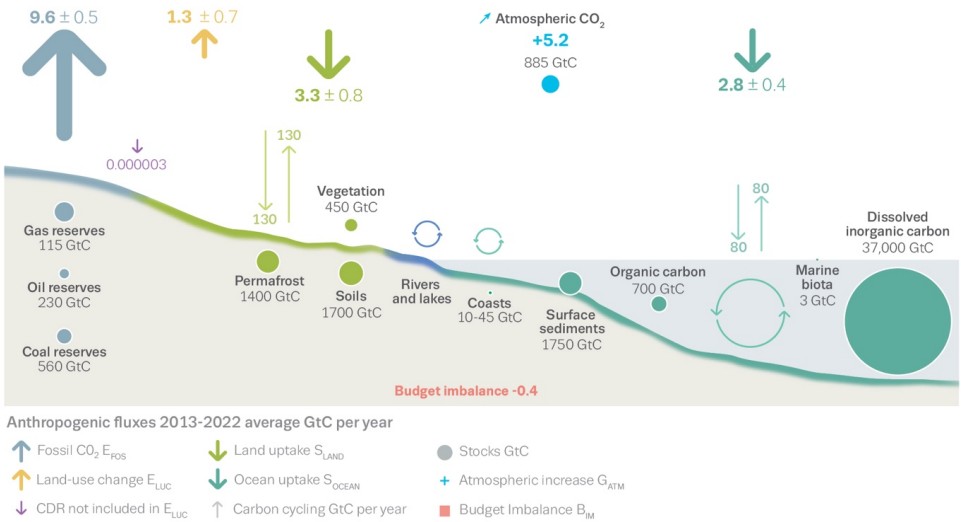



**Figure 2.** Schematic representation of the overall perturbation of the global carbon cycle caused by anthropogenic activities, averaged globally for the decade 2013-2022. See legends for the corresponding arrows and units. The uncertainty in the atmospheric $CO_2$ growth rate is very small ($\pm 0.02$ GtC yr-1) and is neglected for the figure. The anthropogenic perturbation occurs on top of an active carbon cycle, with fluxes and stocks represented in the background and taken from Canadell et al. (2021) for all numbers, except for the carbon stocks in coasts which is from a literature review of coastal marine sediments (Price and Warren, 2016).

3149

3150

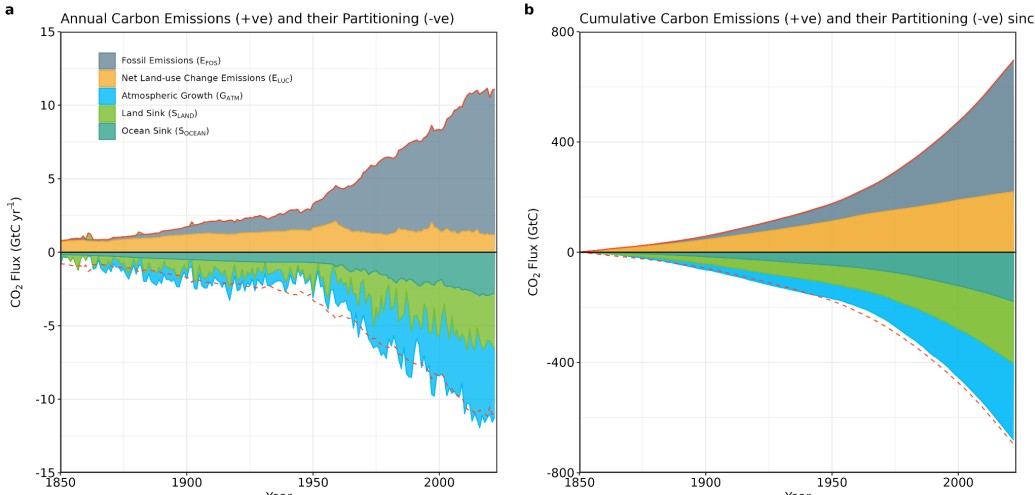

**Figure 3.** Combined components of the global carbon budget as a function of time, for fossil CO2 emissions

(EFOS, including a small sink from cement carbonation; grey) and emissions from land-use change (ELUC;

brown), as well as their partitioning among the atmosphere (GATM; cyan), ocean (SOCEAN; blue), and land

(SLAND; green). Panel (a) shows annual estimates of each flux and panel (b) the cumulative flux (the sum of all

prior annual fluxes) since the year 1850. The partitioning is based on nearly independent estimates from

observations (for GATM) and from process model ensembles constrained by data (for SOCEAN and SLAND)

and does not exactly add up to the sum of the emissions, resulting in a budget imbalance (BIM) which is

represented by the difference between the bottom red line (mirroring total emissions) and the sum of carbon

fluxes in the ocean, land, and atmosphere reservoirs. All data are in GtC yr-1 (panel a) and GtC (panel b). The

EFOS estimate is based on a mosaic of different datasets, and has an uncertainty of ±5% (±1σ). The ELUC

estimate is from three bookkeeping models (Table 4) with uncertainty of ±0.7 GtC yr-1. The GATM estimates

prior to 1959 are from Joos and Spahni (2008) with uncertainties equivalent to about ±0.1-0.15 GtC yr-1 and

from Lan et al. (2023) since 1959 with uncertainties of about +-0.07 GtC yr-1 during 1959-1979 and ±0.02 GtC

yr-1 since 1980. The SOCEAN estimate is the average from Khatiwala et al. (2013) and DeVries (2014) with

uncertainty of about ±30% prior to 1959, and the average of an ensemble of models and an ensemble of fCO2-

products (Table 4) with uncertainties of about ±0.4 GtC yr-1 since 1959. The SLAND estimate is the average of

an ensemble of models (Table 4) with uncertainties of about ±1 GtC yr-1. See the text for more details of each

component and their uncertainties.

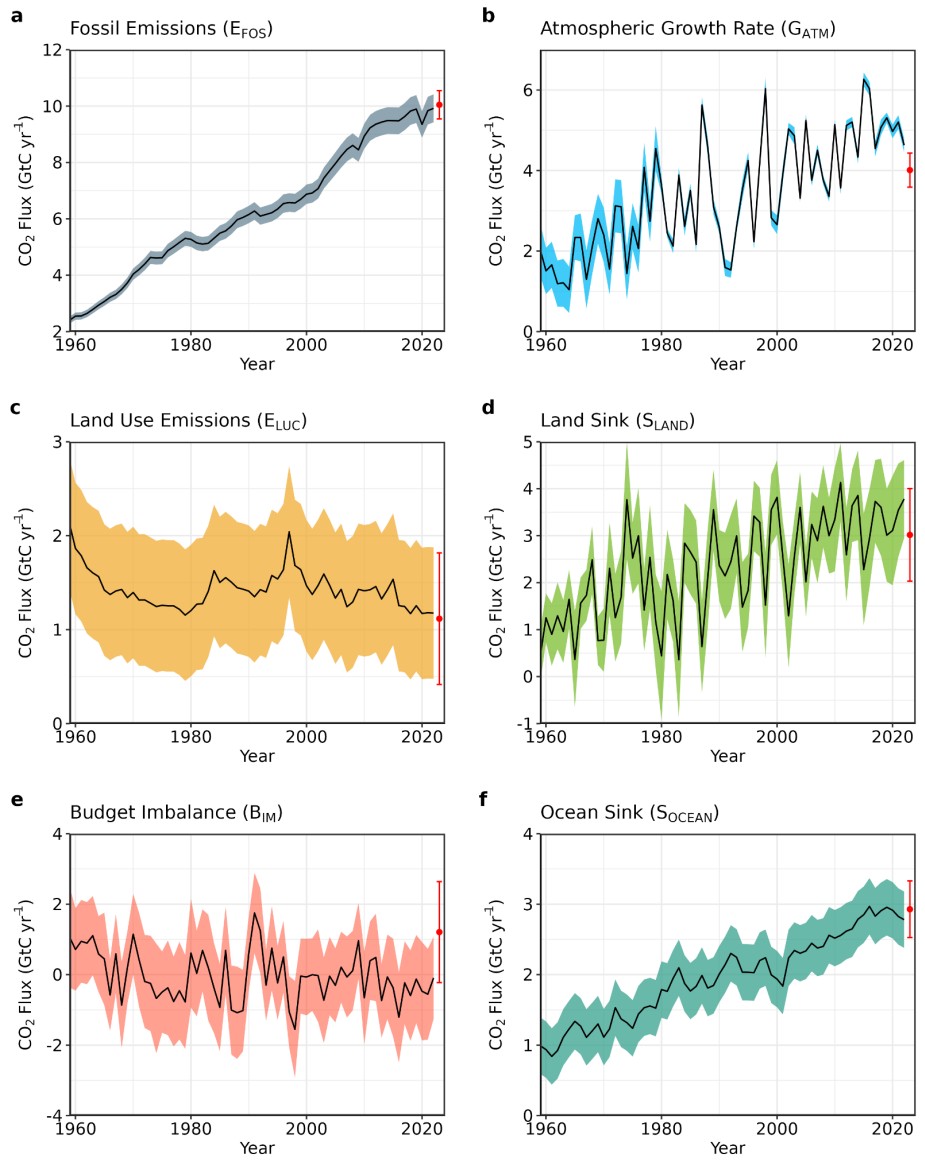

3171

**Figure 4.** Components of the global carbon budget and their uncertainties as a function of time, presented
individually for (a) fossil $CO_2$ and cement carbonation emissions ($E_{FOS}$), (b) growth rate in atmospheric $CO_2$
concentration ($G_{ATM}$), (c) emissions from land-use change ($E_{LUC}$), (d) the land $CO_2$ sink ($S_{LAND}$), (e) the ocean
$CO_2$ sink ($S_{OCEAN}$), (f) the budget imbalance that is not accounted for by the other terms. Positive values of
$S_{LAND}$ and $S_{OCEAN}$ represent a flux from the atmosphere to land or the ocean. All data are in GtC yr$^{-1}$ with the
uncertainty bounds representing ±1 standard deviation in shaded colour. Data sources are as in Figure 3. The red
dots indicate our projections for the year 2023 and the red error bars the uncertainty in the projections (see
methods).





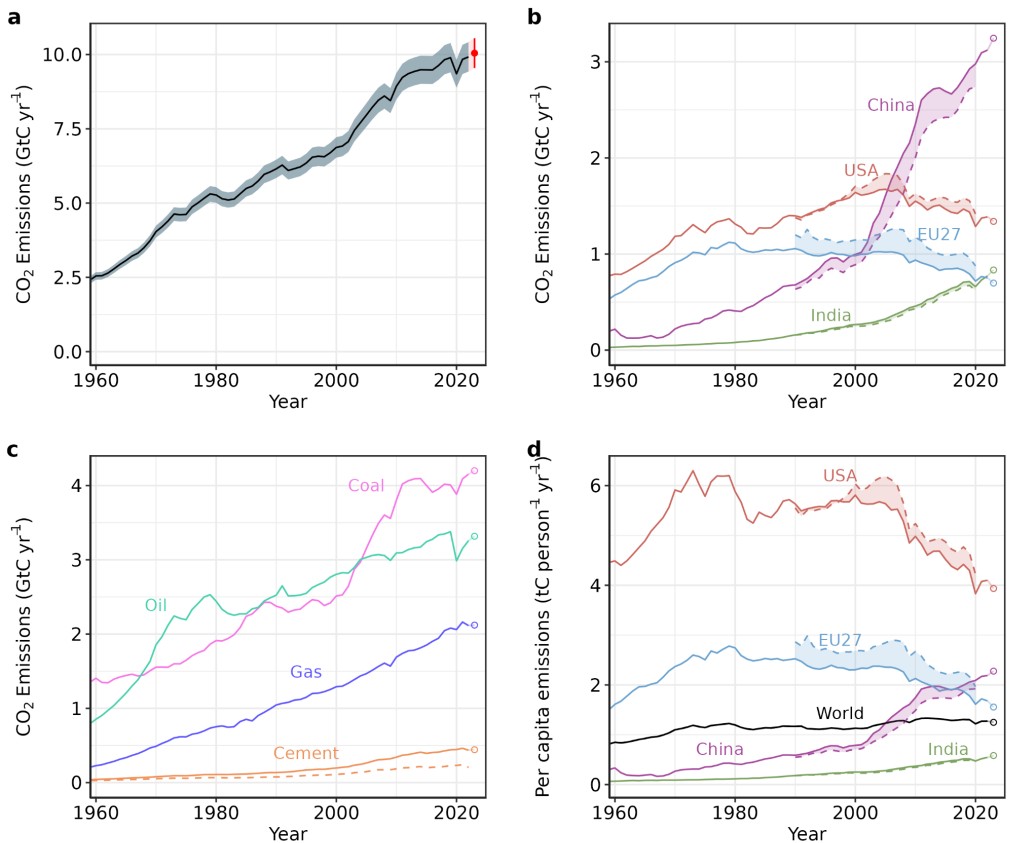

**Figure 5.** Fossil CO₂ emissions for (a) the globe, including an uncertainty of ± 5% (grey shading) and a projection through the year 2023 (red dot and uncertainty range), (b) territorial (solid lines) and consumption (dashed lines) emissions for the top three country emitters (USA, China, India) and for the European Union (EU27), (c) global emissions by fuel type, including coal, oil, gas, and cement, and cement minus cement carbonation (dashed), and (d) per-capita emissions the world and for the large emitters as in panel (b). Territorial emissions are primarily from a draft update of Gilfillan and Marland (2021) except for national data for Annex I countries for 1990-2021, which are reported to the UNFCCC as detailed in the text, as well as some improvements in individual countries, and extrapolated forward to 2022 using data from Energy Institute. Consumption-based emissions are updated from Peters et al. (2011a). See Section 2.1 and Supplement S.1 for details of the calculations and data sources.



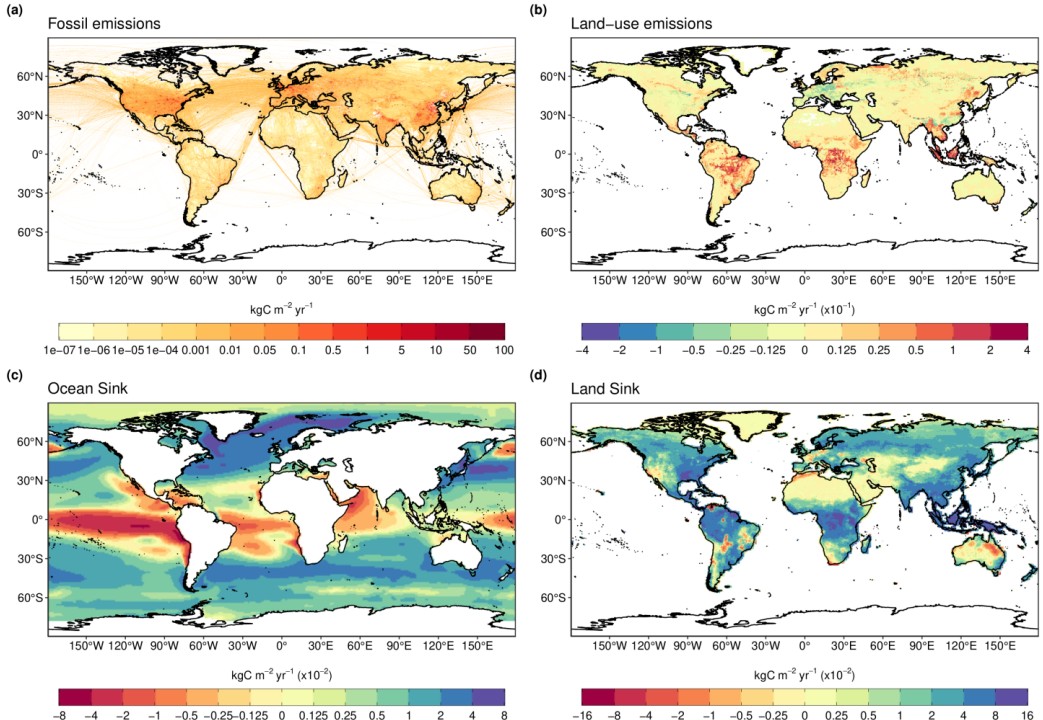

3193
3194
**Figure 6.** The 2013-2022 decadal mean components of the global carbon budget, presented for (a) fossil $CO_2$
emissions ($E_{FOS}$), (b) land-use change emissions ($E_{LUC}$), (c) the ocean $CO_2$ sink ($S_{OCEAN}$), and (d) the land $CO_2$
sink ($S_{LAND}$). Positive values for $E_{FOS}$ and $E_{LUC}$ represent a flux to the atmosphere, whereas positive values of
$S_{OCEAN}$ and $S_{LAND}$ represent a flux from the atmosphere to the ocean or the land (carbon sink). In all panels,
yellow/red colours represent a source (flux from the land/ocean to the atmosphere), green/blue colours represent
a sink (flux from the atmosphere into the land/ocean). All units are in kgC m$^{-2}$ yr$^{-1}$. Note the different scales in
each panel. $E_{FOS}$ data shown is from GCP-GridFEDv2023.1. The $E_{LUC}$ map shows the average $E_{LUC}$ from the
three bookkeeping models plus emissions from peat drainage and peat fires. Gridded $E_{LUC}$ estimates for
H&C2023 and OSCAR are derived by spatially distributing their national data based on the spatial patterns of
BLUE gross fluxes in each country (see Schwingshackl et al., 2022, for more details about the methodology).
$S_{OCEAN}$ data shown is the average of GOBMs and data-products means, using GOBMs simulation A, no
adjustment for bias and drift applied to the gridded fields (see Section 2.5). $S_{LAND}$ data shown is the average of
the DGVMs for simulation S2 (see Section 2.6).

3208
3209

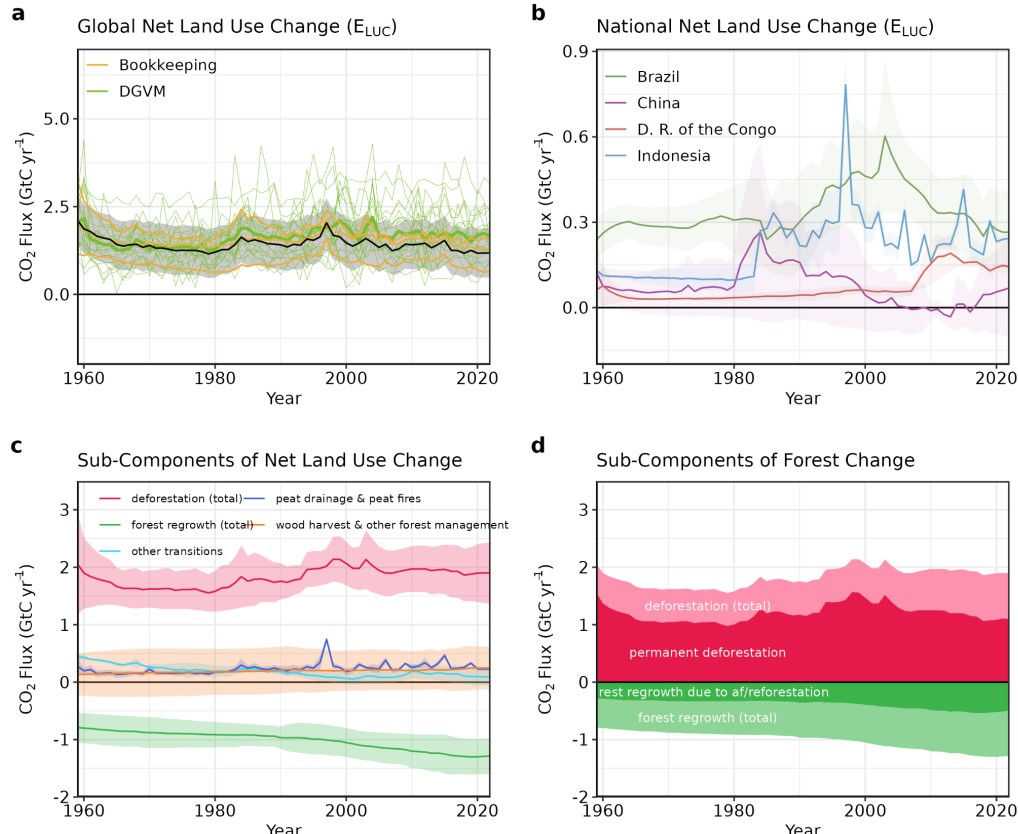

**Figure 7.** Net CO2 exchanges between the atmosphere and the terrestrial biosphere related to land use change.

(a) Net CO2 emissions from land-use change ($E_{LUC}$) with estimates from the three bookkeeping models (yellow lines) and the budget estimate (black with ±1σ uncertainty), which is the average of the three bookkeeping models. Estimates from individual DGVMs (narrow green lines) and the DGVM ensemble mean (thick green line) are also shown. (b) Net CO2 emissions from land-use change from the four countries with largest cumulative emissions since 1959. Values shown are the average of the three bookkeeping models, with shaded regions as ±1σ uncertainty. (c) Sub-components of $E_{LUC}$: (i) emissions from deforestation (including permanent deforestation and deforestation in shifting cultivation cycles), (ii) emissions from peat drainage & peat fires, (iii) removals from forest (re-)growth (including forest (re-)growth due to afforestation and reforestation and forest regrowth in shifting cultivation cycles), (iv) fluxes from wood harvest and other forest management (comprising slash and product decay following wood harvest, regrowth after wood harvest, and fire suppression), and (v) emissions and removals related to other land-use transitions. The sum of the five components is $E_{LUC}$ shown in panel (a). (d) Sub-components of 'deforestation (total)' and of 'forest (re-)growth (total)': (i) deforestation in shifting cultivation cycles, (ii) permanent deforestation, (iii) forest (re-)growth due to afforestation and/or reforestation, and (iv) forest regrowth in shifting cultivation cycles.

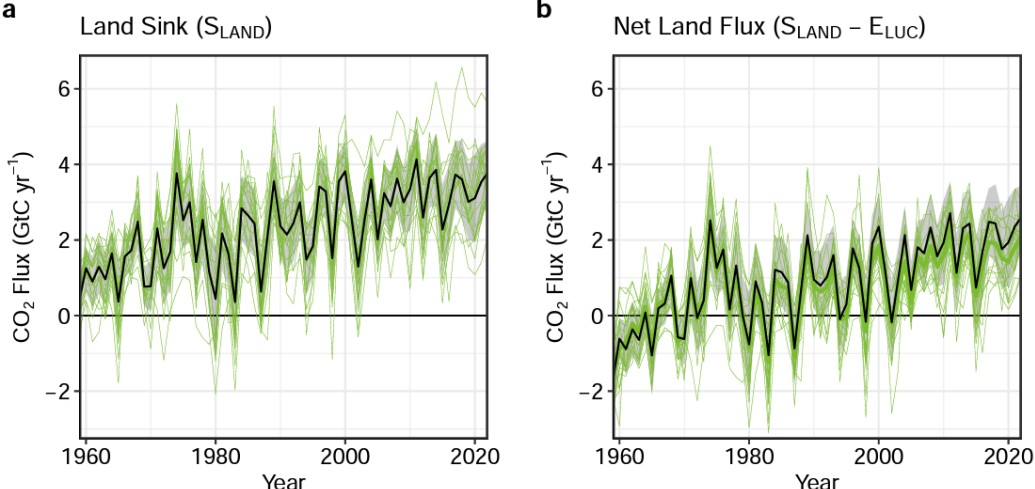


**Figure 8:** (a) The land $CO_2$ sink ($S_{LAND}$) estimated by individual DGVMs (green), as well as the budget estimate
(black with ±1σ uncertainty), which is the average of all DGVMs. (b) Net atmosphere-land $CO_2$ fluxes ($S_{LAND}$ −
$E_{LUC}$). The budget estimate of the net land flux (black with ±1σ uncertainty) combines the DGVM estimate of
$S_{LAND}$ from panel (a) with the bookkeeping estimate of $E_{LUC}$ from Figure 7a. Uncertainties are similarly
propagated in quadrature. DGVMs also provide estimates of $E_{LUC}$ (see Figure 7a), which can be combined with
their own estimates of the land sink. Hence panel (b) also includes an estimate for the net land flux for
individual DGVMs (thin green lines) and their multi-model mean (thick green line).


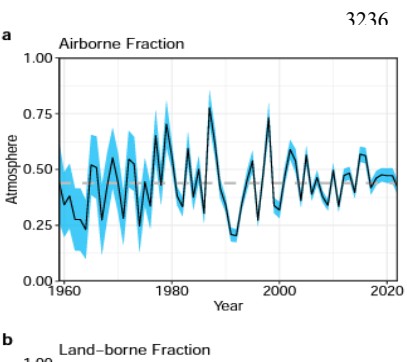

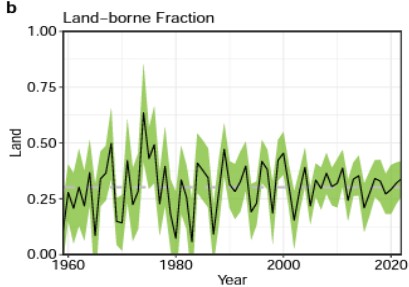

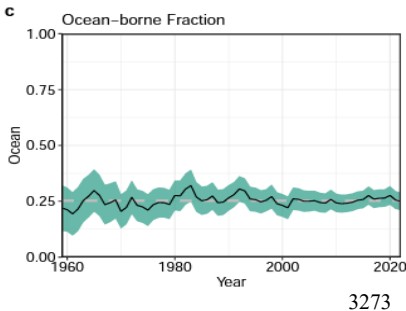



**Figure 9.** The partitioning of total anthropogenic $CO_2$ emissions ($E_{FOS}$ + $E_{LUC}$) across (a) the atmosphere
(airborne fraction), (b) land (land-borne fraction), and (c) ocean (ocean-borne fraction). Black lines represent the
central estimate, and the coloured shading represents the uncertainty. The grey dashed lines represent the long-
term average of the airborne (44%), land-borne (30%) and ocean-borne (25%) fractions during 1960-2022 (with
a BIM of 1%).



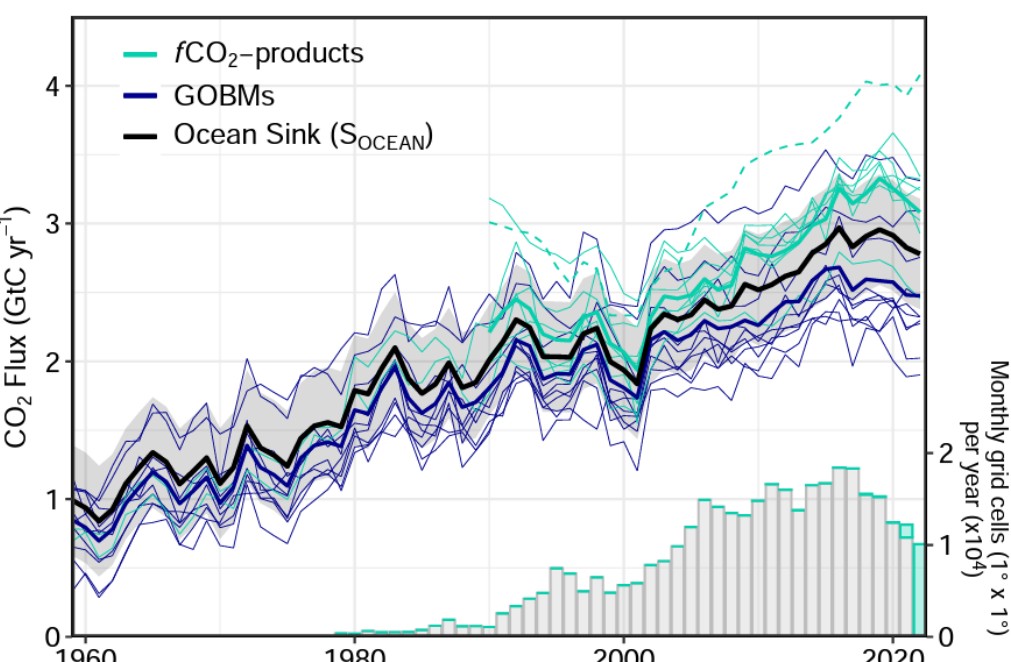

**Figure 10.** Comparison of the anthropogenic atmosphere-ocean $CO_2$ flux showing the budget values of $S_{OCEAN}$ (black; with the uncertainty in grey shading), individual ocean models (royal blue), and the ocean $fCO_2$-products (cyan; with Watson et al. (2020) in dashed line as not used for ensemble mean). Only one $fCO_2$-product (Jena-MLS) extends back to 1959 (Rödenbeck et al., 2022). The $fCO_2$-products were adjusted for the pre-industrial ocean source of $CO_2$ from river input to the ocean, by subtracting a source of 0.65 GtC yr$^{-1}$ to make them comparable to $S_{OCEAN}$ (see Section 2.5). Bar-plot in the lower right illustrates the number of $fCO_2$ observations in the SOCAT v2023 database (Bakker et al., 2023). Grey bars indicate the number of data points in SOCAT v2022, and coloured bars the newly added observations in v2023.



Earth System
Open Access  Science  Discussions
Data

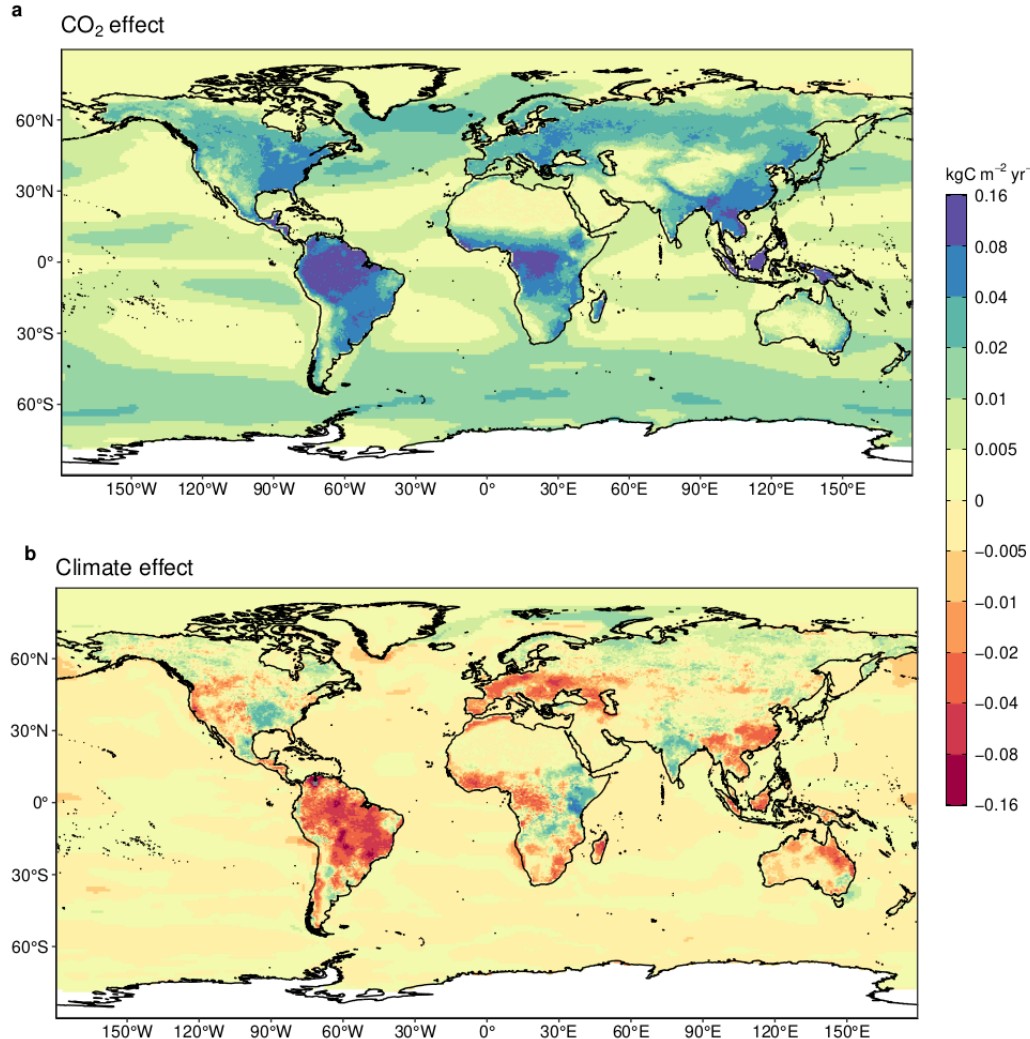

3291

**Figure 11.** Attribution of the atmosphere-ocean (S$_{OCEAN}$) and atmosphere-land (S$_{LAND}$) CO$_2$ fluxes to (a)

increasing atmospheric CO$_2$ concentrations and (b) changes in climate, averaged over the previous decade 2013-

2022. All data shown is from the processed-based GOBMs and DGVMs. Note that the sum of ocean CO$_2$ and

climate effects shown here will not equal the ocean sink shown in Figure 6 which includes the $f$CO$_2$-products.

See Supplement S.3.2 and S.4.1 for attribution methodology. Units are in kgC m$^{-2}$ yr$^{-1}$ (note the non-linear

colour scale).

3298

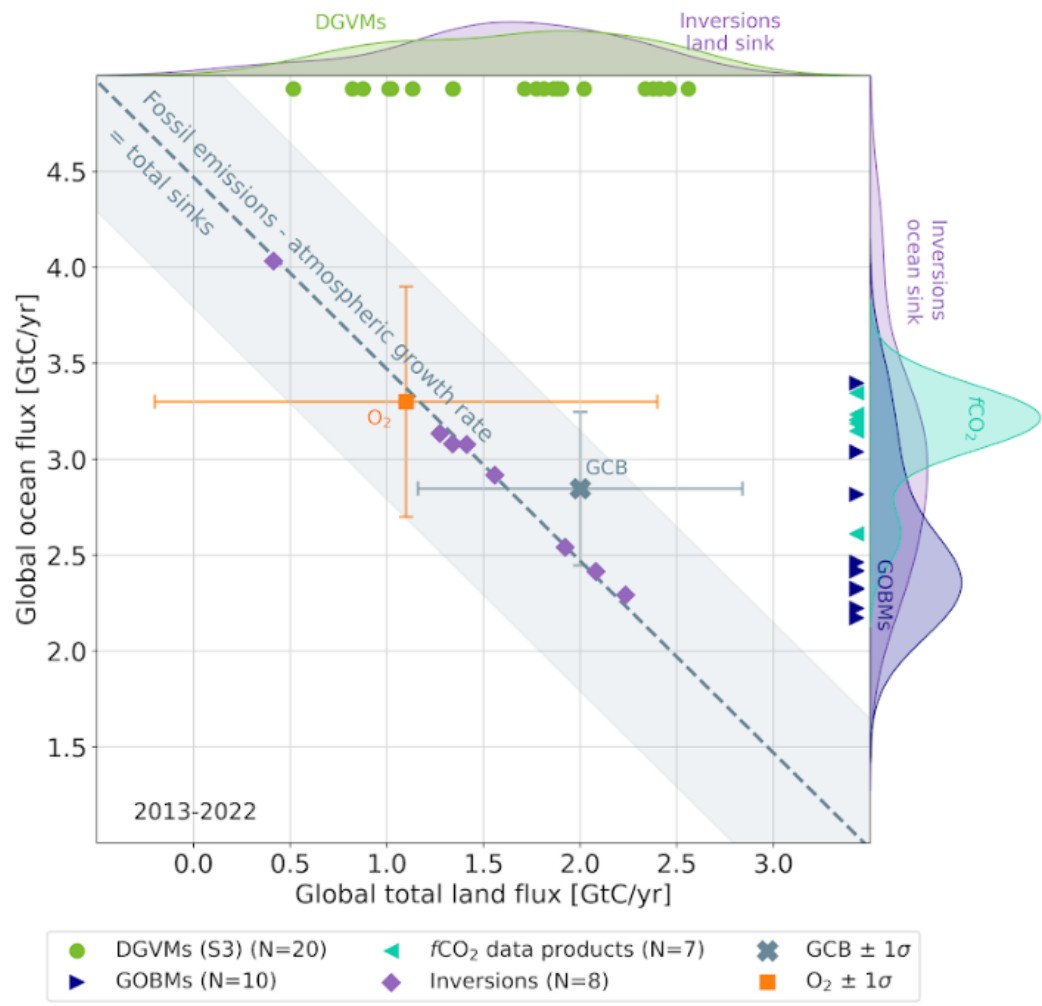

3299

**Figure 12.** The 2013-2022 decadal mean net atmosphere-ocean and atmosphere-land fluxes derived from the ocean models and $f\mathrm{CO_2}$ products (y-axis, right and left pointing blue triangles respectively), and from the DGVMs (x-axis, green symbols), and the same fluxes estimated from the inversions (purple symbols). The shaded distributions show the densities of the ensembles of individual estimates. The grey central cross is the mean ($\pm 1\sigma$) of $S_{OCEAN}$ and ($S_{LAND} - E_{LUC}$) as assessed in this budget. The grey diagonal line represents the global land + ocean net flux, i.e. global fossil fuel emissions minus the atmospheric growth rate from this budget ($E_{FOS} - G_{ATM}$). The orange square represents the ocean and land sink as estimated from the atmospheric $\mathrm{O_2}$ constraint. Positive values are $\mathrm{CO_2}$ sinks. Note that the inverse estimates have been scaled for a minor difference between $E_{FOS}$ and GridFEDv2023.1 (Jones et al., 2023).

3309

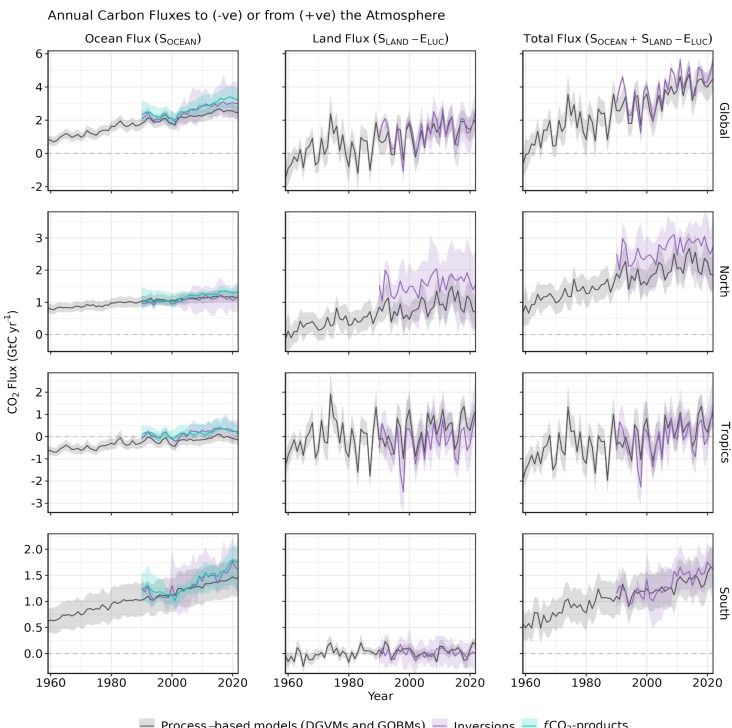

3310

**Figure 13.** $CO_2$ fluxes between the atmosphere and the Earth's surface separated between land and oceans,
globally and in three latitude bands. The ocean flux is $S_{OCEAN}$ and the land flux is the net atmosphere-land fluxes
from the DGVMs. The latitude bands are (top row) global, (2nd row) north (>30°N), (3rd row) tropics (30°S-
30°N), and (bottom row) south (<30°S), and over ocean (left column), land (middle column), and total (right
column). Estimates are shown for: process-based models (DGVMs for land, GOBMs for oceans); inversion
systems (land and ocean); and $f$CO$_2$-products (ocean only). Positive values are $CO_2$ sinks. Mean estimates from
the combination of the process models for the land and oceans are shown (black line) with ±1 standard deviation
(1σ) of the model ensemble (grey shading). For the total uncertainty in the process-based estimate of the total
sink, uncertainties are summed in quadrature. Mean estimates from the atmospheric inversions are shown
(purple lines) with their full spread (purple shading). Mean estimates from the $f$CO$_2$-products are shown for the
ocean domain (light blue lines) with full model spread (light blue shading). The global $S_{OCEAN}$ (upper left) and
the sum of $S_{OCEAN}$ in all three regions represents the anthropogenic atmosphere-to-ocean flux based on the
assumption that the preindustrial ocean sink was 0 GtC yr$^{-1}$ when riverine fluxes are not considered. This
assumption does not hold at the regional level, where preindustrial fluxes can be significantly different from
zero. Hence, the regional panels for $S_{OCEAN}$ represent a combination of natural and anthropogenic fluxes. Bias-
correction and area-weighting were only applied to global $S_{OCEAN}$; hence the sum of the regions is slightly
different from the global estimate (<0.05 GtC yr$^{-1}$).

3328





**Figure 14.** Decadal mean (a) land and (b) ocean fluxes for RECCAP-2 regions over 2013-2022. For land fluxes, $S_{LAND}$ is estimated by the DGVMs (green bars), with the error bar as $\pm 1\sigma$ spread among models. A positive $S_{LAND}$ is a net transfer of carbon from the atmosphere to the land. $E_{LUC}$ fluxes are shown for both DGVMs (green) and bookkeeping models (orange), again with the uncertainty calculated as the $\pm 1\sigma$ spread. Note, a positive $E_{LUC}$ flux indicates a loss of carbon from the land. The net land flux is shown for both DGVMs (green) and atmospheric inversions (purple), including the full model spread for inversions. The net ocean sink ($S_{OCEAN}$) is estimated by GOBMs (royal blue), $f$CO$_2$-products (cyan), and atmospheric inversions (purple). Uncertainty is estimated as the $\pm 1\sigma$ spread for GOBMs, and the full model spread for the other two products. The dotted lines show the $f$CO$_2$-products and inversion results without river flux adjustment. Positive values are CO$_2$ sinks.

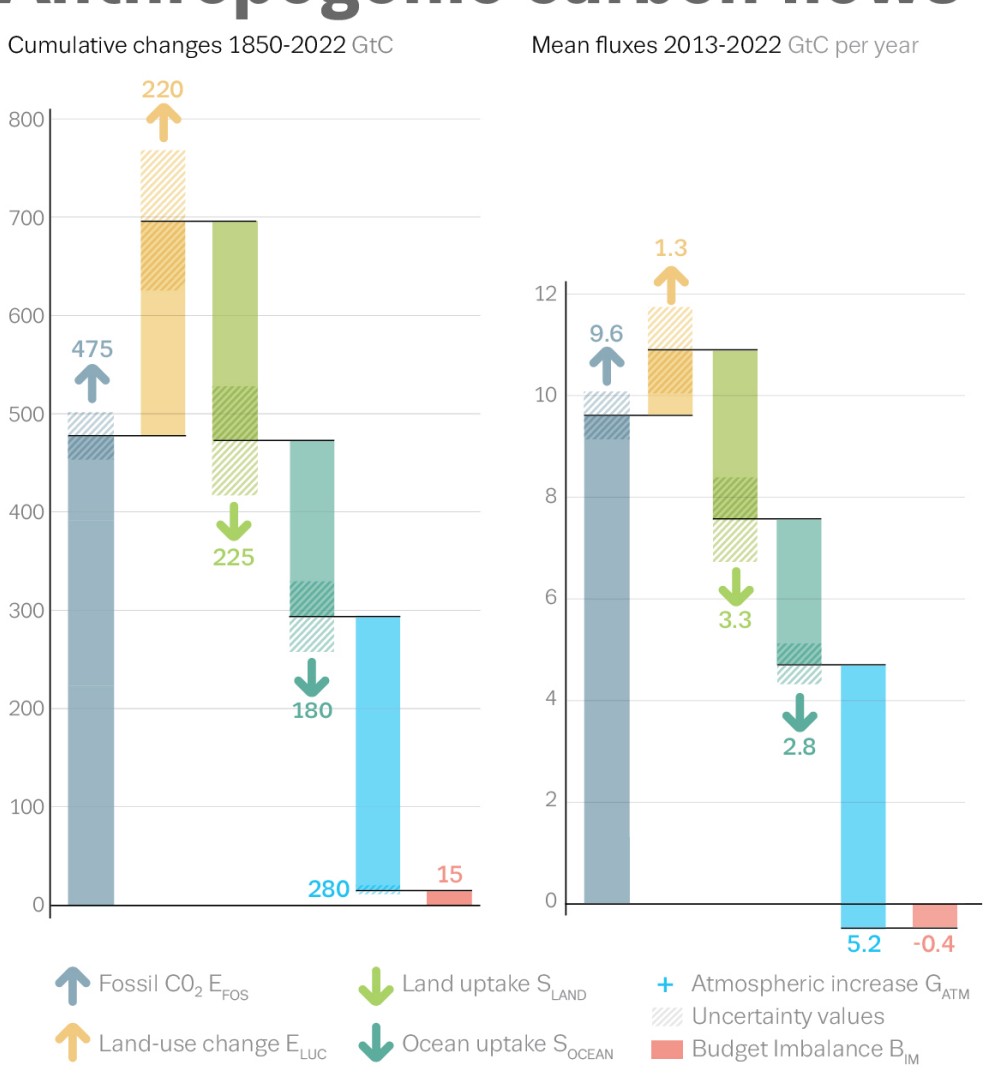

**Figure 15.** Cumulative changes over the 1850-2022 period (left) and average fluxes over the 2013-2022 period (right) for the anthropogenic perturbation of the global carbon cycle. See the caption of Figure 3 for key information and the methods in text for full details.

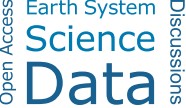

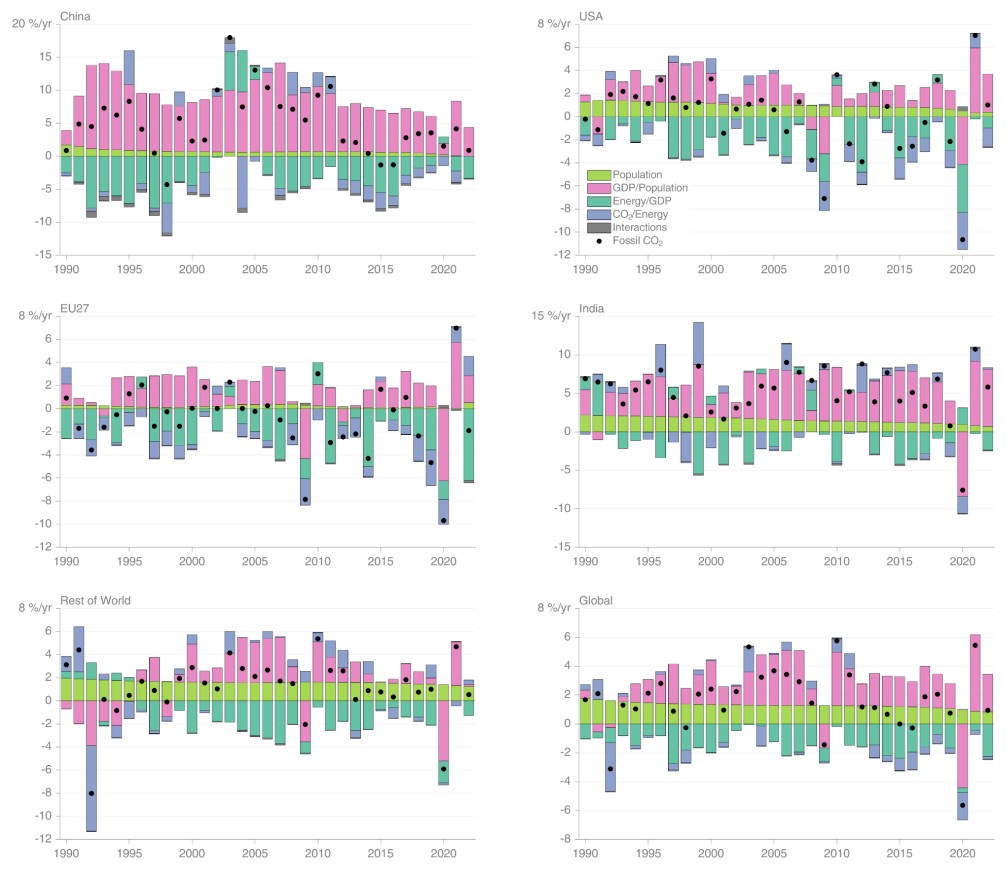

**Figure 16.** Kaya decomposition of the main drivers of fossil $CO_2$ emissions, considering population, GDP per person, Energy per GDP, and $CO_2$ emissions per energy, for China (top left), USA (top right), EU27 (middle left), India (middle right), Rest of the World (bottom left), and World (bottom right). Black dots are the annual fossil $CO_2$ emissions growth rate, coloured bars are the contributions from the different drivers. A general trend is that population and GDP growth put upward pressure on emissions, while energy per GDP and, more recently, $CO_2$ emissions per energy put downward pressure on emissions. Both the COVID-19 induced changes during 2020 and the recovery in 2021 led to a stark contrast to previous years, with different drivers in each region. The EU27 had strong Energy/GDP improvements in 2022.