# Peer review of "Global Carbon Budget 2023"

_Earth System Science Data, 2023_

## Referee Comment (RC1)

Overall, very positive about this effort! Authors conduct (annually) a massive data gathering, comparison and access effort, with outcomes of enormous value to researchers, readers, and - one hopes - policy makers. I make a few technical comments/suggestions then raise two more-substantial issues at the end. None of my comments detracts from excellent open-access product!

Working from 'track-changes' version which helps on initial text sections but proves more difficult for reviewers after graphics sections.

This review starts with picky' critique of abstract language knowing that many readers will look only at abstract.

Lines 780, 781 (Abstract): "a measure of imperfect data and understanding of the contemporary carbon cycle." Small confusion here? Adjective 'imperfect' applies only to following subject 'data' or to both 'data' and 'understanding'? BIM derives from both imperfect data and (remaining) weaknesses in understanding but readers can, unfortunately, interpret this phrase in either manner? Perhaps my misunderstanding or perhaps needs slight adjustment?

Lines 786, 787 (Abstract): same confusion as above? Here 'sources' "marginally" too low or sinks 'marginally' (?) too high. If, in this case, modifier 'marginally' applies to 'too low' and 'too high' (as this reader finds appropriate)? With BIM relatively small due to sources too low or sinks too high (authors can't specify which), additional small clarity would help? Much later, in discussion (around line 4016 in track-changes version), authors discuss possibilities of over-estimate of emissions vs under-estimate of sinks? Do those later more-careful assessments accord with what readers will encounter here in abstract? Small clarifications, please.

Line 787 (Abstract): "global atmospheric CO2 concentration averaged over 2022 reached 417.1 ± 0.1 ppm." Readers learned in previous sentence that $G_{ATM}$ rose by 4.6 GtC yr$^{-1}$. Present sentence references a cumulative increase (averaged over calendar year 2022) to 417 ppm, but not $G_{ATM}$? Clarify for many readers? Subsequent sentence (Line 789) reports this better?

Lines 791, 792 (Abstract): "although discrepancies of up to around 1 GtC yr-1 persist for the representation of annual to semi-decadal variability in CO2 fluxes." Shorten and sharpen as 'although discrepancies of around 1 GtC yr-1 persist for representation of annual to semi-decadal variability of CO2 fluxes'? Authors choice …

Lines 796 to 798 (Abstract): Again, shorten and sharpen as "This living data update documents changes in methods and data sets applied to this most-recent global carbon budget as well as evolving community understanding of the global carbon cycle."

Line 829: Confusion on behalf of this reader. Apparently we assign 2019 as pre-pandemic, 2020 and 2021 as pandemic, and 2022 as post-pandemic. Thus 1%

increase in 2022 returned $E_{FOS}$ to 2019 (pre-pandemic) values, while projected 1.2% increase in 2023 will result in $E_{FOS}$ 1.5% above 2019? Have I missed something here? Cumulative uncertainties?

Line 836: If these data describe "expected" increases or decreases, shouldn't they refer to 2023, not to 2022?

Line 843: One hates to get caught up in acronyms, but technically doesn't LULUCF better describe "land-use, land-use change, and forestry"? I hope I have not opened can of worms that persists through remainder of this manuscript but I try to better understand acronyms as currently applied in IPCC reports?

Line 860: Readers learned in abstract that $G_{ATM}$ increased by 4.6 GtC yr$^{-1}$ (2.2 increase in ppm) to 417.1 ppm in 2022 plus anticipated increase to 419.2 in 2023. Here authors project another 1.9 ppm increase which would indicate 419 rather than 419.2? 419.2 would instead indicate 2.1 ppm increase, requiring more than 4 GtC yr$^{-1}$ of additional emissions? All these estimates fall within uncertainty noise? Note: Daily CO2, visible to authors as easily as to this reviewer, shows CO2 effective 15 October 2023 at 419.91, +3.96 from concentration one year prior. Hard to keep up with highly-variable daily concentrations but - for that reason if no other - some caution on highly-precise predictions?

Lines 1130 to1134: CDR remains "several orders of magnitude smaller than the other components of the budget" but nevertheless merits mention for "illustrative purposes"? Why not include it instead in 'processes not included' section (Section 2.10, line 1800)? This reviewer understands that some CDR processes now qualify as quantitative where only speculative in the past but still fall below uncertainties?

Line 3595: readers need brief introduction to RECCAP2? At least the RECCAP url?

Line 3596: RECCAP2 does not include ENSO processes/signals? No mention here.

Line 3597 to 3697: another numbering gap?

Line 3703: "loss". Sink or source? Readers will expect better precision in language.

Line 3704: word missing here?

Lines 3711 to 3712: "higher" as used here means lower differences among data sources for ocean sink terms compared to each other (e.g. GOBM not greatly different from SOCAT etc), or compared to land terms, or both. Ambiguity here will not help readers.

Lines 3713 to 3714: sloppy language. Largest from Southern Ocean but important contributions from 'vast' Atlantic and Pacific. Not possible to compare ocean basin areas using Fig 14 due to artificially-magnified high-latitude regions in that projection.

Lines 3727 to 3731: important qualification. Move this ahead of data reports rather than after?

Harder to certify all figure numbers and changed figure numbers in track-changes version. Clean manuscript shows very good organization. Authors should check that all text refers to proper figures?

Personally, this reviewer prefers living data versions. But, manuscript seems to have accumulated 'older' language and conclusions. Authors have made good efforts to shorten standard version, by moving sections to supplement, etc. but manuscript as a whole needs fresh overview? Perhaps not the next version but soon we need careful systematic top-down revision and full review?

My second worried comment has to do with time scale of changes in emissions or sinks. Again, not a criticism of current manuscript. None of speculations that follow could have occurred without heroic efforts by authors and full open access of data!

About ocean or land sinks, perspective strongly depends on decadal vs. annual time scales? Authors might provide correct account of decadal change starting in 2002 but more-recent changes (e.g. Figs 3, 4, esp. Fig 10!) suggest a recent decline in ocean sink? Fig 4 same for land? "Did not grow since 2019" (line 1049) seems like a more-accurate assessment? After surprisingly steady rises 2002/2003 to roughly 2016, ocean sink has declined (or, "not grown") since 2016? Or, have we moved back into period of higher annual variability? Whatever reasons or reality, this reader feels that data do not support statements about persistent decadal increases. NOT a criticism of these budget efforts, just a caution that we might not know ocean or land processes as well as we might hope? If readers take a step back, land and ocean sinks may not behave in manner assumed and described here? As correctly stated, inversions force an either-or scenario: land sinks increase while ocean sinks decrease or vice-versa? But, under that scenario, one could not observe decreases in both sinks? Remaining uncertainties in satellite-determined CO2 (forcing term for inversion), remain, as for atmospheric O2 measurement, too uncertain to provide assistance here? Overall, authors seems to invoke one (often, ENSO) process but ignore or dismiss it in later paragraphs. Because BIM remains small (never greater than 1 GtC yr-1 [never greater than 0.5 ppm in atmos concentration terms]), particularly recently, do we not need to at least admit the possibility that we might miss some processes outside of current budget estimates? Or that data sources prove unreliable? I fear I react too strongly to Zeke Hausfather's recent editorial letter in NYT wherein he proposes unusual (steeper) warming over past 15 years. (He could equally, in this readers' view, have proposed steeper warming since 2016.) Has our formerly 'balanced' system of atmospheric CO2 concentrations, emissions, sinks, ocean and atmospheric circulations, etc. changed recently? If, as these authors repeatedly caution, we need a decade to certify real change, don't we in fact hold early evidence that system may have shifted? Many statements here bear on (unfortunately) both sides of this issue: "The evolution of AF [airborne fraction] over the last 60 years shows no significant trend. (line 2651)"; no

change in ocean sink (line 2753, confusing paragraph); decreasing trend ocean sink over prior three years (line 3020); climate changes induce global reduction of land sink (line 3267); but broadly constant land sink over past six decades (line 3302); "all components except land-use change emissions have grown" (line 3840). I defer to authors mastery of specific processes and data sources but must ask for small caution in all these claims in view of possibility already stated: land and ocean systems that we work so hard to understand and quantify may actually have changed? Authors here do admirable job of calculating uncertainties and comparing data sources! Amidst that great effort, have we missed early signs of systematic changes? I repeat: none of these speculation possible without reliable data as compiled by these authors and published by this journal!

Reference above:

https://www.nytimes.com/2023/10/13/opinion/climate-change-excessive-heat-2023.html: Zeke Hausfather, Berkeley Earth. Also: https://doi.org/10.5194/essd-12-3469-2020.

---

## Author Comment (AC1)

Response to Reviewer 1 (in bold)

Lines 780, 781 (Abstract): "a measure of imperfect data and understanding of the contemporary carbon cycle." Small confusion here? Adjective 'imperfect' applies only to following subject 'data' or to both 'data' and 'understanding'? BIM derives from both imperfect data and (remaining) weaknesses in understanding but readers can, unfortunately, interpret this phrase in either manner? Perhaps my misunderstanding or perhaps needs slight adjustment?
**Thank you. Changed to " is a measure of imperfect data and incomplete understanding of the contemporary carbon cycle"**

Lines 786, 787 (Abstract): same confusion as above? Here 'sources' "marginally" too low or sinks 'marginally' (?) too high. If, in this case, modifier 'marginally' applies to 'too low' and 'too high' (as this reader finds appropriate)? With BIM relatively small due to sources too low or sinks too high (authors can't specify which), additional small clarity would help? Much later, in discussion (around line 4016 in track-changes version), authors discuss possibilities of over-estimate of emissions vs under-estimate of sinks? Do those later more-careful assessments accord with what readers will encounter here in abstract? Small clarifications, please.
**Thank you. Changed to "(i.e. total estimated sources marginally too low or sinks marginally too high)." Indeed, for a particular year (as 2022 here) we cannot say if a slightly negative BIM (-0.1GtC) is due to sources being underestimated and/or sinks being overestimated**.

Line 787 (Abstract): "global atmospheric CO2 concentration averaged over 2022 reached 417.1 ± 0.1 ppm." Readers learned in previous sentence that GATM rose by 4.6 GtC yr-1. Present sentence references a cumulative increase (averaged over calendar year 2022) to 417 ppm, but not GATM? Clarify for many readers? Subsequent sentence (Line 789) reports this better?
**GATM is the atmospheric CO_ growth rate, i.e. the change in atmospheric CO_ concentration, from 2021 to 2022, expressed in GtC per year. The 417.7 ppm figure is the global atmospheric concentration for 2022 (annual average of monthly CO2 concentrations).**

Lines 791, 792 (Abstract): "although discrepancies of up to around 1 GtC yr-1 persist for the representation of annual to semi-decadal variability in CO2 fluxes." Shorten and sharpen as 'although discrepancies of around 1 GtC yr-1 persist for representation of annual to semi-decadal variability of CO2 fluxes'? Authors choice ...
**Thank you for the suggestion, but we prefer our formulation "up to around 1 GtC yr-1"**

Lines 796 to 798 (Abstract): Again, shorten and sharpen as "This living data update documents changes in methods and data sets applied to this most-recent global carbon budget as well as evolving community understanding of the global carbon cycle."
**Done, thank you**

Line 829: Confusion on behalf of this reader. Apparently we assign 2019 as pre- pandemic, 2020 and 2021 as pandemic, and 2022 as post-pandemic. Thus 1% increase in 2022 returned EFOS to 2019 (pre-pandemic) values, while projected 1.2% increase in 2023 will result in EFOS 1.5% above 2019? Have I missed something here? Cumulative uncertainties?
**Rounding errors indeed. 2022 was already marginally (0.3%) above 2019.**

Line 836: If these data describe "expected" increases or decreases, shouldn't they refer to 2023, not to 2022?
**They do refer to 2023: Emissions from coal, oil, and gas in 2023 are expected to be slightly above their 2022 levels.**

Line 843: One hates to get caught up in acronyms, but technically doesn't LULUCF better describe "land-use, land-use change, and forestry"? I hope I have not opened can of worms that persists through remainder of this manuscript but I try to better understand acronyms as currently applied in IPCC reports?
**Agreed, text updated as suggested.**

Line 860: Readers learned in abstract that GATM increased by 4.6 GtC yr-1 (2.2 increase in ppm) to 417.1 ppm in 2022 plus anticipated increase to 419.2 in 2023. Here authors project another 1.9 ppm increase which would indicate 419 rather than 419.2? 419.2 would instead indicate 2.1 ppm increase, requiring more than 4 GtC yr-1 of additional emissions? All these estimates fall within uncertainty noise? Note: Daily CO2, visible to authors as easily as to this reviewer, shows CO2 effective 15 October 2023 at 419.91, +3.96 from concentration one year prior. Hard to keep up with highly-variable daily concentrations but - for that reason if no other - some caution on highly-precise predictions?
**The annual growth rate and the annual change in the global averaged CO2 concentration are not strictly identical. We follow the NOAA methodology. For example, the annual growth rate of 2022 is calculated as the average of December 2022 and January 2023 concentrations minus the average of December 2021 and January 2022 concentrations. While the annual change in the global averaged CO2 concentration would be the average of the twelve months concentrations in 2022 minus be the average of the twelve months concentrations in 2021. Hence the potential small difference between the two methods.**

Lines 1130 to1134: CDR remains "several orders of magnitude smaller than the other components of the budget" but nevertheless merits mention for "illustrative purposes"? Why not include it instead in 'processes not included' section (Section 2.10, line 1800)? This reviewer understands that some CDR processes now qualify as quantitative where only speculative in the past but still fall below uncertainties?
**CDR is now assessed in this budget. See sections 2.3 and 3.3.**

Line 3595: readers need brief introduction to RECCAP2? At least the RECCAP url?
**RECCAP-2 has already been defined before, with reference to Ciais et al. (2020)**

Line 3596: RECCAP2 does not include ENSO processes/signals? No mention here.
**Indeed, we only assess the decadal averaged (2013-2022) estimates of SLAND and SOCEAN over the RECCAP-2 regions, not the variability.**

Line 3597 to 3697: another numbering gap?
**Indeed. No text missing though.**

Line 3703: "loss". Sink or source? Readers will expect better precision in language.
**Thank you. Text replaced to "source".**

Line 3704: word missing here?
**Indeed, thank you.**

Lines 3711 to 3712: "higher" as used here means lower differences among data sources for ocean sink terms compared to each other (e.g. GOBM not greatly different from SOCAT etc), or compared to land terms, or both. Ambiguity here will not help readers
**Rephrased, thank you.**

Lines 3713 to 3714: sloppy language. Largest from Southern Ocean but important contributions from 'vast' Atlantic and Pacific. Not possible to compare ocean basin areas using Fig 14 due to artificially-magnified high-latitude regions in that projection.

**Thank you. Rephrased as follow : "All data streams agree that the largest contribution to SOCEAN stems from the Southern Ocean due to a combination of high flux density and large surface area, but with important contributions also from the Atlantic (high flux density) and Pacific (large area) basins."**

Lines 3727 to 3731: important qualification. Move this ahead of data reports rather than after?
**We prefer to leave it after. it is a caveat, but it doesn't affect the overall results presented here.**

Harder to certify all figure numbers and changed figure numbers in track-changes version. Clean manuscript shows very good organization. Authors should check that all text refers to proper figures?
**We checked again. Thank you.**

Personally, this reviewer prefers living data versions. But, manuscript seems to have accumulated 'older' language and conclusions. Authors have made good efforts to shorten standard version, by moving sections to supplement, etc. but manuscript as a whole needs fresh overview? Perhaps not the next version but soon we need careful systematic top-down revision and full review?
**We did a community survey of the GCB report in 2021 and implemented substantial changes in 2022. We made additional changes this year (use of oxygen, ESMs, Analysis over RECCAP2 regions, moved more material in SI). Not sure a "systematic top-down" revision is needed now.**

My second worried comment has to do with time scale of changes in emissions or sinks. Again, not a criticism of current manuscript. None of speculations that follow could have occurred without heroic efforts by authors and full open access of data! About ocean or land sinks, perspective strongly depends on decadal vs. annual time scales? Authors might provide correct account of decadal change starting in 2002 but more-recent changes (e.g. Figs 3, 4, esp. Fig 10!) suggest a recent decline in ocean sink? Fig 4 same for land? "Did not grow since 2019" (line 1049) seems like a more- accurate assessment? After surprisingly steady rises 2002/2003 to roughly 2016, ocean sink has declined (or, "not grown") since 2016? Or, have we moved back into period of higher annual variability? Whatever reasons or reality, this reader feels that data do not support statements about persistent decadal increases. NOT a criticism of these budget efforts, just a caution that we might not know ocean or land processes as well as we might hope? If readers take a step back, land and ocean sinks may not behave in manner assumed and described here? As correctly stated, inversions force an either-or scenario: land sinks increase while ocean sinks decrease or vice-versa? But, under that scenario, one could not observe decreases in both sinks? Remaining uncertainties in satellite-determined CO2 (forcing term for inversion), remain, as for atmospheric O2 measurement, too uncertain to provide assistance here? Overall, authors seems to invoke one (often, ENSO) process but ignore or dismiss it in later paragraphs. Because BIM remains small (never greater than 1 GtC yr-1 [never greater than 0.5 ppm in atmos concentration terms]), particularly recently, do we not need to at least admit the possibility that we might miss some processes outside of current budget estimates? Or that data sources prove unreliable? I fear I react too strongly to Zeke Hausfather's recent editorial letter in NYT wherein he proposes unusual (steeper) warming over past 15 years. (He could equally, in this readers' view, have proposed steeper warming since 2016.) Has our formerly 'balanced' system of atmospheric CO2 concentrations, emissions, sinks, ocean and atmospheric circulations, etc. changed recently? If, as these authors repeatedly caution, we need a decade to certify real change, don't we in fact hold early evidence that system may have shifted? Many statements here bear on (unfortunately) both sides of this issue: "The evolution of AF [airborne fraction] over the last 60 years shows no significant trend. (line 2651)"; no change in ocean sink (line 2753, confusing paragraph); decreasing trend ocean sink over prior three years (line 3020); climate changes induce global reduction of land sink (line 3267); but broadly constant land sink over past six

decades (line 3302); "all components except land-use change emissions have grown" (line 3840). I defer to authors mastery of specific processes and data sources but must ask for small caution in all these claims in view of possibility already stated: land and ocean systems that we work so hard to understand and quantify may actually have changed? Authors here do admirable job of calculating uncertainties and comparing data sources! Amidst that great effort, have we missed early signs of systematic changes? I repeat: none of these speculation possible without reliable data as compiled by these authors and published by this journal! Reference above: https://www.nytimes.com/2023/10/13/opinion/climate-change-excessive-heat-2023.html: Zeke Hausfather, Berkeley Earth. Also: https://doi.org/10.5194/ essd-12-3469-2020."

**We respectfully disagree with the reviewer that the ocean or land sink shows a recent decline. It is clear from Table 7 that both the land and the ocean sinks have been increasing over the last 40 years. What the reviewer probably refers to are the last few years (i.e. since 2019 for the ocean). This is due to natural variability as explained in the manuscript. Nevertheless, the ocean sink over the 2013-2022 decadal average is higher than over the previous decade (see tables 6 and 7). It is not clear what the reviewer means by " inversions force an either-or scenario: land sinks increase while ocean sinks decrease or vice-versa". Inversions show increase in both land and ocean decadal sinks (tables 5 and 6). Likewise, we don't understand the statement from the reviewer than because the BIM is small, we should "at least admit the possibility that we might miss some processes outside of current budget estimates". It is quite the opposite; a small BIM implies that we are not missing any significant process. On the airborne fraction, it looks like the reviewer confuses ocean sink with ocean-borne fraction (line 2753) and land sink with land-borne fraction (line 3302). The rest of this comment seems more speculative about potential unobserved "early signals of systematic changes". Unclear what changes the reviewers is referring to.**

---

## Author Comment (AC3)

Response to Reviewer 2 (in bold)

The authors are to be again greatly complimented for their work, for the outstanding number of data sources used, performed analysis, as well as for the continuous inclusion of new products. I was again a bit overwhelmed with the length of the paper, however, it looks like it shortened a little compared to previous versions. I find the Executive summary and the highlighted key messages of great use. I would only suggest consistency between the information provided in each paragraph, as highlighted below, in the line by line suggestions.
**Thank you for the very positive overall comments.**

Abstract: great to see the inclusion of the ESMs and CDRs estimates as well as inversion systems using both satellites and surface observations (OCO-2 and GOSAT).
**Thank you.**

L846: Please add the value for deforestation in 2022 compared to 2019 similarly done for the CO2 fossil?
**We give the 2019 estimate for fossil fuel as fossil fuel emissions decreased significantly in 2020 because of the COVID pandemic and 2023 is the first year where fossil fuel emissions are above the pre-pandemic level. There is no equivalent for land use change emissions,**

L789 and L858: the increased concentration of CO2 is 51%, could be mentioned as well on line 789 instead of saying more than 50%
**Done, thank you.**

In general I agree with RC1 about comparing estimates for the pre-, post- and pandemic years, if authors want to exclude pandemic years as being atypical, then only 2022 compared to 2019 is enough, with a clear sentence of projection (2023) in the end.
**Unclear what the reviewer is asking here. We certainly do not want to exclude the pandemic years. Reviewer 1 comment was about potential rounding error between 2022 and 2023 estimates.**

L1197: because 2023 is a projection, I would think of using 2022 instead, to compare it with 2019.
**Unclear what the comment refers to. Line 1197 describes the methodology and its changes over the successive global carbon budget publications (from 2019 to 2023 in Table 3 and before 2019 in Table S8).**

L1533: first time RECCAP is mentioned, please add the weblink
**There isn't a dedicated website for RECCAP-2 (apart from the generic global carbon project website). It seems more appropriate to give the reference to Ciais et al. 2020). We now also refer to Poulter et al., 2022 which also describes the RECCAP-2 activity.**

L1638,1641,1704 etc.: consistent use of wording for the numbers throughout the manuscript is needed, now it's a mix of words and numbers.
**Thank you, we will double check.**

L2063: regarding the following paragraph "...relatively constant over the 1960-1999 period. Since the 1990s they have shown a slight decrease of about 0.1 GtC per decade, reaching 1.3 ± 0.7 GtC yr-1 for the 2013-2022 period (Table 7)" . What happened between 2000-2013? **Unclear what the reviewers asks. 2000-2013 is part of "Since the 1990s". ELUC emissions are declining by about 0.1 GtC per decade since the 1990s until now.**

L2297: Perhaps add the projection value of Powis et al., 2023 for blue carbon CDR? **Powis et al does not explicitly quantify blue carbon CDR. their estimate of 0.01 MtCO2yr−1, includes DACCs, mineralization, aquatic biomass growth, and others. Hence, we added "less than 0.003MtC yr-1" in the text.**

L2341 NGHGI already explained at L1333, L1047 and Tables...keep please the first and the rest NGHGI **Thank you. NGHGI is now defined only once at the first occurrence in the main text.**

L2341 and paragraphs after: the authors discuss the subtractions between DGVMs and bookkeeping models to match the NGHGIs estimates, I would suggest they mention that NGHGIs apply only to Annex I Parties while FAOSTAT is used for the non-Annex I. Also, FAOSTAT has global coverage, do they use a mix of the two? I would understand that the GCB estimates which match very closely the NGHGIs refer only to Annex I. **In our study "NGHGI" refers to both Annex-I and non-Annex I. We use the same methodology as described Grassi et al. 2023 (the NGHGI dataset upon which the NGHGI data in the GCB 2023 is based): "This database builds on a detailed analysis of a range of country submissions to the UNFCCC and is complemented by information on managed and unmanaged forest areas. Specifically, for Annex-I countries, data are from annual GHG inventories (including a complete time series from 1990 to 2020). For non-Annex I countries, the most recent and complete information was compiled from different sources, including national communications (NCs), biennial update reports (BURs), submissions to the framework REDD+ (Reducing Emissions from Deforestation and Forest Degradation) and NDCs" The Grassi et al. 2023 dataset is available at https://essd.copernicus.org/articles/15/1093/2023/**

L3594: Interesting inclusion of the RECCAP2 regions paragraph given that not all regions submitted their papers, I assume authors received the agreement of the chapter-lead authors to generate this preview of RECCAP2 results. I would suggest a sentence to clearly mention this. **We only show the ELUC, SLAND and SOCEAN estimates from this Global Carbon Budget paper over the RECCAP regions. We do not report any of the estimates from the RECCAP individual papers. This is clarified in the text now.**

L4049: "emission declines in the USA and the EU27 are primarily driven by slightly weaker economic growth" needs a reference ? **This statement is our analysis of Figure 16, the Kaya identity figure, which shows with lower per capita GDP in EU and US over this past decade than over the 1990s.**

Table 9: please explain to which countries you refer to for the NGHGIs **As explained above, NGHGI refers to both Annex-I and non-Annex I.**

Figure 2: please add a sentence to explain what the uncertainties represent
**Thank you, sentence added: "Fluxes estimates and their 1 standard deviation uncertainty are as reported in Table 7"**

Figure 16: I would add what positive/negative values mean
**Thank you. Text added to clarify.**